# Learning Safe Strategies for Value Maximizing Buyers in Uniform Price Auctions

**Negin Golrezaei** [1]   **Sourav Sahoo** [1]

## Abstract

We study the bidding problem in repeated uniform price multi-unit auctions from the perspective of a single *value-maximizing* buyer who aims to maximize their cumulative value over $T$ rounds while adhering to return-on-investment (RoI) constraints in each round. Buyers adopt $m$-*uniform bidding* format, where they submit $m$ bid-quantity pairs $(b_i, q_i)$ to demand $q_i$ units at bid $b_i$. We introduce *safe* bidding strategies as those that satisfy RoI constraints in every auction, regardless of competing bids. We show that these strategies depend only on the bidder's valuation curve, and the bidder can focus on a finite subset of this class without loss of generality. While the number of strategies in this subset is exponential in $m$, we develop a polynomial-time algorithm to learn the optimal safe strategy that achieves sublinear regret in the online setting, where regret is measured against a clairvoyant benchmark that knows the competing bids *a priori* and selects a fixed hindsight optimal safe strategy. We then evaluate the performance of safe strategies against a clairvoyant that selects the optimal strategy from a richer class of strategies in the online setting. In this scenario, we compute the *richness ratio*, $\alpha \in (0, 1]$ for the class of strategies chosen by the clairvoyant and show that our algorithm, designed to learn safe strategies, achieves $\alpha$-approximate sublinear regret against these stronger benchmarks. Experiments on semi-synthetic data from real-world auctions show that safe strategies substantially outperform the derived theoretical bounds, making them quite appealing in practice.

## 1. Introduction

In a uniform price multi-unit auction, the auctioneer sells $K$ identical units of a single good to buyers who may demand multiple units, with the per-unit price set at the $K^{th}$ highest bid. These auctions are widely used to allocate scarce resources in critical markets such as emissions permits, energy markets, and Treasury auctions. To succeed in these markets, bidders must develop effective bidding strategies that balance long-term value maximization with financial risk management, all while operating under limited information, uncertainty in competing bids and strategic (or adversarial) behavior of other bidders, among other challenges.

In this paper, we model the bidders as a value-maximizing agent with per-round return-on-investment (RoI) constraints. This model reflects real-world decision-making in industries where managing financial risks is as critical as maximizing value. Unlike traditional mechanism design, which assumes bidders with quasilinear utility (i.e., utility decreases linearly with payments), many practical settings involve bidders who optimize total value while adhering to financial constraints such as RoI or budgets. These scenarios frequently arise in industries where agents or algorithms bid on behalf of clients, optimizing for high-level objectives within strict constraints—a context akin to the principal-agent framework (Fadaei & Bichler, 2016; Aggarwal et al., 2024). Examples include autobidders in online advertising, where advertisers seek to maximize clicks while keeping the average cost per click below a threshold (Lucier et al., 2024; Balseiro et al., 2021a; Deng et al., 2023b), and consultants bidding for small firms in emission permit auctions (EEX, 2024).

Building on this, we study the bidding problem in repeated uniform price auctions over $T$ rounds, from the perspective of a single bidder who seeks to maximize cumulative value while adhering to per-round RoI constraints—that is, ensuring that the value obtained in each round is at least a fixed multiple of the corresponding payment (see Eq. (2)).

To address practical bidding interfaces, we assume bidders adopt $m$-*uniform bidding* format, a generalization of the uniform bidding format (De Keijzer et al., 2013; Birmpas et al., 2019) for some $m \in \mathbb{N}$. In a $m$-uniform strategy $\mathbf{b} := \langle (b_1, q_1), \dots, (b_m, q_m) \rangle$, bidders bid $b_1$ for the first $q_1$ units, $b_2$ for the next $q_2$ units and so on (see Definition 1).

### 1.1. Our Contributions

**Safe Bidding Strategies (Section 3).** To ensure that the bidder satisfies the RoI constraint without knowing the com-

[1]Massachusetts Institute of Technology, Cambridge, USA. Correspondence to: Sourav Sahoo <sourav99@mit.edu>.

*Proceedings of the 42$^{nd}$ International Conference on Machine Learning*, Vancouver, Canada. PMLR 267, 2025. Copyright 2025 by the author(s).

peting bids, we introduce the concept of *safe* bidding strategies. These strategies guarantee that the RoI constraint is met regardless of how the other participants bid. We then identify the class of $m$-uniform safe *undominated* bidding strategies, denoted by $\mathcal{S}_m^\star$ (Theorem 3.2), and demonstrate that the strategies within this class depend solely on the bidder's valuation vector $[v_1, \ldots, v_K]$ and exhibit a "nested" structure (illustrated in Fig. 1).

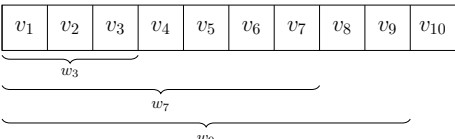

*Figure 1.* Nested structure of the bidding strategies in $\mathcal{S}_3^\star$. Consider the strategy $\mathbf{b} = \langle (w_3, 3), (w_7, 4), (w_9, 2) \rangle \in \mathcal{S}_3^\star$, where we note that $Q_1 = 3$, $Q_2 = 3 + 4 = 7$ and $Q_3 = 3 + 4 + 2 = 9$. The $j^{th}$ highest bid (i.e., $b_j$) is the average of the first $Q_j$ entries of the valuation vector, i.e., $b_j = w_{Q_j}$, where $w_j = \frac{1}{j} \sum_{\ell \le j} v_\ell$.

**Learning Safe Bidding Strategies (Section 4).** Designing an algorithm to learn the optimal safe strategy with at most $m$ bid-quantity pairs poses significant challenges due to the exponential size of the decision space in $m$. So, we first consider an offline setting where the competing bids are known in advance. In this setting, we construct a polynomial-sized directed acyclic graph (DAG) with carefully assigned edge weights and show that determining the maximum weight path in the DAG is equivalent to computing the optimal safe strategy with at most $m$ bid-quantity pairs (Theorem 4.2).

Building on this, we develop an online learning algorithm when the competing bids are unknown.[1] Our algorithm runs in polynomial time and achieves a regret of $\widetilde{O}(M\sqrt{mT})$ in a full-information setting and $\widetilde{O}(m^{3/2}M^2\sqrt{T})$ regret in a bandit setting, where the clairvoyant selects the fixed hindsight optimal safe strategy (Theorem 4.3). Additionally, we establish a regret lower bound of $O(M\sqrt{T})$ (Theorem 4.4).

The problem of learning to bid in multi-unit uniform price and pay-as-bid auctions has been explored recently by Brânzei et al. (2023); Galgana & Golrezaei (2024); Potfer et al. (2024), assuming the bidders are quasilinear utility maximizers. Our work differs from them in two key ways: (a) we consider value-maximizing buyers with RoI constraints, a fundamentally different behavioral model, and (b) from a technical perspective, unlike prior works that require bid spaces to be discretized, our approach does not, due to the structure of safe strategies. Consequently, the time complexity becomes independent of $T$ in both online and offline settings [2], and under bandit feedback, the depen-

dence on $T$ in the regret bound improves from $T^{2/3}$ to $\sqrt{T}$ implying that it is easier to learn safe strategies for value maximizers compared to bidding strategies for quasilinear utility maximizers.

**Richer Classes of Strategies for the Clairvoyant (Section 5).** A key novel aspect of our work is to evaluate the performance of safe bidding strategies with at most $m$ bid-quantity pairs and the online learning algorithm from Section 4 against a clairvoyant that selects the optimal strategy from a richer class of strategies. In this case, the bidder (learner) achieves $\alpha$-approximate sublinear regret (same as Theorem 4.3) for a parameter $\alpha \in (0, 1]$—defined as the *richness ratio*—thereby demonstrating the robustness of the safe bidding strategies against a broad range of benchmarks (Corollary 5.2). One of our main contributions is to compute the richness ratio, $\alpha$, for various classes of bidding strategies. Specifically, when the clairvoyant selects the optimal bidding from the class of

(a) strategies that are RoI-feasible, not necessarily safe and have at most $m$ bid-quantity pairs, we get $\alpha = \frac{1}{2}$, notably *independent of* $m$ (Theorem 5.3).

(b) safe strategies with at most $m'$ bid-quantity pairs, where $m' \ge m$, we obtain $\alpha = \frac{m}{m'}$ (Theorem 5.4).

(c) strategies that are RoI-feasible, not necessarily safe and have at most $m'$ bid-quantity pairs, where $m' \ge m$, we get that $\alpha = \frac{m}{2m'}$ (Theorem 5.5).

Computing $\alpha$ involves two key challenges: (a) deriving an upper bound on the ratio between the value obtained by the clairvoyant's optimal strategy and that of the optimal safe strategy in the worst case (see Definition 4 for details), and (b) proving that this upper bound is tight. The key ideas for deriving the upper bound are outlined in Section 5. The construction of problem instances, each tailored to a specific strategy class, is non-trivial due to their exponential size in $m$ and the need for a careful choice of competing bids and the valuation vector, as detailed in Appendices D.1 to D.4. Simulations using semi-synthetic data show that the empirical richness ratios in practical scenarios are significantly better than the theoretical bounds, making safe strategies highly appealing in practice (Section 5.3).

### 1.2. Related Work

**Value Maximizers and RoI Constraints.** The concept of agents as value maximizers within financial constraints is a well-established notion in microeconomic theory (Mas-Colell et al., 1995). In mechanism design literature, one of the earliest explorations of value-maximizing agents was conducted by Wilkens et al. (2016). Their work primar-

---

[1] For a detailed discussion on leveraging offline algorithms for no-regret learning, we refer readers to Roughgarden & Wang (2019); Niazadeh et al. (2022); Brânzei et al. (2023).

[2] Assuming the discretization level in the offline setting is same

as that in the online setting. In the online setting, time complexity refers to per-round running time.

ily delved into the single-parameter setting, characterizing truthful auctions for value maximizers. Similarly, Fadaei & Bichler (2016) and Lu et al. (2023) studied truthful (approximate) revenue-maximizing mechanisms in combinatorial markets tailored for such agents.

In recent years, there has been growing interest in RoI-constrained value maximizers, particularly in the context of autobidding and online advertising. One of the earliest works in this area is Golrezaei et al. (2021), which studied auction design for RoI-constrained buyers and validated the presence of such soft financial constraints using data from online ad auctions. Broadly, the literature in this space can be divided into two categories: (i) works that design optimal auctions under RoI constraints (Balseiro et al., 2021a;b; 2022; Deng et al., 2021; 2023b), and (ii) works that characterize optimal bidding strategies and/or develop learning algorithms in repeated auction settings (Aggarwal et al., 2019; Deng et al., 2023a; Golrezaei et al., 2023; Castiglioni et al., 2024; Aggarwal et al., 2025; Lucier et al., 2024). Our work primarily aligns with the latter. We study value-maximizing buyers in uniform price auctions under RoI constraints and demonstrate that these buyers can employ safe bidding strategies that are efficiently learnable and robust against a variety of strong benchmarks.

**Multi-unit Auctions.** In this work, we focus on a subset of combinatorial auctions termed as multi-unit auctions in which multiple identical goods are sold to a group of buyers.[3] These auctions find widespread application in various practical scenarios, including Treasury auctions (Hortaçsu & McAdams, 2010), procurement auctions (Cramton & Ausubel, 2006), electricity markets (Tierney et al., 2008), and emissions permit auctions (Schmalensee & Stavins, 2017). While several works have focused on studying the equilbria properties in these auctions (Ausubel et al., 2014; Markakis & Telelis, 2015; De Keijzer et al., 2013), computing equilibrium strategies is usually intractable in general multi-unit auctions due to multi-dimensional valuations. As a result, a growing body of work has emphasized on designing optimal bidding strategies in a prior-free setting, i.e., without any assumptions of the competing bidders' bids or valuations (Galgana & Golrezaei, 2024; Potfer et al., 2024; Brânzei et al., 2023). Our work contributes to this literature by learning optimal strategies for value-maximizing buyers in repeated uniform price auctions.

## 2. Model

**Preliminaries.** There are $n$ buyers (bidders) indexed by $i \in [n]$, and $K$ identical units of a single good. Each bidder $i$ has a fixed, private valuation curve, denoted by $\mathbf{v}_i \in \mathbb{R}_+^K$

---

[3]For a comprehensive survey on combinatorial auctions, we refer the readers to several excellent works by De Vries & Vohra (2003); Blumrosen & Nisan (2007); Palacios-Huerta et al. (2022).

that has diminishing marginal returns property, i.e., $v_{i,1} \geq v_{i,2} \geq \cdots \geq v_{i,K}$ which is standard in literature (Brânzei et al., 2023; Goldner et al., 2020). As we study optimal bidding strategies from the perspective of a single bidder, we drop the index $i$ when the context is clear. The maximum total demand for bidder $i$, denoted by $M \in [K]$, is defined as $\min\{j \in [K-1] : v_{j+1} = 0\}$. If such an index does not exist, we set $M$ as $K$. Hence, without loss of generality, we assume $\mathbf{v} \in \mathbb{R}_+^M$. For each $\mathbf{v} = [v_1, \ldots, v_M]$, we define the *average cumulative valuation* vector as $\mathbf{w} = [w_1, \ldots, w_M]$,

$$\text{where} \quad w_j = \frac{1}{j} \sum_{\ell \leq j} v_\ell, \forall j \in [M]. \tag{1}$$

As $v_1 \geq \cdots \geq v_M$, we also have $w_1 \geq \cdots \geq w_M$.

### 2.1. Auction Format and Bidders' Behavior

**Allocation and Payment Rule.** In a uniform price auction, each bidder $i$ submits a sorted bid vector $\mathbf{b}$ using $m$-uniform bidding language (see Definition 1). The auctioneer collects the bids (entries of the bid vector) from all the bidders, sorts them in non-increasing order, and allocates units to the bidders with the top $K$ bids (also termed as 'winning' bids). That is, if bidder $i$ has $j$ bids in the top $K$ positions, they are allocated $j$ units. For ease of exposition, we assume there are no ties (or ties are always broken in the favor of the bidder in consideration). See Appendix F for our discussion on handling ties. We assume the auction follows the last-accepted-bid payment rule (bidders pay the $K^{th}$ highest bid per unit) which is widely used for uniform price auctions in practice (Regulations, 2019; Garbade & Ingber, 2005).

Let $\boldsymbol{\beta}_-$ denote the bids submitted by all bidders except bidder $i$ and $\boldsymbol{\beta}_-^{(j)}$ be the $j^{th}$ smallest among the top $K$ competing bids (i.e., bids from all bidders except bidder $i$). Suppose bidder $i$ submits $\mathbf{b}$, such that the bid profile is $\boldsymbol{\beta} := (\mathbf{b}; \boldsymbol{\beta}_-)$. Let $x(\boldsymbol{\beta})$ and $p(\boldsymbol{\beta})$ denote the number of units allocated to bidder $i$ and the clearing price (i.e., the per-unit price which is the $K^{th}$ highest submitted bid), respectively. The total value obtained by the bidder is $V(\boldsymbol{\beta}) = \sum_{j \leq x(\boldsymbol{\beta})} v_j$, while the total payment made is $P(\boldsymbol{\beta}) = p(\boldsymbol{\beta}) \cdot x(\boldsymbol{\beta})$.

**Bidding Language.** Multi-unit auctions allocate a large number of identical units, requiring efficient ways for the bidders to express preferences. A common approach is *standard bidding*, where bidders submit a vector of bids, one for each unit (Brânzei et al., 2023; Galgana & Golrezaei, 2024; Babaioff et al., 2023; Birmpas et al., 2019; Potfer et al., 2024). Although expressive, this becomes computationally impractical when the number of units, $K$, is large, as in EU ETS emission permit auctions and Treasury auctions. To address this, we consider a bidding language called $m$-*uniform bidding*, for any $m \in \mathbb{N}$.[4] In $m$-uniform bidding,

---

[4]This format generalizes the *uniform bidding* format (De Kei-

bidders submit $m$ bid-quantity pairs $(b_i, q_i)$, where $b_i$ is the bid value per unit and $q_i$ is the quantity demanded:

**Definition 1** ($m$-Uniform Bidding). For a fixed $m \in \mathbb{N}$, a $m$-uniform bidding strategy is characterized by $m$ bid-quantity pairs, denoted as $\mathbf{b} := \langle (b_1, q_1), \ldots, (b_m, q_m) \rangle$, where $b_1 > b_2 > \cdots > b_m > 0$ and $q_j > 0$, $j \in [m]$. This $m$-uniform bid can be equivalently expressed as a vector (similar to the standard bidding format) in which the first $q_1$ bids are $b_1$, followed by $q_2$ bids of $b_2$, and so on.

We define $\mathbf{b}[1 : \ell] = \langle (b_1, q_1), \ldots, (b_\ell, q_\ell) \rangle$, for all $\ell < m$, to represent the first $\ell$ bid-quantity pairs within a $m$-uniform bidding strategy $\mathbf{b} = \langle (b_1, q_1), \ldots, (b_m, q_m) \rangle$. We further define $Q_j = \sum_{\ell=1}^{j} q_\ell$ for all $j \in [m]$ as the total quantity demanded in the first $j$ bid-quantity pairs, with $Q_0 = 0$, where we assume, without loss of generality that $Q_m \leq M$.[5] If $m = M$, the bidding format is equivalent to standard bidding but in practice, bidders often submit only a few bid-quantity pairs. For instance, in the 2023 EU ETS auctions, bidders submitted $\sim 4.35$ bid-quantity pairs per auction on average (EEX, 2023).

**Bidders' Behavior.** The bidders maximize their total value obtained while adhering to a constraint that ensures the total value obtained in an auction is at least a constant multiple of the payment in that auction. This can be equivalently expressed as a return-on-investment (RoI) constraint:

$$V(\mathbf{b}; \boldsymbol{\beta}_-) \geq (1 + \gamma) P(\mathbf{b}; \boldsymbol{\beta}_-). \tag{2}$$

Here, $\gamma$ is defined as the *target RoI* which is private and fixed. Without loss of generality, we assume $\gamma = 0$ (or equivalently the valuation curve $\mathbf{v}$ is scaled by $\frac{1}{1+\gamma}$) for the rest of this work. For $\gamma = 0$, the RoI constraint implies the value obtained in an auction is at least the payment. We illustrate how the auction operates in Example 1.

**Example 1.** *Consider an auction with $n = 2$ bidders, and $K = 5$ identical units. The valuations are: $\mathbf{v}_1 = [6, 4, 3, 1, 1]$ and $\mathbf{v}_2 = [5, 3, 1, 1, 0]$. Both the bidders are value maximizing buyers with target RoI, $\gamma_1 = \gamma_2 = 0$. Suppose $m = 2$ and the bids submitted by the bidders are $\mathbf{b}_1 = \langle (5, 2), (3, 3) \rangle$ and $\mathbf{b}_2 = \langle (4, 2), (2, 2) \rangle$. The bid profile: $\boldsymbol{\beta} = [5, 5, 4, 4, 3, 3, 3, 2, 2]$ and top $K = 5$ winning bids are $[\underline{5}, \underline{5}, 4, 4, \underline{3}]$. Bidder 1 is allocated 3 units as they have 3 bids (underlined) in the winning bids, and bidder 2 gets the remaining 2 units. The clearing price $p(\boldsymbol{\beta}) = 3$, $V_1(\boldsymbol{\beta}) = 6 + 4 + 3 = 13$, $V_2(\boldsymbol{\beta}) = 5 + 3 = 8$, $P_1(\boldsymbol{\beta}) = 3 \cdot 3 = 9$, and $P_2(\boldsymbol{\beta}) = 2 \cdot 3 = 6$. The RoI constraint is satisfied for both the bidders as $13 > 9$ and $8 > 6$.*

---

jzer et al., 2013; Birmpas et al., 2019) and aligns with practical languages like those in product-mix auctions (Klemperer, 2009) and piecewise-linear bidding (Nisan, 2015).

[5] Suppose the bidder bids for, and wins more than $M$ units. There is no additional value being allocated over $M$ units, but the total payment increases (assuming the clearing price is positive), potentially violating the RoI constraint.

## 2.2. Learning to Bid in Repeated Settings

In practice, most multi-unit auctions, such as emission permit auctions and Treasury auctions, are conducted in a repeated setting. Formally, the auction described in the previous section takes place sequentially over $T$ rounds indexed by $t \in [T]$. We assume that the valuation vector $\mathbf{v}$ is fixed over the $T$ rounds which is a standard assumption in the literature (Brânzei et al., 2023; Galgana & Golrezaei, 2024; Potfer et al., 2024). Similarly, the target RoI is also assumed to be fixed over the $T$ rounds.

We now extend the notations from the previous section to the repeated setting. Formally, in this setting, $\boldsymbol{\beta}_-^t$ denotes the bids submitted by all bidders except bidder $i$ in round $t$ and $\boldsymbol{\beta}_{-,t}^{-(j)}$ is the $j^{th}$ smallest among the top $K$ competing bids in round $t$. In round $t$, if bidder $i$ submits a bid $\mathbf{b}^t$, the bid profile is $\boldsymbol{\beta}^t := (\mathbf{b}^t; \boldsymbol{\beta}_-^t)$. Let $x(\boldsymbol{\beta}^t)$ and $p(\boldsymbol{\beta}^t)$ denote the number of units allocated to bidder $i$ and the clearing price, respectively, in round $t$. The total value obtained by the bidder is $V(\boldsymbol{\beta}^t) = \sum_{j \leq x(\boldsymbol{\beta}^t)} v_j$, while the total payment made is $P(\boldsymbol{\beta}^t) = p(\boldsymbol{\beta}^t) \cdot x(\boldsymbol{\beta}^t)$.

**RoI Constraints in Repeated Setting.** In the repeated setting, we require the bidders to satisfy the RoI constraint described in Eq. (2) in every round. A bidding strategy $\mathbf{b}$ is called *feasible* for a sequence of competing bids $[\boldsymbol{\beta}_-^t]_{t \in [T]}$, if the RoI constraint is satisfied for every round $t \in [T]$.

*Remark* 2.1 (RoI Constraints). Our notion of RoI constraints in the repeated setting aligns with the definitions in Wilkens et al. (2016; 2017); Lv et al. (2023). Similar constraints are considered by Lucier et al. (2024) and Gaitonde et al. (2023), termed as *marginal RoI (or value)* and *maximum bid* constraints (up to scaling factors), respectively. Unlike the aggregate constraints over $T$ rounds typically assumed in online ad auction literature (Deng et al., 2021; 2024; Feng et al., 2023; Deng et al., 2023a), we enforce RoI constraints for each auction individually. This distinction reflects the fundamental differences between the two settings: ad auctions often occur simultaneously and frequently, with values in the order of cents, whereas Treasury and emission permit auctions occur sequentially over longer horizons, with units valued in millions. Hence, bidders in such auctions prioritize profitability in each auction rather than waiting for an indefinite period, leading to per-round RoI constraints.[6]

## 2.3. Objective and Performance Metric

We consider an online setting where a bidder seeks to maximize cumulative value while satisfying RoI constraints

---

[6] Although EU ETS emission permit auctions are scheduled to occur regularly, regulations stipulate that an auction may be canceled if the bidders' demand falls short of the supply of permits or if the clearing price of the auction does not meet the reserve price (Regulations, 2019). Hence, the bidders are more likely to ensure RoI feasibility in each round.

and managing uncertainty about competing bids. In this setting, bidders privately submit their bids, and the auctioneer allocates items and sets prices based on these bids. In round $t$, the learning algorithm maps the feedback information set from the first $t-1$ rounds, denoted by $\mathcal{I}^{t-1}$, to a bid $\mathbf{b}^t$, where this mapping may be deterministic or random. Under full information feedback, the information set $\mathcal{I}^{t-1} = (\boldsymbol{\beta}^1, \boldsymbol{\beta}^2, \ldots, \boldsymbol{\beta}^{t-1})$, which includes all bids from previous rounds. In the bandit setting, the information set $\mathcal{I}^{t-1} = (V(\boldsymbol{\beta}^1), P(\boldsymbol{\beta}^1), \ldots, V(\boldsymbol{\beta}^{t-1}), P(\boldsymbol{\beta}^{t-1}))$ includes only the value and price outcomes of the previous rounds.

We design learning algorithms to minimize *regret*, the difference in the value obtained by a fixed hindsight-optimal strategy chosen by a clairvoyant with *a priori* knowledge of competing bids and that by the learner over time. Formally,

$$\mathsf{REG} = \max_{\mathbf{b} \in \mathcal{B}} \sum_{t=1}^{T} V(\mathbf{b}; \boldsymbol{\beta}_-^t) - \sum_{t=1}^{T} \mathbb{E}[V(\mathbf{b}^t; \boldsymbol{\beta}_-^t)], \quad (3)$$

where the expectation is with respect to any randomness in the learning algorithm. Here, $\mathcal{B}$ is the class of bidding strategies (formally characterized in Section 3). We require the bidding strategies in $\mathcal{B}$ to be RoI feasible for the sequence of competing bids $[\boldsymbol{\beta}_-^t]_{t \in [T]}$. In Section 4, we consider the case when the both the clairvoyant and the learner choose strategies from the same class and aim to obtain sublinear regret. Later, we consider more challenging settings where the clairvoyant can choose the optimal strategy from much richer classes of strategies compared to the learner. In these cases, we obtain sublinear $\alpha$-*approximate* regret, where $\alpha \in (0, 1]$ is the richness ratio (see details in Section 5).

## 3. Safe Bidding Strategies

Recall that strategies chosen by the learner must be causal (mapping the history to a bidding strategy) while remaining RoI feasible for all rounds $t \in [T]$, even under adversarially generated competing bids. This creates an obvious challenge: a bidding strategy that satisfies RoI feasibility in previous rounds may become infeasible even under the slightest change in the competing bids e.g., if $\mathbf{v} = [0.9, 0.5, 0.1]$, the bidding strategy $(0.6, 3)$ is RoI feasible for competing bids $\boldsymbol{\beta}_-^1 = [0.61, 0.59, 0.59]$ but not for $\boldsymbol{\beta}_-^2 = [0.59, 0.59, 0.59]$. To address this, we focus on *safe bidding strategies* that inherently guarantee RoI feasibility, regardless of the adversarial behavior of competing bids:

**Definition 2** (Safe Strategies). A $m$-uniform bidding strategy, $\mathbf{b} = \langle (b_1, q_1), \ldots, (b_m, q_m) \rangle$, is called a *safe* strategy if it is feasible irrespective of the competing bids, i.e.,

$$V(\mathbf{b}; \boldsymbol{\beta}_-) \geq P(\mathbf{b}; \boldsymbol{\beta}_-), \quad \forall \boldsymbol{\beta}_- .$$

The class of all $m$-uniform safe bidding strategies is denoted as $\mathcal{S}_m$. The union of classes of safe bidding strategies with at most $m$ bid-quantity pairs is denoted as $\mathcal{U}_m = \bigcup_{k \in [m]} \mathcal{S}_k$.

### 3.1. Characterizing Safe Bidding Strategies

We begin by defining underbidding (overbidding) under the $m$-uniform bidding format in the given context. Recall that in a single-item auction, if the value of the item is $v$, then a bid $b$ is an underbid (overbid) if $b < v$ ($b > v$).

**Definition 3** (Underbid and Overbid). A $m$-uniform bidding strategy $\mathbf{b} = \langle (b_1, q_1), \ldots, (b_m, q_m) \rangle$ is an

(a) *underbidding* strategy if $b_j \leq w_{Q_j}, \forall j \in [m]$ and $\exists \ell \in [m]$ such that $b_\ell < w_{Q_\ell}$.

(b) *overbidding* strategy if $\exists \ell \in [m]$ such that $b_\ell > w_{Q_\ell}$.

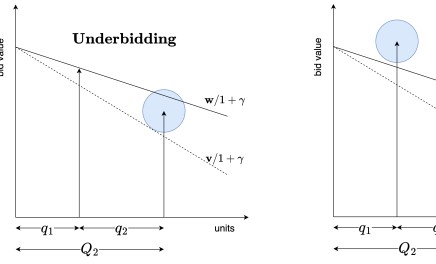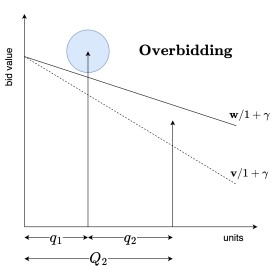

*Figure 2.* The solid line represents the average cumulative valuation curve, $\mathbf{w}$, and the dotted line represents the valuation curve, $\mathbf{v}$. The figure in the left (resp. right) illustrates underbidding (resp. overbidding) for a 2-uniform bidding strategy. Note that the notions of underbidding and overbidding in Definition 3 are defined with respect to $\mathbf{w}$ and *not* $\mathbf{v}$.[8] Here, the plots of $\mathbf{v}$ and $\mathbf{w}$ are shown to be linear for illustrative purposes only.

In other words, if $\gamma = 0$, $\mathbf{b} = \langle (b_1, q_1), \ldots, (b_m, q_m) \rangle$ is an underbidding strategy if, for all $j \in [m]$, the bid $b_j$ is at most the average of the first $Q_j$ entries of the valuation vector, denoted as $w_{Q_j}$, and there exists $\ell \in [m]$ where the inequality is strict. Recall that $Q_j = \sum_{\ell \leq j} q_\ell$ denotes the maximum number of demanded units in the first $j$ bid-quantity pairs. Similarly, $\mathbf{b}$ is an overbidding strategy if there exists some $\ell \in [m]$ such that $b_\ell$ is strictly greater than the average of the first $Q_\ell$ entries of the valuation vector. Having defined the notions of overbidding and underbidding, we characterize the class of safe bidding strategies, $\mathcal{S}_m$:

**Theorem 3.1.** *For any $m \in \mathbb{N}$, no overbidding is allowed in $\mathcal{S}_m$. So, the collection of all $m$-uniform safe strategies is*

$$\mathcal{S}_m = \left\{ \mathbf{b} = \langle (b_1, q_1), \ldots, (b_m, q_m) \rangle : b_\ell \leq w_{Q_\ell}, \forall \ell \in [m] \right\},$$

*where $\mathbf{w}$ is defined in Eq. (1).*

The proofs of this section are presented in Appendix A. Since there are infinitely many safe strategies in $\mathcal{S}_m$, we

---

[8]In Definition 3 (and throughout the paper), we assume $\gamma = 0$ without loss of generality for ease of exposition. Fig. 2 illustrates how different values of $\gamma$ impacts the results. Specifically, for any $\gamma \geq 0$, underbidding and overbidding are defined with respect to the scaled average cumulative valuation curve, $\frac{\mathbf{w}}{1+\gamma}$.

focus on the subset of safe *undominated* strategies, $\mathcal{S}^\star_m$, obtained by removing *very weakly dominated* strategies. A safe strategy $\mathbf{b}$ is said to be very weakly dominated if there exists another safe strategy $\mathbf{b}'$ such that, for any competing bid $\boldsymbol{\beta}_-$, $V(\mathbf{b}, \boldsymbol{\beta}_-) \leq V(\mathbf{b}', \boldsymbol{\beta}_-)$ (Shoham & Leyton-Brown, 2008, pp. 79). Keeping these strategies does not improve the bidder's performance, but removing them yields a *finite* strategy class (as shown below), which significantly simplifies the design of online learning algorithms. We now present the main result of this section:

**Theorem 3.2.** *The class of $m$-uniform safe undominated bidding strategies, $\mathcal{S}^\star_m$, is given as follows:*

$$\mathcal{S}^\star_m = \Big\{ \mathbf{b} = \langle (b_1, q_1), \ldots, (b_m, q_m) \rangle : b_\ell = w_{Q_\ell}, \ell \in [m] \Big\}.$$

*The union of classes of safe undominated strategies with at most $m$ bid-quantity pairs is denoted by $\mathcal{U}^\star_m = \bigcup_{k \in [m]} \mathcal{S}^\star_k$.*

Observe that $\mathcal{S}^\star_m$ is a *finite* class of bidding strategies that depend only the valuation curve and is *independent* of the competing bids. The strategies in $\mathcal{S}^\star_m$ exhibit a "nested" structure, referred to as nested $m$-uniform bidding strategies hereafter, where the $j^{th}$ highest bid ($b_j$) is the average of the first $Q_j = \sum_{\ell=1}^j q_\ell$ entries of the valuation curve (illustrated in Fig. 1). Importantly, fixing $Q_j$'s uniquely determines the bidding strategy, as $b_j = w_{Q_j}$ for strategies in $\mathcal{S}^\star_m$ which will be crucial in learning the optimal safe bidding strategy, as discussed in the following section.

# 4. Learning Safe Bidding Strategies

Here, the clairvoyant and the learner choose strategies from the class of safe bidding strategies with at most $m$ bid-quantity pairs, $\mathcal{U}^\star_m$. Thus, the regret defined in Eq. (3) is

$$\mathsf{REG} = \max_{\mathbf{b} \in \mathcal{U}^\star_m} \sum_{t=1}^T V(\mathbf{b}; \boldsymbol{\beta}^t_-) - \sum_{t=1}^T \mathbb{E}[V(\mathbf{b}^t; \boldsymbol{\beta}^t_-)].$$

To design learning algorithms, we first consider the offline setting, where we aim to solve the following problem given the bid history, $[\boldsymbol{\beta}^t_-]_{t \in [T]}$.

$$\max_{\mathbf{b} \in \mathcal{U}^\star_m} \sum_{t=1}^T V(\mathbf{b}; \boldsymbol{\beta}^t_-) \qquad \text{(OFFLINE)}$$

Since $\mathcal{U}^\star_m$ contains $O(M^m)$ strategies, evaluating each strategy individually is intractable. To overcome this, we reduce the offline problem to finding the maximum weight path in a carefully constructed edge-weighted directed acyclic graph (DAG), which allows for the computation of the optimal offline bidding strategy in poly($m, M$) time.

## 4.1. Offline Problem

Fix a bid history, $\mathcal{H}_- = [\boldsymbol{\beta}^t_-]_{t \in [T]}$, and consider (OFFLINE). The following lemma shows that the objective function can

be decomposed across the bid-quantity pairs. Formally,

**Lemma 4.1.** *Problem* (OFFLINE) *can be formulated as*

$$\max_{\ell \in [m]} \max_{Q_1, \ldots, Q_\ell} \sum_{j=1}^\ell \sum_{t=1}^T \sum_{k=Q_{j-1}+1}^{Q_j} v_k \cdot \mathbb{I}\Big[w_{Q_j} \geq \boldsymbol{\beta}^{-(k)}_{-,t}\Big].$$

The proofs of this section are presented in Appendix B. Building on the decomposition in Lemma 4.1, we now construct an edge-weighted DAG $\mathcal{G}(V, E)$ as proposed earlier.

**Vertices.** The DAG has a 'layered' structure with a source node ($s$), destination node ($d$) and $m$ intermediate layers. The $\ell^{th}$ intermediate layer contains $M + 1 - \ell$ nodes, denoted by $(\ell, j)$ where $\ell \in [m], j \in \{\ell, \ell+1, \ldots, M\}$ as shown in Fig. 3. For convenience, we set $s = (0, 0)$.

**Edges and Edge Weights.** A directed edge exists from vertex $x$ to $y$ under the following conditions:

(i) $x = (\ell - 1, j)$ and $y = (\ell, j')$, $\forall \ell \in [m]$ and $j < j'$. This edge connects consecutive layers in the graph, and its weight is given by:

$$\mathsf{w}(e) = \sum_{t=1}^T \sum_{k=j+1}^{j'} v_k \cdot \mathbb{I}\Big[w_{j'} \geq \boldsymbol{\beta}^{-(k)}_{-,t}\Big]. \qquad (4)$$

(ii) $x = (\ell, j)$ and $y = d$, $\forall \ell \in [m], j \in [M]$. This edge connects the current layer to the destination node $d$, and its weight is $\mathsf{w}(e) = 0$.

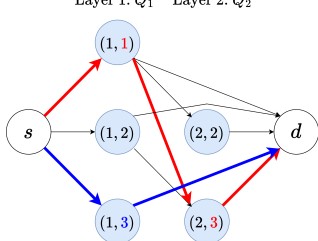

*Figure 3.* In this DAG, $M = 3, m = 2$. The red path refers to $\mathbf{b} = \langle (w_1, 1), (w_3, 2) \rangle \in \mathcal{U}^\star_2$. The values in red are the corresponding $Q_j$'s. Similarly, the blue path refers to $\mathbf{b} = (w_3, 3) \in \mathcal{U}^\star_2$.

**Theorem 4.2.** • *There exists a bijective mapping between the $s$-$d$ paths in $\mathcal{G}(V, E)$ and bidding strategies in $\mathcal{U}^\star_m$, i.e., the path $s \to (1, z_1) \to \cdots \to (k, z_k) \to d$ for $k \in [m]$ refers to the strategy $\mathbf{b} = \langle (b_1, q_1), \ldots, (b_k, q_k) \rangle$ where*

$$b_\ell = w_{z_\ell} \quad \text{and} \quad q_\ell = z_\ell - z_{\ell-1}, \forall \ell \in [k].$$

*where $w_{z_\ell}$ is defined as per Eq. (1), $\forall \ell \in [k]$. Conversely, the strategy $\mathbf{b} = \langle (w_{Q_1}, q_1), \ldots, (w_{Q_k}, q_k) \rangle$ maps to the path $s \to (1, Q_1) \to \cdots \to (k, Q_k) \to d$.*

• *The value obtained by a bidding strategy is the weight of the corresponding $s$-$d$ path. Thus, the Problem (OFFLINE) is equivalent to finding the maximum weight $s$-$d$ path in $\mathcal{G}(V, E)$ which can be computed in poly($m, M$) time.*

## 4.2. Online Setting

In the online setting, in each round $t \in [T]$, the learner submits a strategy $\mathbf{b}^t \in \mathcal{U}_m^\star$ and receives feedback from the auction. We consider two feedback models: (a) full information setting and (b) bandit setting, with the latter discussed in Appendix B.3.1. Leveraging the DAG formulation, we design an online learning algorithm such that $\mathsf{REG} = o(T)$. Without loss of generality, assume $v_j \in [0, 1], \forall j \in [M]$.

In every round $t$, a DAG $\mathcal{G}^t(V, E)$ is constructed as described earlier. After receiving feedback at the end of the round, in the full information setting,

(i) If $x = (\ell - 1, j)$ and $y = (\ell, j')$ with $\ell \in [m]$ and $j < j'$,

$$\mathsf{w}^t(e) = \sum_{k=j+1}^{j'} v_k \cdot \mathbb{I}\left[w_{j'} \geq \boldsymbol{\beta}_{-,t}^{-(k)}\right]. \tag{5}$$

(ii) If $x = (\ell, j)$ and $y = d$, then $\mathsf{w}^t(e) = 0, \forall \ell \in [m], j \in [M]$.

The online algorithm in Algorithm 1 consists of three main steps: (1) UPDATE, (2) SAMPLE, and (3) MAP. In the first two steps, the algorithm maintains and updates the probabilities over the edges in the constructed DAG and selects a $s$-$d$ path in the DAG by sampling from these probabilities. These steps implement exponential weight updates using the weight-pushing method proposed by Takimoto & Warmuth (2003). Naively implementing exponential weight updates by treating each $s$-$d$ path (safe strategy) as an expert is intractable as there are $O(M^m)$ such experts. However, following the decomposition in Lemma 4.1, Algorithm 1 only needs to maintain weights for individual edges and then sample a path based on these weights, avoiding the need to track weights for each path separately. The MAP step is the key component of the algorithm that maps the sampled path to a safe strategy (by Theorem 4.2).

---

**Algorithm 1** Learning Safe Bidding Strategies (Full Info.)

---

**Require:** valuation curve $\mathbf{v}$, time horizon $T$, learning rate $\eta > 0$.
1: **for** $t = 1, 2, \ldots, T$ **do**
2:     Construct $\mathcal{G}^t(V, E)$ similar to $\mathcal{G}(V, E)$ without weights.
3:     UPDATE : Obtain edge probabilities $\varphi^t(\cdot)$ by Algorithm 2. SAMPLE: Define initial node $u = s$ and path $\mathfrak{p}^t = s$.
4:     **while** $u \neq d$ **do**
5:         Sample $v$ with probability $\varphi^t(u \to v)$.
6:         Append $v$ to the path $\mathfrak{p}^t$; set $u \leftarrow v$.
7:     **end while**
8:     MAP: If $\mathfrak{p}^t = s \to (1, z_1) \to \cdots \to (k, z_k) \to d$ for some $k \in [m]$, submit $\mathbf{b}^t = \langle (b_1, q_1), \ldots, (b_k, q_k) \rangle$ where

$$b_\ell = w_{z_\ell} \quad \text{and} \quad q_\ell = z_\ell - z_{\ell-1}, \forall \ell \in [k].$$

    where $w_{z_\ell}$ is defined as per Eq. (1) for all $\ell \in [k]$.
9:     The bidder observes $\boldsymbol{\beta}_-^t$ and sets $\mathsf{w}^t(e)$ as per Eq. (5).
10: **end for**

---

**Theorem 4.3.** *Algorithm 1 runs in* $\mathrm{poly}(m, M)$ *time per round and achieves* $\mathsf{REG} \leq O(M\sqrt{mT \log M})$ *under full information feedback, and* $\mathsf{REG} \leq O(M^2\sqrt{m^3 T \log M})$ *in the bandit setting.*

---

**Algorithm 2** Update Edge Probabilities

---

**Require:** For $t \geq 1$, $\mathcal{G}^t(V, E)$. For $t \geq 2$, additionally require $\eta$, $\varphi^{t-1}(e)$ and $\mathsf{w}^{t-1}(e)$, $\forall e \in E$.
1: If $t = 1$,
    – Initialize the edge probability of each edge $e = x \to y$ as $\varphi^1(e) = 1/|\mathcal{N}_x|$ where $\mathcal{N}_x$ is the set of out-neighbors of $x$.
2: If $t \geq 2$:
    – Set $\Gamma^{t-1}(d) = 1$ and recursively compute in bottom-to-top fashion for every node $u$ in $\mathcal{G}^t(V, E)$:

$$\Gamma^{t-1}(u) = \sum_{v:u \to v=e \ni E} \Gamma^{t-1}(v) \cdot \varphi^{t-1}(e) \cdot \exp(\eta \mathsf{w}^{t-1}(e))$$

    – For edge $e = u \to v$ in $\mathcal{G}^t(V, E)$, update edge probability: $\varphi^t(e) = \varphi^{t-1}(e) \cdot \exp(\eta \mathsf{w}^{t-1}(e)) \cdot \frac{\Gamma^{t-1}(v)}{\Gamma^{t-1}(u)}$.

---

We prove Theorem 4.3 by showing that Algorithm 1 is an efficient (polynomial time) and equivalent implementation of the exponential weight updates algorithm. We complement Theorem 4.3 by establishing the following lower bound.

**Theorem 4.4.** *Suppose $M \geq 2$ and $m = 1$. Then, there exist competing bids, $[\boldsymbol{\beta}_-^t]_{t \in [T]}$, such that, under any learning algorithm, the expected regret $\mathbb{E}[\mathsf{REG}] = \Omega(M\sqrt{T})$ in both full information and bandit settings.*

## 5. Competing Against Richer Classes of Bidding Strategies

In previous sections, we characterized the class of safe bidding strategies and proposed a learning algorithm that obtain sublinear regret, where the regret is computed against a clairvoyant that also selects the optimal safe bidding strategy. In this section, we consider the cases where the clairvoyant can choose the optimal strategy from richer bidding classes, which we will describe shortly. We show that under Algorithm 1, the class of safe bidding strategies achieves sublinear (approximate) regret when competing against these stronger benchmarks. Here, the performance metric is

$$\alpha\text{-}\mathsf{REG}_{\mathcal{B}_c} = \alpha \cdot \max_{\mathbf{b} \in \mathcal{B}_c} \sum_{t=1}^{T} V(\mathbf{b}; \boldsymbol{\beta}_-^t) - \sum_{t=1}^{T} \mathbb{E}[V(\mathbf{b}^t; \boldsymbol{\beta}_-^t)].$$

Here, $\mathcal{B}_c$ is the class of bidding strategies from which the clairvoyant (which serves as our benchmark) chooses the optimal bidding strategy whereas the bidder (learner) submits bids, $\mathbf{b}^t \in \mathcal{U}_m^\star$, in each round unless stated otherwise and $\alpha \in (0, 1]$ is defined as the *richness ratio*.

*Remark* 5.1 (Richness Ratio $\alpha$). In prior works on approximate regret, $\alpha$ typically measures the hardness of the offline

problem, such as $\max_{\mathbf{b}\in\mathcal{B}_c}\sum_{t=1}^{T}V(\mathbf{b};\boldsymbol{\beta}_-^t)$ (see Streeter & Golovin (2008); Niazadeh et al. (2022)). In contrast, our work reinterprets $\alpha$ as *richness ratio*, capturing the relative richness of the clairvoyant's strategy class compared to the learner's. Specifically, it quantifies how closely the value achieved by the optimal strategy in $\mathcal{U}_m^\star$ approximates that by the optimal strategy in $\mathcal{B}_c$. If $\mathcal{B}_c = \mathcal{U}_m^\star$, then $\alpha = 1$ (see Section 4). By considering a richer class of strategies for the clairvoyant, we aim to quantify the robustness and performance of safe strategies against stronger benchmarks.

Our main contribution in this section is to compute the richness ratio $\alpha$ for different classes of bidding strategies. To this end, for any $\mathcal{B}_c$, define:

$$\Lambda_{\mathcal{B}_c,\mathcal{U}_m^\star} := \max_{\mathbf{v},\mathcal{H}_-}\Lambda_{\mathcal{B}_c,\mathcal{U}_m^\star}(\mathcal{H}_-,\mathbf{v}),\ \text{where}$$

$$\Lambda_{\mathcal{B}_c,\mathcal{U}_m^\star}(\mathcal{H}_-,\mathbf{v}) = \frac{\max_{\mathbf{b}\in\mathcal{B}_c}\sum_{t=1}^{T}V(\mathbf{b};\boldsymbol{\beta}_-^t)}{\max_{\mathbf{b}'\in\mathcal{U}_m^\star}\sum_{t=1}^{T}V(\mathbf{b}';\boldsymbol{\beta}_-^t)} \quad (6)$$

for $\mathcal{H}_- = [\boldsymbol{\beta}_-^t]_{t\in[T]}$. In words, $\Lambda_{\mathcal{B}_c}$ is the maximum ratio of the value obtained by the optimal bidding strategy in $\mathcal{B}_c$ and that by the optimal strategy in $\mathcal{U}_m^\star$ over all valuation curves and bid histories in an *offline* setting. We drop the argument $\mathbf{v}$ from $\Lambda_{\mathcal{B}_c,\mathcal{U}_m^\star}(\mathcal{H}_-,\mathbf{v})$ when the context is clear.

**Definition 4** (Richness Ratio). Suppose the clairvoyant chooses the optimal strategy from $\mathcal{B}_c$ (which *may depend on the bid history*, $\mathcal{H}_-$) such that $\Lambda_{\mathcal{B}_c,\mathcal{U}_m^\star} \leq \lambda$, and the bound is tight, i.e., there exist a bid history $\mathcal{H}_-$, a valuation curve $\mathbf{v}$, and $\overline{\delta} > 0$ such that $\Lambda_{\mathcal{B}_c,\mathcal{U}_m^\star}(\mathcal{H}_-,\mathbf{v}) \geq \lambda - \delta$ for any $\delta \in (0,\overline{\delta}]$. Then, the richness ratio of $\mathcal{B}_c$ is $\alpha = 1/\lambda$.

The key idea is that if, in the worst-case scenario for the offline problem, the optimal safe strategy achieves no more than a $\frac{1}{\lambda}$-fraction of the value obtained by the optimal bidding strategy in $\mathcal{B}_c$, then the learner can achieve at most a $\frac{1}{\lambda}$-fraction of that value in the online setting as well. In the subsequent sections, we consider different classes of bidding strategies for the clairvoyant and compute its richness ratio $\alpha$. An immediate corollary of Definition 4 is:

**Corollary 5.2.** *Suppose the clairvoyant chooses the optimal strategy from the class $\mathcal{B}_c$ such that $\Lambda_{\mathcal{B}_c,\mathcal{U}_m^\star} \leq \lambda$ and this bound is tight. Then, Algorithm 1 obtains $\frac{1}{\lambda}$-$\mathsf{REG}_{\mathcal{B}_c} \leq O(M\sqrt{mT\log M})$ under full information feedback and $\frac{1}{\lambda}$-$\mathsf{REG}_{\mathcal{B}_c} \leq O(M^2\sqrt{m^3T\log M})$ under bandit feedback.*

*Proof.* As $\Lambda_{\mathcal{B}_c,\mathcal{U}_m^\star} \leq \lambda$,

$$\frac{1}{\lambda}\text{-}\mathsf{REG}_{\mathcal{B}_c} = \frac{1}{\lambda}\cdot\max_{\mathbf{b}\in\mathcal{B}_c}\sum_{t=1}^{T}V(\mathbf{b};\boldsymbol{\beta}_-^t) - \sum_{t=1}^{T}\mathbb{E}[V(\mathbf{b}^t;\boldsymbol{\beta}_-^t)]$$

$$\leq \max_{\mathbf{b}\in\mathcal{U}_m^\star}\sum_{t=1}^{T}V(\mathbf{b};\boldsymbol{\beta}_-^t) - \sum_{t=1}^{T}\mathbb{E}[V(\mathbf{b}^t;\boldsymbol{\beta}_-^t)] = \mathsf{REG}.$$

By Theorem 4.3, we get the stated regret upper bound. □

### 5.1. Richness Ratio of $m$-uniform Non-Safe Strategies

Here, we consider the clairvoyant class $\mathcal{B}_c$ to be the class of bidding strategies containing at most $m$ bid-quantity pairs, which are feasible only for the given bid history $\mathcal{H}_-$, rather than for every possible sequence of competing bids as in the case of safe strategies. The following theorem characterizes the richness ratio of this class, denoted as $\mathcal{F}_m^{\mathcal{H}_-}$:

**Theorem 5.3.** *For any $m \in \mathbb{N}$, $\Lambda_{\mathcal{F}_m^{\mathcal{H}_-},\mathcal{U}_m^\star} \leq 2$. Furthermore, there exist a bid history and valuation curve such that for any $\delta \in (0,\frac{1}{2}]$, $\Lambda_{\mathcal{F}_m^{\mathcal{H}_-},\mathcal{U}_m^\star}(\mathcal{H}_-,\mathbf{v}) \geq 2-\delta$. Thus, the richness ratio of the class $\mathcal{F}_m^{\mathcal{H}_-}$ is $\alpha = 1/2$.*

Theorem 5.3 implies that restricting the learner to safe bidding strategies has a bounded cost and does not lead to an arbitrary loss in the obtained value as the upper bound is *independent of $m$*. Building on this, Corollary 5.2 shows that Algorithm 1 results in $\frac{1}{2}$-approximate sublinear regret in the online setting when the safe strategies compete against $\mathcal{F}_m^{\mathcal{H}_-}$. Additionally, the factor-of-two loss represents a worst-case scenario, occurring in a highly non-trivial setting (see Appendix D.2). In practice, we expect these strategies to perform near-optimally, a claim further supported by the numerical simulations presented in Section 5.3.

**Proof Sketch of Theorem 5.3 (Upper Bound).** We now present the central ideas to prove the upper bound, which are also used to show the subsequent theorems, albeit with changes specific to each case. Fixing a bid history, $\mathcal{H}_-$, we consider the expression in Eq. (6). Maximizing over all bid histories and valuation curves, we get the desired result.

Firstly, we bridge the we bridge the gap between the bidding strategies of the clairvoyant and the learner by constructing a restricted class of safe strategies, $\mathcal{U}_m^\star(\mathcal{H}_-) \subseteq \mathcal{U}_m^\star$. This restricted class is exponentially smaller in size and is derived based on the optimal strategy selected by the clairvoyant (see exact characterization in Eq. (17) for Theorem 5.3 and Algorithm 3 for Theorem 5.4). We demonstrate that considering this restricted class of safe strategies is sufficient to derive tight upper bounds.

Second, as a crucial step, we show that for any $m$-uniform safe bidding strategy $\mathbf{b} = \langle(w_{Q_1},q_1),\ldots,(w_{Q_m},q_m)\rangle$,

$$V(\mathbf{b};\boldsymbol{\beta}_-) = \max_{\ell\in[m]}V((w_{Q_\ell},Q_\ell);\boldsymbol{\beta}_-)$$

for all competing bids $\boldsymbol{\beta}_-$, where, $Q_\ell = \sum_{j=1}^{\ell}q_j$. Thus, the value obtained by *any* $m$-uniform safe strategy can be determined by knowing the value obtained by the strategies in $\mathcal{U}_1^\star$, allowing us to focus solely on 1-uniform strategies.

As the third key idea, we partition the $T$ rounds into sets $T_j$, based on the least winning bid of the optimal bidding strategy chosen by the clairvoyant (see details in Eq. (15)) and derive lower bounds on the value obtained by 1-uniform

strategies in $\mathcal{U}_m^\star(\mathcal{H}_-)$ in terms of the value obtained by optimal strategy chosen by the clairvoyant in each partition $T_j$. With these three key ideas, we bound the ratio in Eq. (6). The detailed proof is presented in Appendix C.1.

**Proof Sketch of Theorem 5.3 (Tight Lower Bound).** For any $\delta \in (0,1]$, we construct a tight lower-bound instance for any $m \in \mathbb{N}$ by considering an offline problem where $M = T = N^{2m}$ and $K = M + 1$, with $N = O(1/\delta)$. We set $\mathbf{v} = [1, v, \dots, v] \in \mathbb{R}^M$, where $v = 1 - O(\delta)$. The sequence of $T$ auctions is divided into $2m$ partitions of exponentially varying sizes, each with an identical competing bid profile. In each bid profile, there are two types of bids: 'big' and 'small'. All 'big' bids are set to $C \gg 1$, while the 'small' bids remain identical within a partition. The partition sizes, the number of 'small' competing bids, and the exact values of these bids are carefully chosen to ensure that $\Lambda_{\mathcal{F}_m^{\mathcal{H}_-}, \mathcal{U}_m^\star}(\mathcal{H}_-; \mathbf{v}) \geq 2 - \delta$ (see Appendices D.1 to D.2).

## 5.2. Richness Ratio of $m'$-uniform Strategies

We first consider the case when the clairvoyant selects the optimal strategy from the class of *safe* bidding strategies with at most $m'$ bid-quantity pairs where $m' \geq m$.

**Theorem 5.4.** *For any $m, m' \in \mathbb{N}$ such that $m' \geq m$, $\Lambda_{\mathcal{U}_{m'}^\star, \mathcal{U}_m^\star} \leq \frac{m'}{m}$. Additionally, there exist $\mathcal{H}_-$ and $\mathbf{v}$ such that for any $\delta \in (0, \frac{1}{2}]$, $\Lambda_{\mathcal{U}_{m'}^\star, \mathcal{U}_m^\star}(\mathcal{H}_-, \mathbf{v}) \geq \frac{m'}{m} - \delta$. Thus, the richness ratio of the class $\mathcal{U}_{m'}^\star$ is $\alpha = m/m'$.*

Suppose the clairvoyant can now select the optimal strategy from $\mathcal{F}_{m'}^{\mathcal{H}_-}$, the class of bidding strategies with at most $m'$ bid-quantity pairs ($m' \geq m$) that are feasible for the bid history $\mathcal{H}_- = [\boldsymbol{\beta}_-^t]_{t \in [T]}$, but not necessarily safe. Then,

**Theorem 5.5.** *For any $m, m' \in \mathbb{N}$ such that $m' \geq m$, $\Lambda_{\mathcal{F}_{m'}^{\mathcal{H}_-}, \mathcal{U}_m^\star} \leq \frac{2m'}{m}$. Moreover, there exist $\mathcal{H}_-$ and $\mathbf{v}$ such that for any $\delta \in (0, \frac{1}{2}]$, $\Lambda_{\mathcal{F}_{m'}^{\mathcal{H}_-}, \mathcal{U}_m^\star}(\mathcal{H}_-, \mathbf{v}) \geq \frac{2m'}{m} - \delta$. Thus, the richness ratio of the class $\mathcal{F}_{m'}^{\mathcal{H}_-}$ is $\alpha = m/2m'$.*

By Corollary 5.2, for $m' \geq m$, Algorithm 1 obtains $\frac{m}{m'}$-approximate (resp. $\frac{m}{2m'}$-approximate) sublinear regret in the online setting when the clairvoyant draws the optimal strategy from the class of safe (resp. feasible but not necessarily safe) strategies with at most $m'$ bid-quantity pairs.

The richness ratio in Theorem 5.4 and Theorem 5.5 can theoretically become arbitrarily small as $m'$ grows, *for a fixed $m$*. However, the bidder has the flexibility to choose $m$; increasing $m$, though, incurs higher space and time complexity for Algorithm 1. Moreover, for any given pair $(m, m')$, the bound is tight under a highly non-trivial setting (see details in Appendices D.3 to D.4). In practice, as shown in Section 5.3, the observed richness ratios are substantially better than the theoretical bounds, suggesting that the worst-case scenarios are unlikely to arise in real-world applications.

## 5.3. Estimating The Richness Ratio: A Case Study

In this section, we estimate the richness ratio (equivalently $\Lambda_{\mathcal{B}_c, \mathcal{U}_m^\star}$) for the discussed classes of bidding strategies by experiments using EU ETS emission permit auction data from 2022 and 2023. Detailed methods are provided in Appendix E, and the results are shown in Fig. 4.

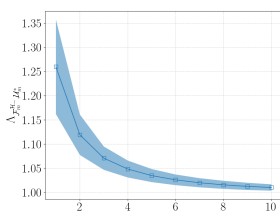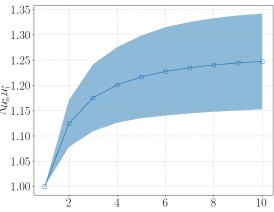

*Figure 4.* The left (resp. right) figure refers to (the upper bound on) $\Lambda_{\mathcal{F}_m^{\mathcal{H}_-}, \mathcal{U}_m^\star}(\mathcal{H}_-)$ (resp. $\Lambda_{\mathcal{U}_m^\star, \mathcal{U}_1^\star}(\mathcal{H}_-)$) as a function of $m$.

**Estimating $\Lambda_{\mathcal{F}_m^{\mathcal{H}_-}, \mathcal{U}_m^\star}(\mathcal{H}_-)$.** The left plot of Fig. 4 shows that (the upper bound for) $\Lambda_{\mathcal{F}_m^{\mathcal{H}_-}, \mathcal{U}_m^\star}(\mathcal{H}_-)$ is significantly better than the theoretical bound of 2. The decaying trend of the plot with increasing $m$ can be attributed to the fact that we are computing an upper bound for the value achieved by the optimal bidding strategy in $\mathcal{F}_m^{\mathcal{H}_-}$, which is *independent of $m$*, rather than the exact value (see Appendix E.2). For $m \geq 4$, even the upper bound of $\Lambda_{\mathcal{F}_m^{\mathcal{H}_-}, \mathcal{U}_m^\star}(\mathcal{H}_-) \sim 1.05$ indicating that the safe bidding strategies are near-optimal.

**Estimating $\Lambda_{\mathcal{U}_m^\star, \mathcal{U}_1^\star}(\mathcal{H}_-)$.** From the right plot of Fig. 4, we observe that the empirical values of $\Lambda_{\mathcal{U}_m^\star, \mathcal{U}_1^\star}(\mathcal{H}_-)$ is significantly better than the worst-case bound $m$ (Theorem 5.4). In fact, the gain obtained by increasing the number of bid-quantity pairs plateaus after $m = 4$ and even for $m = 10$, the ratio of the value obtained by the optimal bidding strategy with at most 10 bid-quantity pairs to that by optimal 1-uniform bidding strategy is $\sim 1.25$.

Both plots show that the richness ratio is significantly better in practice than the theoretical bounds, suggesting that safe strategies with small $m$ are near-optimal in practice.

## 6. Conclusion and Open Problems

We studied bidding in repeated uniform price auctions for a value maximizing buyer with per-round RoI constraints and characterized safe bidding strategies that can be efficiently learnt and are robust against various strong benchmarks. This study suggests several interesting future research directions: (1) analyzing bidding strategies in discriminatory-price ("pay-as-bid") auctions; (2) modeling bidders with time-varying valuations and cumulative RoI constraints; (3) incorporating both RoI and budget constraints for a more realistic model of bidder behavior; and (4) extending the analysis to the more complex setting of combinatorial auctions with non-identical items.

## Impact Statement

This is primarily a theoretical work that discusses a class of bidding strategies in multi-unit auctions. There are many potential societal consequences of our work, none which we feel must be specifically highlighted here.

## Acknowledgements

N.G. and S.S. are partially supported by the MIT Junior Faculty Research Assistance Grant and the Young Investigator Award from the Office of Naval Research (ONR), Award No. N00014-21-1-2776. We thank the anonymous reviewers whose comments greatly improved the presentation of this manuscript.

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

# Contents

# A. Omitted Proofs from Section 3

## A.1. Proof of Theorem 3.1

Before proving Theorem 3.1, we present the following lemma about the bidder's per unit payments, which we will also use to prove several other results.

**Lemma A.1** (Per-unit Payments). *Suppose the bidder bids* $\mathbf{b} = \langle (b_1, q_1), \ldots, (b_m, q_m) \rangle$. *Recall that* $Q_j = \sum_{\ell=1}^{j} q_\ell, \forall j \in [m]$. *Recall that,* $\boldsymbol{\beta}_{-,t}^{-(j)}$ *is the* $j^{th}$ *smallest winning bid in the absence of bids from bidder* $i$ *for round* $t$. *If* $j = 0$, $\boldsymbol{\beta}_{-,t}^{-(j)} = 0$ *and* $j > K$, $\boldsymbol{\beta}_{-,t}^{-(j)} = \infty$. *Then, the per-unit payments by the bidder in round* $t$ *is*

$$p(\boldsymbol{\beta}^t) = \begin{cases} 0, & \text{if } x(\boldsymbol{\beta}^t) = 0 \\ b_\ell, & \text{if } Q_{\ell-1} < x(\boldsymbol{\beta}^t) < Q_\ell \\ \min(b_\ell, \boldsymbol{\beta}_{-,t}^{-(Q_\ell+1)}), & \text{if } x(\boldsymbol{\beta}^t) = Q_\ell \end{cases} . \tag{7}$$

Define the following class of no-overbidding (NOB) strategies with $m$ bid-quantity pairs:

$$\mathcal{S}_m^{\mathbf{NOB}} := \left\{ \mathbf{b} = \langle (b_1, q_1), \ldots, (b_m, q_m) \rangle : b_\ell \leq w_{Q_\ell}, \forall \ell \in [m] \right\} .$$

We first show that $\mathcal{S}_m \subseteq \mathcal{S}_m^{\mathbf{NOB}}$, and then complete the proof by showing $\mathcal{S}_m^{\mathbf{NOB}} \subseteq \mathcal{S}_m$.

**Proof of** $\mathcal{S}_m \subseteq \mathcal{S}_m^{\mathbf{NOB}}$.

**Observation 1.** Overbidding is not a safe bidding strategy. To see this, let $\mathbf{b} = \langle (b_1, q_1), \ldots, (b_\ell, q_\ell), \ldots, (b_m, q_m) \rangle$ be an overbid such that $b_\ell > w_{Q_\ell}$. Now, consider an auction in which the competing bids are:

$$\boldsymbol{\beta}_{-,}^{-(j)} = \begin{cases} \epsilon, & \text{if } 1 \leq j \leq Q_\ell \\ 2b_1, & \text{if } Q_\ell < j \leq K \end{cases} ,$$

where $\epsilon < \frac{b_m}{2}$. Submitting $\mathbf{b}$ gets allocated $Q_\ell$ units and from Lemma A.1, we conclude that the clearing price of auction is $b_\ell$. As $b_\ell > w_{Q_\ell}$ by assumption, the RoI constraint is violated.

As overbidding is not a safe strategy, every safe bidding strategy is a NOB strategy, i.e., $\mathcal{S}_m \subseteq \mathcal{S}_m^{\mathbf{NOB}}$.

**Proof of** $\mathcal{S}_m^{\mathbf{NOB}} \subseteq \mathcal{S}_m$. We now prove that the converse is also true, i.e., every NOB strategy is also a safe strategy. To show this, fix any competing bid, $\boldsymbol{\beta}_-$, and consider any $\mathbf{b} = \langle (b_1, q_1), \ldots, (b_m, q_m) \rangle \in \mathcal{S}_m^{\mathbf{NOB}}$. Suppose bidding $\mathbf{b}$ wins $x(\boldsymbol{\beta})$ units. So, from Lemma A.1, if $x(\boldsymbol{\beta}) = 0$, trivially, $P(\boldsymbol{\beta}) = 0 = V(\boldsymbol{\beta})$. If $Q_{\ell-1} < x(\boldsymbol{\beta}) \leq Q_\ell$, for some $\ell \in [m]$,

$$P(\boldsymbol{\beta}) = x(\boldsymbol{\beta}) \cdot p(\boldsymbol{\beta}) \leq x(\boldsymbol{\beta}) \cdot b_\ell \leq x(\boldsymbol{\beta}) \cdot w_{Q_\ell} \leq x(\boldsymbol{\beta}) \cdot w_{x(\boldsymbol{\beta})} = V(\boldsymbol{\beta}) .$$

The first inequality holds true because bidders' per-unit payment is at most their least winning bid (individual rationality of the auction format), the second is true by definition, and the third is true because the $w_j$ is non-decreasing in $j$ and $x(\boldsymbol{\beta}) \leq Q_\ell$. As the choice of competing bids and bidding strategy $\mathbf{b}$ was arbitrary, we conclude that every strategy in $\mathcal{S}_m^{\mathbf{NOB}}$ is safe, i.e., $\mathcal{S}_m^{\mathbf{NOB}} \subseteq \mathcal{S}_m$, which completes the proof.

### A.1.1. PROOF OF LEMMA A.1

Consider the following three cases:

(1) If $x(\boldsymbol{\beta}^t) = 0$, trivially, $p(\boldsymbol{\beta}^t) = 0$.

(2) Let $Q_{\ell-1} < x(\boldsymbol{\beta}^t) < Q_\ell$. Let $b$ be the last accepted bid after including $\mathbf{b}$, i.e., the smallest bid in $\boldsymbol{\beta}^t = (\mathbf{b}, \boldsymbol{\beta}_-^t)$. Then, $b_\ell \overset{(a)}{\geq} b \overset{(b)}{\geq} b_\ell$.

   I. $(a)$ holds true because the bidder is allocated at least one unit for bid $b_\ell$ and

   II. $(b)$ is correct because they do not acquire at least one unit for bid $b_\ell$. Hence, $p(\boldsymbol{\beta}^t) = b_\ell$.

(3) Suppose $x(\boldsymbol{\beta}^t) = Q_\ell$. If $b_\ell > \boldsymbol{\beta}_{-,t}^{-(Q_\ell+1)}$, then $p(\boldsymbol{\beta}^t) = \boldsymbol{\beta}_{-,t}^{-(Q_\ell+1)}$. However, if $b_\ell \leq \boldsymbol{\beta}_{-,t}^{-(Q_\ell+1)}$, $p(\boldsymbol{\beta}^t) = b_\ell$. So, $p(\boldsymbol{\beta}^t) = \min(b_\ell, \boldsymbol{\beta}_{-,t}^{-(Q_\ell+1)})$.

### A.2. Proof of Theorem 3.2

We show that for any $m \in \mathbb{N}$, most safe bidding strategies are very weakly dominated for which they can removed from $\mathcal{S}_m$, resulting in $\mathcal{S}_m^\star$. We begin by establishing a general result regarding the monotonocity of feasible bid vectors (not necessarily $m$-uniform strategies) for value maximizing bidders. As the result holds for any bid vector, it is also true for $m$-uniform bidding strategies.

**Lemma A.2** (Monotonocity of feasible bids). *Consider two* sorted *bid vectors:* $\mathbf{b} = [b_1, b_2, \ldots, b_k]$ *and* $\mathbf{b}' = [b_1', b_2', \ldots, b_k']$ *such that* $b_j \geq b_j', \forall j \in [k]$. *Suppose* $\mathbf{b}$ *is RoI feasible for some competing bid,* $\boldsymbol{\beta}_-$. *Then,* $\mathbf{b}'$ *is also feasible for* $\boldsymbol{\beta}_-$ *and* $V(\mathbf{b}; \boldsymbol{\beta}_-) \geq V(\mathbf{b}'; \boldsymbol{\beta}_-)$.

Suppose $\mathbf{b} = \langle (b_1, q_1), \ldots, (b_m, q_m) \rangle$ is an underbidding strategy per Definition 3. Consider $\mathbf{b}' = \langle (b_1', q_1'), \ldots, (b_m', q_m') \rangle$ such that $q_j' = q_j$ and $b_j' = w_{Q_j}, \forall j \in [m]$. By Theorem 3.1, we establish that $\mathbf{b}, \mathbf{b}' \in \mathcal{S}_m$. Invoking Lemma A.2 gives us that the underbidding strategy is very weakly dominated, and hence can be removed from $\mathcal{S}_m$ to obtain $\mathcal{S}_m^\star$.

#### A.2.1. PROOF OF LEMMA A.2

Let $\boldsymbol{\beta} = (\mathbf{b}, \boldsymbol{\beta}_-)$ and $\boldsymbol{\beta}' = (\mathbf{b}', \boldsymbol{\beta}_-)$. We first prove that $\mathbf{b}'$ is also feasible for $\boldsymbol{\beta}_-$. Contrary to our claim, suppose $\mathbf{b}'$ is infeasible, i.e., the value obtained by $\mathbf{b}'$ when the competing bids are $\boldsymbol{\beta}_-$ is strictly less than the payment. Suppose $\mathbf{b}'$ is allocated $r'$ units with clearing price $p(\boldsymbol{\beta}')$ such that the RoI constraint is violated:

$$p(\boldsymbol{\beta}') > w_{r'}. \tag{8}$$

Suppose $\mathbf{b}$ is allocated $r$ units when the competing bids are $\boldsymbol{\beta}_-$. By definition of allocation and payment rule, the units allocated and the clearing price in an auction are weakly increasing in bids, so $r' \leq r$ and $p(\boldsymbol{\beta}') \leq p(\boldsymbol{\beta})$. As $\mathbf{b}$ is feasible,

$$p(\boldsymbol{\beta}) \leq w_r. \tag{9}$$

Combining Equations (8) and (9), we have

$$p(\boldsymbol{\beta}) \leq w_r \overset{(a)}{\leq} w_{r'} < p(\boldsymbol{\beta}') \implies p(\boldsymbol{\beta}) < p(\boldsymbol{\beta}'),$$

which is a contradiction. Here $(a)$ is true as $w_j$ is non-increasing in $j$ and $r' \leq r$. So, $\mathbf{b}'$ is feasible.

By definition of the allocation rule, the value obtained in an auction is weakly increasing in the bids submitted. As $\mathbf{b}$ and $\mathbf{b}'$ are both feasible, $V(\mathbf{b}; \boldsymbol{\beta}_-) \geq V(\mathbf{b}'; \boldsymbol{\beta}_-)$.

## B. Omitted Proofs From Section 4

### B.1. Proof of Lemma 4.1

By Theorem 3.2, a strategy $\mathbf{b} = \langle (b_1, q_1), \ldots, (b_\ell, q_\ell) \rangle \in \mathcal{U}_m^\star$ for some $\ell \in [m]$ can be uniquely identified by $\{Q_1, \ldots, Q_\ell\}$ where $Q_j = \sum_{k \leq j} q_k$ because $b_j = w_{Q_j}$ for all $j \in [\ell]$. Let the bid history $\mathcal{H}_- = [\boldsymbol{\beta}_-^t]_{t \in [T]}$. Then, for any round $t$,

$$V(\mathbf{b}; \boldsymbol{\beta}_-^t) = \sum_{j=1}^{\ell} \sum_{k=Q_{j-1}+1}^{Q_j} v_k \cdot \mathbb{I}\left[ w_{Q_j} \geq \boldsymbol{\beta}_{-,t}^{-(k)} \right]$$

Hence, the problem (OFFLINE) can be equivalently expressed as

$$\max_{\mathbf{b} \in \mathcal{U}_m^\star} \sum_{t=1}^{T} V(\mathbf{b}; \boldsymbol{\beta}_-^t) = \max_{\ell \in [m]} \max_{Q_1, \ldots, Q_\ell} \sum_{j=1}^{\ell} \sum_{t=1}^{T} \sum_{k=Q_{j-1}+1}^{Q_j} v_k \cdot \mathbb{I}\left[ w_{Q_j} \geq \boldsymbol{\beta}_{-,t}^{-(k)} \right].$$

### B.2. Proof of Theorem 4.2

$\mathcal{G}(V, E)$ **is a DAG.** For convenience, let $d = (m + 1, 0)$. In the constructed graph, any node $(\ell, j)$ where $\ell \in [m]$ has edges edges either to the next layer, i.e. nodes of the form $(\ell + 1, j')$ where $j' > j$ or to the destination node $(m + 1, 0)$. Hence, the directed graph has a topological sorting of the nodes implying that $\mathcal{G}(V, E)$ is a DAG.

**Bijection between $s$-$d$ paths and strategies in $\mathcal{U}_m^\star$.** Consider a path[9]

$$\mathfrak{p} = s \to (1, z_1) \to \cdots \to (m, z_m) \to d\,.$$

By construction of $\mathcal{G}(V, E)$, edges $e = s \to (1, z_1)$ and $e = (m, z_m) \to d$ always exist. For any $\ell \in [m-1]$, the edge $(\ell, z_\ell) \to (\ell+1, z_{\ell+1})$ exists if $z_\ell < z_{\ell+1}$. With this path $\mathfrak{p}$, we associate the strategy

$$\mathbf{b} = \langle (b_1, q_1), \ldots, (b_m, q_m) \rangle \quad \text{where} \quad b_\ell = w_{z_\ell} \quad \text{and} \quad q_\ell = z_\ell - z_{\ell-1}, \forall \ell \in [m]\,.$$

where $z_0 = 0$. By definition, $Q_j = \sum_{\ell=1}^{j} q_\ell = z_j$ and $b_j = w_{z_j} = w_{Q_j}, \forall j \in [m]$. Hence, $\mathbf{b} \in \mathcal{U}_m^\star$.

Conversely, consider any safe strategy $\mathbf{b} = \langle (b_1, q_1), \ldots, (b_m, q_m) \rangle$ with $b_j = w_{Q_j}$. With this strategy, we associate the $s$-$d$ path $\mathfrak{p}' = s \to (1, Q_1) \to \cdots \to (m, Q_m) \to d$ where $Q_j = \sum_{\ell=1}^{j} q_\ell, \forall j \in [m]$. We claim that $\mathfrak{p}'$ is a *valid* path, i.e., all the edges exist, because by definition $e = s \to (1, Q_1)$ and $e = (m, Q_m) \to d$ always exist. Furthermore, for any $\ell \in [m-1]$, the edge $(\ell, Q_l) \to (\ell+1, Q_{\ell+1})$ also exists as $Q_{\ell+1} - Q_\ell = q_{\ell+1} > 0$.

**Weight of $s$-$d$ paths.** By assumption, $s = (0, 0)$ and $z_0 = 0$. The weight of $\mathfrak{p} = s \to (1, z_1) \to \cdots \to (m, z_m) \to d$ is:

$$
\begin{aligned}
\mathsf{w}(\mathfrak{p}) &= \sum_{e \in \mathfrak{p}} \mathsf{w}(e) = \sum_{\ell=1}^{m} \mathsf{w}((\ell-1, z_{\ell-1}) \to (\ell, z_\ell)) \\
&\stackrel{(4)}{=} \sum_{\ell=1}^{m} \sum_{t=1}^{T} \sum_{k=z_{\ell-1}+1}^{z_\ell} v_k \cdot \mathbb{I}\left[ w_{z_\ell} \geq \boldsymbol{\beta}_{-,t}^{-(k)} \right] \\
&= \sum_{t=1}^{T} \sum_{\ell=1}^{m} \sum_{k=z_{\ell-1}+1}^{z_\ell} v_k \cdot \mathbb{I}\left[ w_{z_\ell} \geq \boldsymbol{\beta}_{-,t}^{-(k)} \right],
\end{aligned}
$$

which is the value obtained by the safe strategy $\mathbf{b} = \langle (b_1, q_1), \ldots, (b_m, q_m) \rangle \in \mathcal{U}_m^\star$ where $b_\ell = w_{z_\ell}, \forall \ell \in [m]$.

**Computing maximum weight path.** As $\mathcal{G}(V, E)$ is a DAG, the edge weights can be negated and the maximum (resp. minimum) weight path problem in the original (resp. 'negated') DAG can be solved in space and time complexity of $O(|V| + |E|) = O(mM^2)$ which is polynomial in the parameters of the problem.

### B.3. Proof of Theorem 4.3

We begin by observing that there is a bijective mapping between $s$-$d$ paths in the DAG $\mathcal{G}^t(V, E)$ and bidding strategies $\mathbf{b} \in \mathcal{U}_m^\star$ (see proof of Theorem 4.2). The core idea of the proof is to show that Algorithm 1 is an *equivalent and efficient* implementation of the Hedge algorithm (Freund & Schapire, 1997) where every $s$-$d$ path is treated as an individual expert. Similar ideas have been utilized in Brânzei et al. (2023); Potfer et al. (2024); Galgana & Golrezaei (2024). For the sake of completeness, we provide the details below.

In Algorithm 1, $\varphi^t(u \to \cdot)$ denotes the probability distribution over the outgoing neighbors of node $u$. By the recursive sampling of nodes (Line 4-7 of Algorithm 1), we get that the probability of selecting $s$-$d$ path $\mathfrak{p}$ is

$$\mathbb{P}^t(\mathfrak{p}) = \prod_{e \in \mathfrak{p}} \varphi^t(e), \tag{10}$$

and for any edge $e = u \to v$ in $\mathcal{G}^t(V, E)$, the edge probabilities, $\varphi^t(e)$, are updated as

$$\varphi^t(e) = \varphi^{t-1}(e) \cdot \exp(\eta \mathsf{w}^{t-1}(e)) \cdot \frac{\Gamma^{t-1}(v)}{\Gamma^{t-1}(u)}, \tag{11}$$

where $\Gamma^{t-1}(d) = 1$ and $\Gamma^{t-1}(\cdot)$ is computed recursively in bottom-to-top fashion for every node $u$ in $\mathcal{G}^t(V, E)$ as follows:

$$\Gamma^{t-1}(u) = \sum_{v: u \to v \in E} \Gamma^{t-1}(v) \cdot \varphi^{t-1}(u \to v) \cdot \exp(\eta \mathsf{w}^{t-1}(u \to v))\,. \tag{12}$$

---

[9]Here, we assume the path has $m+1$ edges without loss of generality. The same argument follows if $\mathfrak{p} = s \to (1, z_1) \to \cdots \to (k, z_k) \to d$ with $k+1$ edges for some $k \in [m]$ is considered instead.

Now, consider a naïve implementation of the Hedge algorithm with learning rate $\eta$ in which each $s$-$d$ path is treated as an individual expert. For $t = 1$, define $\mathbb{P}^1_{\text{HEDGE}}(\mathfrak{p}) = \prod_{e \in \mathfrak{p}} \varphi^1(e)$. For $t \geq 2$, the probability of selecting path $\mathfrak{p}$ in round $t$ is

$$\mathbb{P}^t_{\text{HEDGE}}(\mathfrak{p}) = \frac{\mathbb{P}^{t-1}_{\text{HEDGE}}(\mathfrak{p}) \exp(\eta \sum_{e \in \mathfrak{p}} \mathsf{w}^{t-1}(e))}{\sum_{\mathfrak{p}'} \mathbb{P}^{t-1}_{\text{HEDGE}}(\mathfrak{p}') \exp(\eta \sum_{e \in \mathfrak{p}'} \mathsf{w}^{t-1}(e))} \, .$$

Now, we show that for any path $\mathfrak{p}$ and $t \in [T]$, $\mathbb{P}^t(\mathfrak{p}) = \mathbb{P}^t_{\text{HEDGE}}(\mathfrak{p})$. To this end, we first present the following result:

**Claim B.1.** For any node $u$ in the graph, let $\mathcal{P}(u)$ be the set of paths starting at $u$ and terminating at $d$. Then,

$$\Gamma^{t-1}(u) = \sum_{\mathfrak{p} \in \mathcal{P}(u)} \prod_{e \in \mathfrak{p}} \varphi^{t-1}(e) \cdot \exp(\eta \mathsf{w}^{t-1}(e))$$

*Proof.* We prove the result by induction in a bottom-to-top order. For the base case, $u = d$, $\Gamma^{t-1}(d) = 1$. Suppose the result is true for all the nodes in layer $\ell + 1$ for some $0 \leq \ell \leq m$. By the recursion in Eq. (12), for any node $u$ in layer $\ell$,

$$\begin{aligned}
\Gamma^{t-1}(u) &= \sum_{v: u \to v \in E} \Gamma^{t-1}(v) \cdot \varphi^{t-1}(u \to v) \cdot \exp(\eta \mathsf{w}^{t-1}(u \to v)) \\
&= \sum_{v: u \to v \in E} \Big( \sum_{\mathfrak{p} \in \mathcal{P}(v)} \prod_{e \in \mathfrak{p}} \varphi^{t-1}(e) \cdot \exp(\eta \mathsf{w}^{t-1}(e)) \Big) \varphi^{t-1}(u \to v) \cdot \exp(\eta \mathsf{w}^{t-1}(u \to v)) \\
&= \sum_{\mathfrak{p} \in \mathcal{P}(u)} \prod_{e \in \mathfrak{p}} \varphi^{t-1}(e) \cdot \exp(\eta \mathsf{w}^{t-1}(e)),
\end{aligned}$$

where the first equality follows from induction hypothesis. $\square$

We will show by induction on $t \in [T]$ that for any path $\mathfrak{p}$, $\mathbb{P}^t(\mathfrak{p}) = \mathbb{P}^t_{\text{HEDGE}}(\mathfrak{p})$. For $t = 1$ and any path $\mathfrak{p}$, $\mathbb{P}^1_{\text{HEDGE}}(\mathfrak{p}) = \prod_{e \in \mathfrak{p}} \varphi^1(e) = \mathbb{P}^1(\mathfrak{p})$. Here, the second equality holds by Eq. (10). Suppose the result holds for round $t - 1$, i.e., $\mathbb{P}^{t-1}(\mathfrak{p}) = \mathbb{P}^{t-1}_{\text{HEDGE}}(\mathfrak{p}) = \prod_{e \in \mathfrak{p}} \varphi^{t-1}(e)$. For round $t$,

$$\mathbb{P}^t(\mathfrak{p}) = \prod_{e \in \mathfrak{p}} \varphi^t(e) \overset{(11)}{=} \prod_{e = u \to v \in \mathfrak{p}} \varphi^{t-1}(e) \cdot \exp(\eta \mathsf{w}^{t-1}(e)) \cdot \frac{\Gamma^{t-1}(v)}{\Gamma^{t-1}(u)} = \mathbb{P}^{t-1}_{\text{HEDGE}}(\mathfrak{p}) \cdot \exp\Big(\eta \sum_{e \in \mathfrak{p}} \mathsf{w}^{t-1}(e)\Big) \cdot \frac{\Gamma^{t-1}(d)}{\Gamma^{t-1}(s)},$$

where the last equality follows by telescoping product and invoking the induction hypothesis. By definition, $\Gamma^{t-1}(d) = 1$. By Claim B.1,

$$\Gamma^{t-1}(s) = \sum_{\mathfrak{p}'} \prod_{e \in \mathfrak{p}'} \varphi^{t-1}(e) \cdot \exp(\eta \mathsf{w}^{t-1}(e)) = \sum_{\mathfrak{p}'} \mathbb{P}^{t-1}_{\text{HEDGE}}(\mathfrak{p}') \cdot \exp\Big(\eta \sum_{e \in \mathfrak{p}'} \mathsf{w}^{t-1}(e)\Big)$$

Substituting the values gives $\mathbb{P}^t(\mathfrak{p}) = \mathbb{P}^t_{\text{HEDGE}}(\mathfrak{p})$ implying Algorithm 1 is a correct implementation of the Hedge algorithm.

### B.3.1. BANDIT FEEDBACK MODEL

In the bandit setting, the bidder learns about their own allocation only for which $\mathsf{w}^t(e)$ can not be exactly computed for all the edges. Hence, we use an unbiased estimator, $\widehat{\mathsf{w}}^t(e)$, of $\mathsf{w}^t(e)$ as follows:

$$\widehat{\mathsf{w}}^t(e) = \overline{\mathsf{w}}(e) - \frac{\overline{\mathsf{w}}(e) - \mathsf{w}^t(e)}{p^t(e)} \cdot \mathbb{I}\big[e \in \mathfrak{p}^t\big] \quad \text{where} \quad p^t(e) = \sum_{\mathfrak{p}: e \in \mathfrak{p}} \mathbb{P}^t(\mathfrak{p}) \, .$$

Here, $p^t(e)$ is the probability of selecting edge $e$ in round $t$, $\mathbb{P}^t$ is the distribution over all $s$-$d$ paths in $\mathcal{G}^t(V, E)$. Observe that $\widehat{\mathsf{w}}^t(e)$ is well defined as $p^t(e) > 0, \forall e \in E, \forall t \in [T]$. Here, $\overline{\mathsf{w}}(e)$ is an edge-dependent upper bound of the edge weight $\mathsf{w}^t(e)$. Formally, for $\ell \in [m]$,

$$\overline{\mathsf{w}}(e) = \begin{cases} j' - j, & \text{if } e = (\ell - 1, j) \to (\ell, j') \\ 0, & \text{if } e = (\ell, j) \to d \, . \end{cases}$$

The algorithm followed by the bidder in the bandit setting is identical to Algorithm 1, with each instance of $\mathsf{w}^t(e)$ replaced by $\widehat{\mathsf{w}}^t(e), \forall t \in [T], \forall e \in E$. Clearly, $\mathbb{E}[\widehat{\mathsf{w}}^t(e)] = \mathsf{w}^t(e)$. The unbiased estimator $\widehat{\mathsf{w}}^t(e)$ is different from the natural importance-weighted estimator $\frac{\mathsf{w}^t(e)}{p^t(e)} \cdot \mathbb{I}[e \in \mathfrak{p}^t]$ for technical reasons (similar to Lattimore & Szepesvári (2020, Eq. 11.6)) which we will clarify shortly.

### B.3.2. REGRET UPPER BOUND

**Full Information Setting.** Having shown that Algorithm 1 is equivalent to the Hedge algorithm, we now recall its regret bound from Brânzei et al. (2023, Corollary 1) which states that for learning rate $\eta$:

$$\mathsf{REG} \leq -\frac{1}{\eta} \max_{\mathfrak{p}} \log(\mathbb{P}^1(\mathfrak{p})) + \frac{\eta T M^2}{8} \,.$$

Observe that $\mathbb{P}^1(\mathfrak{p}) \geq M^{-m}$, which implies,

$$\mathsf{REG} \leq \frac{m \log M}{\eta} + \frac{\eta T M^2}{8} \,.$$

Setting $\eta = \frac{1}{M}\sqrt{\frac{8m \log M}{T}}$, we get $\mathsf{REG} \leq O(M\sqrt{mT \log M})$.

**Bandit Setting.** Using the standard analysis for EXP3 algorithm (see Lattimore & Szepesvári (2020, Chapter 11)), we get

$$\mathsf{REG} = -\frac{1}{\eta} \max_{\mathfrak{p}} \log(\mathbb{P}^1(\mathfrak{p})) + \sum_{t=1}^{T} \mathbb{E}\left[ \frac{1}{\eta} \log\left( \sum_{\mathfrak{p}} \mathbb{P}^t(\mathfrak{p}) e^{\eta \widehat{\mathsf{w}}^t(\mathfrak{p})} \right) - \sum_{\mathfrak{p}} \mathbb{P}^t(\mathfrak{p}) \widehat{\mathsf{w}}^t(\mathfrak{p}) \right]$$

where $\widehat{\mathsf{w}}^t(\mathfrak{p}) = \sum_{e \in \mathfrak{p}} \widehat{\mathsf{w}}^t(e)$. Note that $\widehat{\mathsf{w}}^t(\mathfrak{p}) = \sum_{e \in \mathfrak{p}} \widehat{\mathsf{w}}^t(e) \leq \sum_{e \in \mathfrak{p}} \overline{\mathsf{w}}(e) \leq M$ and $\mathbb{P}^1(\mathfrak{p}) \geq M^{-m}$. For $\eta \leq \frac{1}{M}$,

$$\mathsf{REG} \leq \frac{m \log M}{\eta} + \eta \sum_{t=1}^{T} \sum_{\mathfrak{p}} \mathbb{P}^t(\mathfrak{p}) \mathbb{E}[\widehat{\mathsf{w}}^t(\mathfrak{p})^2]$$

$$\leq \frac{m \log M}{\eta} + \eta m \sum_{t=1}^{T} \sum_{\mathfrak{p}} \mathbb{P}^t(\mathfrak{p}) \sum_{e \in \mathfrak{p}} \mathbb{E}[\widehat{\mathsf{w}}^t(e)^2]$$

where in the first inequality, we used $e^x \leq 1 + x + x^2$ for $x \leq 1$ and $\log(1 + x) \leq x$ for $x \geq 0$ and the second inequality follows from Cauchy-Schwarz. Observe that

$$\sum_{\mathfrak{p}} \mathbb{P}^t(\mathfrak{p}) \sum_{e \in \mathfrak{p}} \mathbb{E}[\widehat{\mathsf{w}}^t(e)^2] = \sum_{e} \mathbb{E}[\widehat{\mathsf{w}}^t(e)^2] \sum_{\mathfrak{p}:e \in \mathfrak{p}} \mathbb{P}^t(\mathfrak{p}) = \sum_{e} p^t(e) \mathbb{E}[\widehat{\mathsf{w}}^t(e)^2],$$

where the last equality holds as $p^t(e) = \sum_{\mathfrak{p}:e \in \mathfrak{p}} \mathbb{P}^t(\mathfrak{p})$.

$$\sum_{e} p^t(e) \mathbb{E}[\widehat{\mathsf{w}}^t(e)^2] = \sum_{e} p^t(e) \left[ \overline{\mathsf{w}}(e)^2 (1 - p^t(e)) + \left( \overline{\mathsf{w}}(e) - \frac{\overline{\mathsf{w}}(e) - \mathsf{w}^t(e)}{p^t(e)} \right)^2 p^t(e) \right]$$

$$= \sum_{e} \mathsf{w}^t(e)^2 p^t(e) + (\overline{\mathsf{w}}(e) - \mathsf{w}^t(e))^2 (1 - p^t(e)) \leq \sum_{e} \overline{\mathsf{w}}(e)^2 \,.$$

Observe that $\sum_{e} \overline{\mathsf{w}}(e)^2 \leq |E| M^2 \lesssim m M^4$. Hence,

$$\mathsf{REG} \leq \frac{m \log M}{\eta} + \eta m \sum_{t=1}^{T} \sum_{\mathfrak{p}} \mathbb{P}^t(\mathfrak{p}) \sum_{e \in \mathfrak{p}} \mathbb{E}[\widehat{\mathsf{w}}^t(e)^2]$$

$$\lesssim \frac{m \log M}{\eta} + \eta T m^2 M^4$$

Setting $\eta = \frac{1}{M^2}\sqrt{\frac{\log M}{mT}}$, we get $\mathsf{REG} \leq O(M^2 \sqrt{m^3 T \log M})$.

### B.3.3. TIME COMPLEXITY

In the full information setting, the running time bottleneck is Eq. (12). Thus, Algorithm 1 runs in $|E| = O(mM^2)$ time per round. The bottleneck of running Algorithm 1 under the bandit feedback is the efficient computation of the marginal distribution, $p^t(e)$. We claim that this can be done in $O(M|E|) = O(mM^3)$ time per-round.

We compute the marginal probabilities in a recursive manner over the topological sorting of the nodes in the DAG as follows: for any edge starting at the source, $e = s \to u$, $p^t(e) = \varphi^t(e)$. For any other edge of the form $e = u \to v$, using law of total probabilities

$$p^t(u \to v) = \sum_{x \in \mathcal{N}_u^{in}} \mathbb{P}[(u \to v)|(x \to u)] \cdot p^t(x \to u) = \sum_{x \in \mathcal{N}_u^{in}} \varphi^t(u \to v) \cdot p^t(x \to u),$$

where $\mathcal{N}_u^{in}$ is the set of in-neighbors of node $u$ and $\mathbb{P}[(u \to v)|(x \to u)]$ is the conditional probability of selecting edge $u \to v$ once the edge $x \to u$ is chosen. The final equality holds because by Algorithm 1, we get that at any node, edges are chosen in a Markovian manner implying $\mathbb{P}[(u \to v)|(x \to u)] = \varphi^t(u \to v)$. Observe that probabilities $p^t(x \to u)$ for any $x \in \mathcal{N}_u^{in}$ are known while computing $p^t(u \to v)$ as the edge $x \to u$ precedes $u \to v$ in the topological sort. So, the total run time is

$$O\Big(|E| \cdot \max_{u \in V: u \neq d} |\mathcal{N}_u^{in}|\Big) = O(M|E|) = O(mM^3).$$

Here, we exclude the destination node $d$ while maximizing over the nodes as $d$ has no outgoing edges.

### B.4. Proof of Theorem 4.4

For $m = 1$, let $K = M$ be an even integer. Let $\mathbf{v} = [1, \dots, 1, v, \dots, v]$. Here, the first $\frac{M}{2}$ entries are 1 and the remaining $\frac{M}{2}$ entries are $v$. Let $v = 1 - \delta$. Define $\delta' = \frac{\delta}{2M}$. Consider two scenarios:

**Scenario 1.** In this scenario, for every $t \in [T]$, the competing bids $\boldsymbol{\beta}_-^t$ are:

$$\boldsymbol{\beta}_-^t = \begin{cases} [1 - \delta', \dots, 1 - \delta'], & \text{w.p. } \frac{1}{2} + \delta, \\ [\frac{1+v}{2} - \delta', \dots, \frac{1+v}{2} - \delta'], & \text{w.p. } \frac{1}{2} - \delta, \end{cases}$$

**Scenario 2.** In this scenario, for every $t \in [T]$, the competing bids $\boldsymbol{\beta}_-^t$ are:

$$\boldsymbol{\beta}_-^t = \begin{cases} [1 - \delta', \dots, 1 - \delta'], & \text{w.p. } \frac{1}{2} - \delta, \\ [\frac{1+v}{2} - \delta', \dots, \frac{1+v}{2} - \delta'], & \text{w.p. } \frac{1}{2} + \delta, \end{cases}$$

for some $\delta \in (0, 1/4)$ that would be determined shortly. Assume the randomness used in different rounds are independent.

Let $P$ and $Q$ be the distribution of $[\boldsymbol{\beta}_-^t]_{t \in [T]}$ for scenario 1 and 2 respectively. Then, for $\delta \in (0, \frac{1}{4})$,

$$\text{KL}(P||Q) = T \cdot \text{KL}(\text{BERN}(0.5 + \delta)||\text{BERN}(0.5 - \delta)) = 2T\delta \log\left(\frac{1 + 2\delta}{1 - 2\delta}\right) \leq \frac{8T\delta^2}{1 - 2\delta} \leq 16T\delta^2$$

where the first inequality follows from $\log(\frac{1+x}{1-x}) \leq \frac{2x}{1-x}$. By Tsybakov (2009, Lemma 2.6),

$$1 - \text{TV}(P, Q) \geq \frac{1}{2} \exp\left(-\text{KL}(P||Q)\right) \geq \frac{1}{2} \exp\left(-16T\delta^2\right).$$

Consider the class of 1-uniform safe strategies. We first show that we need to consider only two strategies out of the $M$ possible strategies. Recall that any strategy in this class is of the form $(w_q, q)$.

**Case 1.** $1 \leq q \leq \frac{M}{2}$: Here, $w_q = 1, \forall q$. So, $(1, \frac{M}{2})$ weakly dominates all the strategies of the form $(1, q)$ for $1 \leq q \leq \frac{M}{2}$.

**Case 2.** $\frac{M}{2} + 1 \leq q \leq M$: In this interval, the highest bid value is for $q = \frac{M}{2} + 1$ due to diminishing marginal returns property. Note that,

$$w_{\frac{M}{2}+1} = \frac{\frac{M}{2} + 1 - \delta}{\frac{M}{2} + 1} = 1 - \frac{\delta}{\frac{M}{2} + 1} \leq 1 - \frac{\delta}{M} < 1 - \delta'.$$

Hence, no strategy of form $(w_q, q)$ for $q \geq \frac{M}{2} + 1$ is allocated any unit in when the competing bids are $[1 - \delta', \ldots, 1 - \delta']$. The smallest bid value for $\frac{M}{2} + 1 \leq q \leq M$ is for $q = M$. Note that, $w_M = \frac{1+v}{2} > \frac{1+v}{2} - \delta'$. Hence, $(w_M, M) = (\frac{1+v}{2}, M) = (1 - \frac{\delta}{2}, M)$ is allocated all the units when the competing bids are $[\frac{1+v}{2} - \delta', \ldots, \frac{1+v}{2} - \delta']$ and, by definition, weakly dominates all the strategies of the form $(w_q, q)$ for $q \geq \frac{M}{2} + 1$. Hence, we have two undominated (for the constructed competing bids) strategies: $\mathbf{b}_1 = (1, \frac{M}{2})$ and $\mathbf{b}_2 = (1 - \frac{\delta}{2}, M)$.

Now, we compute the expected value obtained by $\mathbf{b}_1$ and $\mathbf{b}_2$ for scenarios $P$ and $Q$.

$$\mathbb{E}_P[V(\mathbf{b}_1; \boldsymbol{\beta}_-^t)] = \mathbb{E}_Q[V(\mathbf{b}_1; \boldsymbol{\beta}_-^t)] = \frac{M}{2}$$

$$\mathbb{E}_P[V(\mathbf{b}_2; \boldsymbol{\beta}_-^t)] = M\left(\frac{1}{2} - \delta\right)\left(\frac{1+v}{2}\right) = \frac{M}{2}\left(\frac{1}{2} - \delta\right)(2 - \delta) < \frac{M}{2}$$

$$\mathbb{E}_Q[V(\mathbf{b}_2; \boldsymbol{\beta}_-^t)] = M\left(\frac{1}{2} + \delta\right)\left(\frac{1+v}{2}\right) = \frac{M}{2}\left(\frac{1}{2} + \delta\right)(2 - \delta) > \frac{M}{2}$$

So, $\mathbf{b}_1$ (resp. $\mathbf{b}_2$) is optimal for scenario $P$ (resp. scenario $Q$). Now, consider the distribution $\frac{P+Q}{2}$, i.e., $\text{BERN}(0.5)$.

$$\mathbb{E}_{\frac{P+Q}{2}}[V(\mathbf{b}_1; \boldsymbol{\beta}_-^t)] = \frac{M}{2} \quad \text{and} \quad \mathbb{E}_{\frac{P+Q}{2}}[V(\mathbf{b}_2; \boldsymbol{\beta}_-^t)] = \frac{M}{2}\left(\frac{1+v}{2}\right) \leq \frac{M}{2}$$

$$\implies \max_{\mathbf{b} \in \mathcal{U}_1^\star} \mathbb{E}_{\frac{P+Q}{2}}[V(\mathbf{b}; \boldsymbol{\beta}_-^t)] \leq \frac{M}{2} \tag{13}$$

Hence, for any $\mathbf{b} \in \mathcal{U}_1^\star$, and any round $t \in [T]$,

$$\max_{\mathbf{b}^* \in \mathcal{U}_1^\star} \mathbb{E}_P[V(\mathbf{b}^*; \boldsymbol{\beta}_-^t) - V(\mathbf{b}; \boldsymbol{\beta}_-^t)] + \max_{\mathbf{b}^* \in \mathcal{U}_1^\star} \mathbb{E}_Q[V(\mathbf{b}^*; \boldsymbol{\beta}_-^t) - V(\mathbf{b}; \boldsymbol{\beta}_-^t)]$$

$$\geq \max_{\mathbf{b}^* \in \mathcal{U}_1^\star} \mathbb{E}_P[V(\mathbf{b}^*; \boldsymbol{\beta}_-^t)] + \max_{\mathbf{b}^* \in \mathcal{U}_1^\star} \mathbb{E}_Q[V(\mathbf{b}^*; \boldsymbol{\beta}_-^t)] - 2 \max_{\mathbf{b}^* \in \mathcal{U}_1^\star} \mathbb{E}_{\frac{P+Q}{2}}[V(\mathbf{b}^*; \boldsymbol{\beta}_-^t)]$$

$$= \mathbb{E}_P[V(\mathbf{b}_1; \boldsymbol{\beta}_-^t)] + \mathbb{E}_Q[V(\mathbf{b}_2; \boldsymbol{\beta}_-^t)] - 2 \max_{\mathbf{b}^* \in \mathcal{U}_1^\star} \mathbb{E}_{\frac{P+Q}{2}}[V(\mathbf{b}^*; \boldsymbol{\beta}_-^t)]$$

$$\overset{(13)}{\geq} \frac{M}{2} + \frac{M}{2}\left(\frac{1}{2} + \delta\right)(2 - \delta) - 2 \cdot \frac{M}{2} = \frac{M}{2}\left(\frac{3\delta}{2} - \delta^2\right) \geq \frac{5M\delta}{8},$$

where the last inequality follows as $\delta \in (0, \frac{1}{4})$. Hence, any strategy $\mathbf{b} \in \mathcal{U}_1^\star$ incurs a total regret of $\frac{5MT\delta}{16}$ either under $P$ or under $Q$. By two-point method from Tsybakov (2009, Theorem 2.2),

$$\mathbb{E}_{\frac{P+Q}{2}}[\mathsf{REG}] \geq \frac{5MT\delta}{16} \cdot (1 - \mathsf{TV}(P, Q)) \geq \frac{5MT\delta}{32} \exp\left(-16T\delta^2\right)$$

Setting $\delta = \frac{1}{4\sqrt{2T}}$, we get $\mathbb{E}_{\frac{P+Q}{2}}[\mathsf{REG}] \geq \frac{M\sqrt{T}}{36\sqrt{e}}$. Hence, $\mathbb{E}_{\frac{P+Q}{2}}[\mathsf{REG}] = \Omega(M\sqrt{T})$.

# C. Omitted Proofs from Section 5

## C.1. Proof of Theorem 5.3

To prove that $\Lambda_{\mathcal{F}_m^{\mathcal{H}_-}, \mathcal{U}_m^\star} \leq 2$ for any $m \in \mathbb{N}$, recall from the proof sketch that we first defined the following metric for any bid history, $\mathcal{H}_- = [\boldsymbol{\beta}_-^t]_{t \in [T]}$,

$$\Lambda_{\mathcal{F}_m^{\mathcal{H}_-}, \mathcal{U}_m^\star}(\mathcal{H}_-) = \frac{\max_{\mathbf{b} \in \mathcal{F}_m^{\mathcal{H}_-}} \sum_{t=1}^T V(\mathbf{b}; \boldsymbol{\beta}_-^t)}{\max_{\mathbf{b} \in \mathcal{U}_m^\star} \sum_{t=1}^T V(\mathbf{b}; \boldsymbol{\beta}_-^t)}. \tag{14}$$

Let $\mathbf{b}_m^{\mathsf{OPT}}(\mathcal{H}_-) = \langle(b_1^*, q_1^*), \ldots, (b_m^*, q_m^*)\rangle$ where $\mathbf{b}_m^{\mathsf{OPT}}(\mathcal{H}_-) := \text{argmax}_{\mathbf{b} \in \mathcal{F}_m^{\mathcal{H}_-}} \sum_{t=1}^T V(\mathbf{b}; \boldsymbol{\beta}_-^t)$. [10]Define $Q_\ell^* = \sum_{j \leq \ell} q_j^*$. Suppose $\mathbf{b}_m^{\mathsf{OPT}}(\mathcal{H}_-)$ is allocated $r_t^*$ units in round $t$. For any $j \in [m]$, let $T_j$ be the set of rounds in which the

---

[10]We assumed $\mathbf{b}_m^{\mathsf{OPT}}(\mathcal{H}_-) = \langle(b_1^*, q_1^*), \ldots, (b_m^*, q_m^*)\rangle$ without loss of generality. The proof also follows if we considered $\mathbf{b}_m^{\mathsf{OPT}}(\mathcal{H}_-) = \langle(b_1^*, q_1^*), \ldots, (b_k^*, q_k^*)\rangle$ for some $k \in [m]$ instead.

least winning bid is $b_j^*$, i.e.,

$$T_j = \left\{ t \in [T] : Q_{j-1}^* < r_t^* \le Q_j^* \right\}. \tag{15}$$

For any $j \in [m]$, partition $T_j$ into $T_{j,0}$ and $T_{j,1}$ such that $T_{j,0}$ is the set of rounds where the bidder gets *strictly less than* $Q_j^*$ units and $T_{j,1}$ is the set of rounds when they get *exactly* $Q_j^*$ units:

$$T_{j,0} = \left\{ t \in T_j : r_t^* < Q_j^* \right\}, \qquad T_{j,1} = \left\{ t \in T_j : r_t^* = Q_j^* \right\}. \tag{16}$$

For any $j \in [m]$, let

$$\widehat{Q}_j = \begin{cases} \max\{r_t^* : t \in T_{j,0}\}, & \text{if } T_{j,0} \ne \emptyset \\ Q_j^* & \text{if } T_{j,0} = \emptyset. \end{cases}$$

In other words, $\widehat{Q}_j$ is the $2^{nd}$ highest number of units allocated to $\mathbf{b}_m^{\mathsf{OPT}}(\mathcal{H}_-)$ over the rounds in $T_j$ (or the highest if $T_{j,0} = \emptyset$ or $T_{j,1} = \emptyset$).

**Step 1. Constructing a Restricted Class of Safe Bidding Strategies.** Here, as a crucial part of the proof, we construct a restricted class of safe bidding strategies, denoted by $\mathcal{U}_m^{\star}(\mathcal{H}_-)$, where $\mathcal{U}_m^{\star}(\mathcal{H}_-) \subset \mathcal{U}_m^{\star}$. This construction serves two purposes. First, it reduces the search space for the optimal safe bidding strategy. Second, and more importantly, it enables us to establish a connection between the optimal safe bidding strategy and $\mathbf{b}_m^{\mathsf{OPT}}(\mathcal{H}_-)$.

In defining the restricted class, we use the quantities, $\{\widehat{Q}_j, Q_j^*\}_{j \in [m]}$ as follows:

$$\mathcal{U}_m^{\star}(\mathcal{H}_-) = \left\{ \mathbf{b} = \langle (b_1, q_1), \ldots, (b_m, q_m) \rangle : b_\ell = w_{Q_\ell}, \ Q_\ell \in \{\widehat{Q}_\ell, Q_\ell^*\}, \quad \forall \ell \in [m] \right\}. \tag{17}$$

Recall that any strategy in $\mathcal{U}_m^{\star}$ with $m$ bid-quantity pairs takes the form of $\mathbf{b} = \langle (b_1, q_1), \ldots, (b_m, q_m) \rangle : b_\ell = w_{Q_\ell}, \forall \ell \in [m]$. The strategies in $\mathcal{U}_m^{\star}(\mathcal{H}_-)$ also have the same structure but as an important difference, for any $\ell \in [m]$, we enforce $Q_\ell \in \{\widehat{Q}_\ell, Q_\ell^*\}$. Observe that for any $\ell \in [m-1]$,

$$Q_\ell \le Q_\ell^* < \widehat{Q}_{\ell+1} \le Q_{\ell+1},$$

where the first and third inequalities follow directly from the definition of $Q_\ell$ and the second one follows from the definition of $\widehat{Q}_{\ell+1}$. So, $Q_\ell$'s are distinct and ordered. Further observe that the number of bidding strategies in $\mathcal{U}_m^{\star}(\mathcal{H}_-)$ is $O(2^m)$; significantly smaller than the number of strategies in $\mathcal{U}_m^{\star}$, which is $O(M^m)$.

With the definition of the restricted class of safe bidding strategies, we have

$$\Lambda_{\mathcal{F}_m^{\mathcal{H}_-}, \mathcal{U}_m^{\star}}(\mathcal{H}_-) = \frac{\max_{\mathbf{b} \in \mathcal{F}_m^{\mathcal{H}_-}} \sum_{t=1}^{T} V(\mathbf{b}; \boldsymbol{\beta}_-^t)}{\max_{\mathbf{b} \in \mathcal{U}_m^{\star}} \sum_{t=1}^{T} V(\mathbf{b}; \boldsymbol{\beta}_-^t)} \le \frac{\max_{\mathbf{b} \in \mathcal{F}_m^{\mathcal{H}_-}} \sum_{t=1}^{T} V(\mathbf{b}; \boldsymbol{\beta}_-^t)}{\max_{\mathbf{b} \in \mathcal{U}_m^{\star}(\mathcal{H}_-)} \sum_{t=1}^{T} V(\mathbf{b}; \boldsymbol{\beta}_-^t)}. \tag{18}$$

**Step 2. Value Decomposition for $m$-uniform Strategies.** Let $V_{j,k}$ be the time-average value obtained by $\mathbf{b}_m^{\mathsf{OPT}}(\mathcal{H}_-)$ in the set of rounds in $T_{j,k}$ (as defined in Eq. (16)). Formally, $\forall j \in [m], k \in \{0, 1\}$,

$$V_{j,k} = \frac{1}{|T_{j,k}|} \sum_{t \in T_{j,k}} V(\mathbf{b}_m^{\mathsf{OPT}}(\mathcal{H}_-), \boldsymbol{\beta}_-^t).$$

Define $W_k = \sum_{j=1}^{k} v_j$. Note that $V_{j,1} = W_{Q_j^*}$ because for any $t \in T_{j,1}$, we have $r_t^* = Q_j^*$. For any $\mathbf{b} \in \mathcal{U}_m^{\star}(\mathcal{H}_-)$, where $\mathbf{b} = \langle (b_1, q_1), \ldots, (b_m, q_m) \rangle : b_\ell = w_{Q_\ell}$, and $Q_\ell \in \{Q_\ell^*, \widehat{Q}_\ell\}$ for any $\ell \in [m]$, define $N^*, \widehat{N} \subseteq [m]$ as follows:

$$N^* = \{j : Q_j = Q_j^*, j \in [m]\} \quad \text{and} \quad \widehat{N} = \{j : Q_j = \widehat{Q}_j < Q_j^*, j \in [m]\}.$$

We now present a crucial result that expresses the value obtained by a $m$-uniform bidding strategy as a function of the value obtained by $m$ 1-uniform strategies.

**Lemma C.1.** *Let* $\mathbf{b} = \langle (w_{Q_1}, q_1), \ldots, (w_{Q_m}, q_m) \rangle$ *be a $m$-uniform safe bidding strategy for some $m \in \mathbb{N}$. Then, for any competing bids $\boldsymbol{\beta}_-$,*

$$V(\mathbf{b}; \boldsymbol{\beta}_-) = \max_{\ell \in [m]} V((w_{Q_\ell}, Q_\ell); \boldsymbol{\beta}_-),$$

*where we recall that $Q_\ell = \sum_{j=1}^{\ell} q_j, \forall \ell \in [m]$.*

A key consequence of Lemma C.1 is that knowing the value obtained by the strategies in $\mathcal{U}_1^\star$ is sufficient to compute the value obtained *any* $m$-uniform safe bidding strategy. By Lemma C.1, for any round $t \in T_j$, and $\mathbf{b} \in \mathcal{U}_m^\star(\mathcal{H}_-)$, we get

$$V(\mathbf{b}; \boldsymbol{\beta}_-^t) = \max_{\ell \in [m]} V((w_{Q_\ell}, Q_\ell); \boldsymbol{\beta}_-^t) \geq V((w_{Q_j}, Q_j); \boldsymbol{\beta}_-^t), \tag{19}$$

$$\implies \sum_{t=1}^{T} V(\mathbf{b}; \boldsymbol{\beta}_-^t) \geq \sum_{j \in N^*} \sum_{t \in T_j} V((w_{Q_j}, Q_j); \boldsymbol{\beta}_-^t) + \sum_{j \in \widehat{N}} \sum_{t \in T_j} V((w_{Q_j}, Q_j); \boldsymbol{\beta}_-^t). \tag{20}$$

**Step 3. Allocation Lower Bounds for 1-uniform Strategies.** For any $j \in [m]$, we now invoke the following lemma to establish lower bound on $\sum_{t \in T_j} V((w_{Q_j}, Q_j); \boldsymbol{\beta}_-^t)$.

**Lemma C.2.** *Let $\mathbf{b} \in \mathcal{U}_m^\star(\mathcal{H}_-)$, where $\mathbf{b} = \langle (b_1, q_1), \ldots, (b_m, q_m) \rangle : b_j = w_{Q_j}$, and $Q_j \in \{Q_j^*, \widehat{Q}_j\}, \forall j \in [m]$. Then, for any $j \in [m]$,*

*(a) if $Q_j = \widehat{Q}_j < Q_j^*$ (i.e., $j \in \widehat{N}$), we have*

$$\sum_{t \in T_j} V((w_{Q_j}, Q_j); \boldsymbol{\beta}_-^t) \geq V_{j,0}|T_{j,0}| + W_{\widehat{Q}_j}|T_{j,1}|. \tag{21}$$

*(b) If $Q_j = Q_j^*$ (i.e., $j \in N^*$), we have*

$$\sum_{t \in T_j} V((w_{Q_j}, Q_j); \boldsymbol{\beta}_-^t) \geq W_{Q_j^*}|T_{j,1}|. \tag{22}$$

Note that the right hand side of Eq. (20) depends only on the choice of the partitions $N^*$ and $\widehat{N}$. Substituting the lower bound from Lemma C.2 in Eq. (20), we establish that,

$$\Lambda_{\mathcal{F}_m^{\mathcal{H}_-}, \mathcal{U}_m^\star}(\mathcal{H}_-) \overset{(18)}{\leq} \frac{\max_{\mathbf{b} \in \mathcal{F}_m^{\mathcal{H}_-}} \sum_{t=1}^{T} V(\mathbf{b}; \boldsymbol{\beta}_-^t)}{\max_{\mathbf{b} \in \mathcal{U}_m^\star(\mathcal{H}_-)} \sum_{t=1}^{T} V(\mathbf{b}; \boldsymbol{\beta}_-^t)}$$

$$\leq \frac{\max_{\mathbf{b} \in \mathcal{F}_m^{\mathcal{H}_-}} \sum_{t=1}^{T} V(\mathbf{b}; \boldsymbol{\beta}_-^t)}{\max_{(N^*, \widehat{N})} \left\{ \sum_{j \in N^*} \left( W_{Q_j^*}|T_{j,1}| \right) + \sum_{j \in \widehat{N}} \left( V_{j,0}|T_{j,0}| + W_{\widehat{Q}_j}|T_{j,1}| \right) \right\}}. \tag{23}$$

Let $(N_0^*, \widehat{N}_0) = \operatorname{argmax}_{(N^*, \widehat{N})} \left\{ \sum_{j \in N^*} \left( W_{Q_j^*}|T_{j,1}| \right) + \sum_{j \in \widehat{N}} \left( V_{j,0}|T_{j,0}| + W_{\widehat{Q}_j}|T_{j,1}| \right) \right\}$. Then,

$$\max_{\mathbf{b} \in \mathcal{F}_m^{\mathcal{H}_-}} \sum_{t=1}^{T} V(\mathbf{b}; \boldsymbol{\beta}_-^t) = \sum_{t=1}^{T} \sum_{\ell=0}^{1} V_{j,\ell}|T_{j,\ell}|$$

$$= \sum_{j \in N_0^*} \left( V_{j,0}|T_{j,0}| + W_{Q_j^*}|T_{j,1}| \right) + \sum_{j \in \widehat{N}_0} \left( V_{j,0}|T_{j,0}| + W_{Q_j^*}|T_{j,1}| \right). \tag{24}$$

Consider a partition of $[m]$ that differs from the maximizing partition $(N_0^*, \widehat{N}_0)$ by exactly one element, i.e., for any $a \in \widehat{N}_0$, consider the following partition: $(N_0^* \cup \{a\}, \widehat{N}_0 \setminus \{a\})$. By definition,

$$\sum_{j \in N_0^*} \left( W_{Q_j^*}|T_{j,1}| \right) + \sum_{j \in \widehat{N}_0} \left( V_{j,0}|T_{j,0}| + W_{\widehat{Q}_j}|T_{j,1}| \right) \geq \sum_{j \in N_0^* \cup \{a\}} \left( W_{Q_j^*}|T_{j,1}| \right) + \sum_{j \in \widehat{N}_0 \setminus \{a\}} \left( V_{j,0}|T_{j,0}| + W_{\widehat{Q}_j}|T_{j,1}| \right)$$

$$\implies V_{a,0}|T_{a,0}| + W_{\widehat{Q}_a}|T_{a,1}| \geq W_{Q_a^*}|T_{a,1}| . \tag{25}$$

Now, for any $b \in N_0^*$, consider the following partition: $(N_0^* \setminus \{b\}, \widehat{N}_0 \cup \{b\})$. By definition,

$$\sum_{j \in N_0^*} \left( W_{Q_j^*}|T_{j,1}| \right) + \sum_{j \in \widehat{N}_0} \left( V_{j,0}|T_{j,0}| + W_{\widehat{Q}_j}|T_{j,1}| \right) \geq \sum_{j \in N_0^* \setminus \{b\}} \left( W_{Q_j^*}|T_{j,1}| \right) + \sum_{j \in \widehat{N}_0 \cup \{b\}} \left( V_{j,0}|T_{j,0}| + W_{\widehat{Q}_j}|T_{j,1}| \right)$$

$$\implies W_{Q_b^*}|T_{b,1}| - W_{\widehat{Q}_b}|T_{b,1}| \geq V_{b,0}|T_{b,0}| . \tag{26}$$

Plugging in the values,

$$\Lambda_{\mathcal{F}_m^{\mathcal{H}_-}, \mathcal{U}_m^\star}(\mathcal{H}_-) \overset{(24)}{\leq} \frac{\sum_{j \in N_0^*} \left( V_{j,0}|T_{j,0}| + W_{Q_j^*}|T_{j,1}| \right) + \sum_{j \in \widehat{N}_0} \left( V_{j,0}|T_{j,0}| + W_{Q_j^*}|T_{j,1}| \right)}{\sum_{j \in N_0^*} \left( W_{Q_j^*}|T_{j,1}| \right) + \sum_{j \in \widehat{N}_0} \left( V_{j,0}|T_{j,0}| + W_{\widehat{Q}_j}|T_{j,1}| \right)}$$

$$\overset{(25)}{\leq} \frac{\sum_{j \in N_0^*} \left( V_{j,0}|T_{j,0}| + W_{Q_j^*}|T_{j,1}| \right) + \sum_{j \in \widehat{N}_0} \left( 2V_{j,0}|T_{j,0}| + W_{\widehat{Q}_j}|T_{j,1}| \right)}{\sum_{j \in N_0^*} \left( W_{Q_j^*}|T_{j,1}| \right) + \sum_{j \in \widehat{N}_0} \left( V_{j,0}|T_{j,0}| + W_{\widehat{Q}_j}|T_{j,1}| \right)}$$

$$\overset{(26)}{\leq} \frac{\sum_{j \in N_0^*} \left( 2W_{Q_j^*}|T_{j,1}| - W_{\widehat{Q}_j}|T_{j,1}| \right) + \sum_{j \in \widehat{N}_0} \left( 2V_{j,0}|T_{j,0}| + W_{\widehat{Q}_j}|T_{j,1}| \right)}{\sum_{j \in N_0^*} \left( W_{Q_j^*}|T_{j,1}| \right) + \sum_{j \in \widehat{N}_0} \left( V_{j,0}|T_{j,0}| + W_{\widehat{Q}_j}|T_{j,1}| \right)}$$

$$= \frac{2\left\{ \sum_{j \in N_0^*} W_{Q_j^*}|T_{j,1}| + \sum_{j \in \widehat{N}_0} \left( V_{j,0}|T_{j,0}| + W_{\widehat{Q}_j}|T_{j,1}| \right) \right\} - \sum_{j=1}^m W_{\widehat{Q}_j}|T_{j,1}|}{\sum_{j \in N_0^*} \left( W_{Q_j^*}|T_{j,1}| \right) + \sum_{j \in \widehat{N}_0} \left( V_{j,0}|T_{j,0}| + W_{\widehat{Q}_j}|T_{j,1}| \right)}$$

$$= 2 - \theta_{\mathcal{H}_-} ,$$

where

$$0 < \theta_{\mathcal{H}_-} = \frac{\sum_{j=1}^m W_{\widehat{Q}_j}|T_{j,1}|}{\sum_{j \in N_0^*} \left( W_{Q_j^*}|T_{j,1}| \right) + \sum_{j \in \widehat{N}_0} \left( V_{j,0}|T_{j,0}| + W_{\widehat{Q}_j}|T_{j,1}| \right)}$$

$$\overset{(25)}{\leq} \frac{\sum_{j=1}^m W_{\widehat{Q}_j}|T_{j,1}|}{\sum_{j \in N_0^*} \left( W_{Q_j^*}|T_{j,1}| \right) + \sum_{j \in \widehat{N}_0} \left( W_{Q_j^*}|T_{j,1}| \right)}$$

$$= \frac{\sum_{j=1}^m W_{\widehat{Q}_j}|T_{j,1}|}{\sum_{j=1}^m W_{Q_j^*}|T_{j,1}|} \overset{(a)}{\leq} \max_{j \in [m]} \frac{W_{\widehat{Q}_j}}{W_{Q_j^*}} \leq 1 ,$$

and $(a)$ follows from Fact 1. Finally,

$$\Lambda_{\mathcal{F}_m^{\mathcal{H}_-}, \mathcal{U}_m^\star} = \max_{\mathcal{H}_-} \Lambda_{\mathcal{F}_m^{\mathcal{H}_-}, \mathcal{U}_m^\star}(\mathcal{H}_-) \leq 2 - \min_{\mathcal{H}_-} \theta_{\mathcal{H}_-} =: 2 - \theta, \quad \text{where} \quad \theta = \min_{\mathcal{H}_-} \theta_{\mathcal{H}_-} \in (0, 1] .$$

**Fact 1** (Williamson & Shmoys (2011, pp. 25))**.** For positive numbers $a_1, \ldots, a_m$ and $b_1, \ldots, b_m$,

$$\frac{\sum_{j=1}^m a_j}{\sum_{j=1}^m b_j} \leq \max_{j \in [m]} \frac{a_j}{b_j} .$$

### C.2. Proof of Lemma C.1

We state and prove a stronger version of the result in Lemma C.1. Formally,

**Lemma C.3.** *For any $m \in \mathbb{N}$ and competing bid $\boldsymbol{\beta}_-$, let $\mathbf{b} = \langle (b_1, q_1), \ldots, (b_m, q_m) \rangle$ be a feasible (not necessarily safe) $m$-uniform strategy for $\boldsymbol{\beta}_-$. Then,*

$$V(\mathbf{b}; \boldsymbol{\beta}_-) = \max_{\ell \in [m]} V((b_\ell, Q_\ell); \boldsymbol{\beta}_-),$$

*where we recall that $Q_\ell = \sum_{j=1}^\ell q_j, \forall \ell \in [m]$.*

Lemma C.3 states a similar result as Lemma C.1, except a key difference that the bidding strategies are not necessarily safe.

C.2.1. PROOF OF LEMMA C.3

We prove the lemma via induction on $m$. The base case is $m = 1$ for which the result is trivially true. Now assume that the result holds for any $m$-uniform bidding strategy for any competing bid $\boldsymbol{\beta}_-$. We now show that the result holds for any $(m + 1)$-uniform bidding strategy which is feasible for $\boldsymbol{\beta}_-$.

Consider any $m + 1$-uniform bidding strategy $\mathbf{b} = \langle (b_1, q_1), \ldots, (b_m, q_m), (b_{m+1}, q_{m+1}) \rangle$ feasible for $\boldsymbol{\beta}_-$. Then,

**Claim C.4.** The bid-quantity pair $(b_{m+1}, Q_{m+1})$ is feasible for $\boldsymbol{\beta}_-$ and $V(\mathbf{b}; \boldsymbol{\beta}_-) \geq V((b_{m+1}, Q_{m+1}); \boldsymbol{\beta}_-)$.

*Proof.* By assumption, $\mathbf{b}$ is feasible, the total demand of $\mathbf{b}$ and $(b_{m+1}, Q_{m+1})$ are equal and $b_{(j)} \geq b_{m+1}, \forall j \in [Q_{m+1}]$, where $b_{(j)}$ denotes the bid value in $j$th position in the sorted bid vector. So, by Lemma A.2, $(b_{m+1}, Q_{m+1})$ is feasible and $V(\mathbf{b}; \boldsymbol{\beta}_-) \geq V((b_{m+1}, Q_{m+1}), \boldsymbol{\beta}_-)$. $\square$

Suppose that by bidding $\mathbf{b}$, the bidder is allocated $r$ units. There are two cases: (a) $r \leq Q_m$ and (b) $r > Q_m$.

**Case I.** $r \leq Q_m$. In this case, we have

$$V(\mathbf{b}; \boldsymbol{\beta}_-) = V(\mathbf{b}[1:m]; \boldsymbol{\beta}_-). \tag{27}$$

Hence, by Claim C.4 and Eq. (27),

$$V(\mathbf{b}; \boldsymbol{\beta}_-) = \max \left\{ V(\mathbf{b}[1:m]; \boldsymbol{\beta}_-), V((b_{m+1}, Q_{m+1}); \boldsymbol{\beta}_-) \right\}. \tag{28}$$

**Case II.** $r > Q_m$. As $r > Q_m$, $b_{m+1}$ is the least winning bid which implies $b_{m+1} \geq \boldsymbol{\beta}_-^{-(r)}$, where we recall that $\boldsymbol{\beta}_-^{-(r)}$ is the $r^{th}$ smallest competing bid in $\boldsymbol{\beta}_-$. So, $(b_{m+1}, Q_{m+1})$ is allocated at least $r$ units which implies $V((b_{m+1}, Q_{m+1}), \boldsymbol{\beta}_-) \geq V(\mathbf{b}; \boldsymbol{\beta}_-)$. By Claim C.4, we also have $V(\mathbf{b}; \boldsymbol{\beta}_-) \geq V((b_{m+1}, Q_{m+1}), \boldsymbol{\beta}_-)$. Hence,

$$V(\mathbf{b}; \boldsymbol{\beta}_-) = V((b_{m+1}, Q_{m+1}); \boldsymbol{\beta}_-). \tag{29}$$

As $r > Q_m$, $(b_{m+1}, Q_{m+1})$ is allocated at least $Q_m + 1$ units, whereas $\mathbf{b}[1:m]$ has demand for (hence, can be allocated) at most $Q_m$ units. So,

$$V(\mathbf{b}; \boldsymbol{\beta}_-) \geq V(\mathbf{b}[1:m]; \boldsymbol{\beta}_-)$$
$$\implies V(\mathbf{b}; \boldsymbol{\beta}_-) = \max \left\{ V(\mathbf{b}[1:m]; \boldsymbol{\beta}_-), V((b_{m+1}, Q_{m+1}); \boldsymbol{\beta}_-) \right\}. \tag{30}$$

For both **Case I** and **Case II**, we get the same result (cf. (28) and (30)). Hence,

$$V(\mathbf{b}; \boldsymbol{\beta}_-) = \max \left\{ V(\mathbf{b}[1:m]; \boldsymbol{\beta}_-), V((b_{m+1}, Q_{m+1}); \boldsymbol{\beta}_-) \right\}$$
$$\overset{(a)}{=} \max \left\{ \max_{\ell \in [m]} V((b_\ell, Q_\ell); \boldsymbol{\beta}_-), V((b_{m+1}, Q_{m+1}); \boldsymbol{\beta}_-) \right\}$$
$$= \max_{\ell \in [m+1]} V((b_\ell, Q_\ell); \boldsymbol{\beta}_-).$$

Here, $(a)$ holds as $\mathbf{b}[1:m]$ is feasible for $\boldsymbol{\beta}_-$ allowing us to apply the induction hypothesis for $m$.

## C.3. Proof of Lemma C.2

To prove this result, we use the following key lemma that measures the value obtained by $\mathbf{b}_m^{\mathsf{OPT}}(\mathcal{H}_-)$ in terms of 1-uniform safe bidding strategies.

**Lemma C.5.** *Let* $\mathbf{b}_m^{\mathsf{OPT}}(\mathcal{H}_-) = \langle (b_1^*, q_1^*), \ldots, (b_m^*, q_m^*) \rangle$. *Suppose* $Q_j^* = \sum_{\ell \leq j} q_\ell^*, \forall j \in [m]$ *and* $\mathbf{b}_m^{\mathsf{OPT}}(\mathcal{H}_-)$ *is allocated* $r_t^*$ *units in any round* $t \in [T]$. *Then,*

*1. For any* $t \in [T]$ *and* $q \leq r_t^*$, $(w_q, q)$ *gets* exactly $q$ *units.*

*2. For any $j \in [m]$, let $T_j \subseteq [T]$ be the rounds in which $b_j^*$ is the least winning bid, i.e.,*

$$T_j = \left\{ t \in [T] : Q_{j-1}^* < r_t^* \le Q_j^* \right\}.$$

*Suppose that $\exists t \in T_j$ such that $r_t^* < Q_j^*$. Then in any round $t' \in T_j$ in which $\mathbf{b}_m^{\mathsf{OPT}}(\mathcal{H}_-)$ is allocated at most $r_t^*$ units, i.e., $r_{t'}^* \le r_t^*$, $(w_{r_t^*}, r_t^*)$ is allocated at least $r_{t'}^*$ units.*

With this lemma, we are ready to prove Lemma C.2.

**Case I:** $Q_j = \widehat{Q}_j < Q_j^*$. For any $t \in T_{j,1}$, invoking Lemma C.5 (1) with $q = \widehat{Q}_j$, we conclude that $(w_{\widehat{Q}_j}, \widehat{Q}_j)$ is allocated exactly $\widehat{Q}_j$ units. Summing over all rounds in $T_{j,1}$,

$$\sum_{t \in T_{j,1}} V((w_{Q_j}, Q_j); \boldsymbol{\beta}_-^t) = W_{\widehat{Q}_j} |T_{j,1}|. \tag{31}$$

As $\widehat{Q}_j < Q_j^*$ and $\widehat{Q}_j \ge r_s^*$ for any $s \in T_{j,0}$. So, for any $s \in T_{j,0}$, invoking Lemma C.5 (2) with $r_t^* = \widehat{Q}_j$, we conclude that $(w_{\widehat{Q}_j}, \widehat{Q}_j)$ is allocated at least $r_s^*$ units. Hence, summing over all rounds, $(w_{\widehat{Q}_j}, \widehat{Q}_j)$ gets at least the value obtained by $\mathbf{b}_m^{\mathsf{OPT}}(\mathcal{H}_-)$ over the rounds in $T_{j,0}$. So,

$$\sum_{t \in T_{j,0}} V((w_{Q_j}, Q_j); \boldsymbol{\beta}_-^t) \ge \sum_{t \in T_{j,0}} V\left(\mathbf{b}_m^{\mathsf{OPT}}(\mathcal{H}_-), \boldsymbol{\beta}_-^t\right) = V_{j,0} |T_{j,0}|. \tag{32}$$

Combining Eq. (31) and (32), for $Q_j = \widehat{Q}_j$,

$$\sum_{t \in T_j} V((w_{Q_j}, Q_j); \boldsymbol{\beta}_-^t) \ge V_{j,0} |T_{j,0}| + W_{\widehat{Q}_j} |T_{j,1}|. \tag{33}$$

**Case II:** $Q_j = Q_j^*$. So, for any $t \in T_{j,1}$, using Lemma C.5 (1) with $q = Q_j^*$, we get that $(w_{Q_j^*}, Q_j^*)$ is allocated exactly $Q_j^*$ units, which is the same as the allocation for $\mathbf{b}_m^{\mathsf{OPT}}(\mathcal{H}_-)$. Summing over all rounds in $T_{j,1}$,

$$\sum_{t \in T_{j,1}} V((w_{Q_j}, Q_j); \boldsymbol{\beta}_-^t) = W_{Q_j^*} |T_{j,1}|. \tag{34}$$

For the rounds in $T_{j,0}$, trivially,

$$\sum_{t \in T_{j,0}} V((w_{Q_j}, Q_j); \boldsymbol{\beta}_-^t) \ge 0. \tag{35}$$

Combining Eq. (34) and (35), for $Q_j = Q_j^*$,

$$\sum_{t \in T_j} V((w_{Q_j}, Q_j); \boldsymbol{\beta}_-^t) \ge W_{Q_j^*} |T_{j,1}|. \tag{36}$$

### C.3.1. PROOF OF LEMMA C.5

(1) First we show that for any $t \in [T]$ and $q \le r_t^*$, the 1-uniform bid $(w_q, q)$ is allocated *exactly* $q$ units.

As $(w_q, q) \in \mathcal{U}_1^\star$, it is a safe strategy. By assumption, $\mathbf{b}_m^{\mathsf{OPT}}(\mathcal{H}_-)$ is allocated $r_t^*$ units in round $t$. Let $\boldsymbol{\beta}^t = (\mathbf{b}_m^{\mathsf{OPT}}(\mathcal{H}_-), \boldsymbol{\beta}_-^t)$. Recall that, $\boldsymbol{\beta}_{-,t}^{-(j)}$ is the $j^{th}$ smallest winning bid in the absence of bids from bidder $i$ for round $t$. If $r_t^* = 0$, the result is vacuously true. Suppose $r_t^* > 0$, then

$$\boldsymbol{\beta}_{-,t}^{-(q)} \le \boldsymbol{\beta}_{-,t}^{-(r_t^*)} \le p(\boldsymbol{\beta}^t) \le w_{r_t^*} \le w_q,$$

where the first inequality holds by definition of $\boldsymbol{\beta}_{-,t}^{-(j)}$ and our assumption that $q \le r_t^*$. For the second inequality, suppose $Q_{\ell-1}^* < r_t^* \le Q_\ell^*$ for some $\ell \in [m]$ which by definition implies that $b_\ell^*$ is the least winning bid. By Lemma A.1,

1. If $p(\boldsymbol{\beta}^t) = b_\ell^*$, we have $p(\boldsymbol{\beta}^t) \geq \boldsymbol{\beta}_{-,t}^{-(r_t^*)}$ as $b_\ell^*$ is the least winning bid.

2. If $p(\boldsymbol{\beta}^t) = \boldsymbol{\beta}_{-,t}^{-(Q_\ell^*+1)}$, we have $r_t^* = Q_\ell^*$ and by definition of $\boldsymbol{\beta}_{-,t}^{-(j)}$, $p(\boldsymbol{\beta}^t) = \boldsymbol{\beta}_{-,t}^{-(Q_\ell^*+1)} \geq \boldsymbol{\beta}_{-,t}^{-(r_t^*)}$.

The third inequality is true as RoI constraint is satisfied by $\mathbf{b}_m^{\mathsf{OPT}}(\mathcal{H}_-)$ for round $t$, and the fourth is true as $w_j$ is a non-decreasing function of $j$ and $q \leq r_t^*$. From the first and last expressions, $\boldsymbol{\beta}_{-,t}^{-(q)} \leq w_q$ which implies that $(w_q, q)$ is allocated at least $q$ units in round $t$. Moreover, $(w_q, q)$ can be allocated at most $q$ units. Hence, the 1-uniform bid $(w_q, q)$ is allocated *exactly* $q$ units.

(2) For any $j \in [m]$, let $T_j \subseteq [T]$, defined in Eq. (15), be the rounds in which $b_j^*$ is the least winning bid. Suppose that $\exists t \in T_j$ such that $r_t^* < Q_j^*$. We show that in any round $t' \in T_j$ in which $\mathbf{b}_m^{\mathsf{OPT}}(\mathcal{H}_-)$ is allocated at most $r_t^*$ units, i.e., $r_{t'}^* \leq r_t^*$, the 1-uniform strategy $(w_{r_t^*}, r_t^*)$ is allocated *at least* $r_{t'}^*$ units.

Observe that $(w_{r_t^*}, r_t^*) \in \mathcal{U}_1^\star$, so it is a safe strategy. If $r_t^* = 0$, the result is trivially true. Hence, suppose $r_t^* > 0$ and consider the set of rounds in $T_j$ for any $j \in [m]$. As $r_t^* < Q_j^*$, by Lemma A.1, $p(\boldsymbol{\beta}^t) = b_j^*$. So, for any $t' \in T_j$ such that $r_{t'}^* \leq r_t^*$,

$$\boldsymbol{\beta}_{-,t'}^{-(r_{t'}^*)} \leq b_j^* \leq w_{r_t^*} \,.$$

Here, the first inequality holds as $b_j^*$ is the least winning bid for round $t' \in T_j$ and the second one holds as the RoI constraint is true for $\mathbf{b}_m^{\mathsf{OPT}}(\mathcal{H}_-)$ for round $t$. Hence, $(w_{r_t^*}, r_t^*)$ is allocated *at least* $r_{t'}^*$ units in round $t'$.

### C.4. Proof of Theorem 5.4

We state and prove a stronger result which recovers the upper bound in Theorem 5.4 as a corollary and is also crucial to prove the upper bound in Theorem 5.5. Let $\mathcal{F}_m^{\mathcal{H}_-}$ be the class of bidding strategies with at most $m$ bid-quantity pairs that are feasible for the bid history $\mathcal{H}_- = [\boldsymbol{\beta}_-^t]_{t \in [T]}$. Then,

**Theorem C.6.** *For any $m, m' \in \mathbb{N}$ such that $m' \geq m$, $\Lambda_{\mathcal{F}_{m'}^{\mathcal{H}_-}, \mathcal{F}_m^{\mathcal{H}_-}} \leq \frac{m'}{m}$.*

*Proof.* For $\mathcal{H}_- = [\boldsymbol{\beta}_-^t]_{t \in [T]}$, we define

$$\Lambda_{\mathcal{F}_{m'}^{\mathcal{H}_-}, \mathcal{F}_m^{\mathcal{H}_-}}(\mathcal{H}_-) = \frac{\max_{\mathbf{b}' \in \mathcal{F}_{m'}^{\mathcal{H}_-}} \sum_{t=1}^T V(\mathbf{b}'; \boldsymbol{\beta}_-^t)}{\max_{\mathbf{b} \in \mathcal{F}_m^{\mathcal{H}_-}} \sum_{t=1}^T V(\mathbf{b}; \boldsymbol{\beta}_-^t)} \,.$$

Then, maximizing over all bid histories, we get the desired result.

Let $\mathbf{b}_{m'}^{\mathsf{OPT}}(\mathcal{H}_-) = \operatorname{argmax}_{\mathbf{b} \in \mathcal{F}_{m'}^{\mathcal{H}_-}} \sum_{t=1}^T V(\mathbf{b}; \boldsymbol{\beta}_-^t)$. If $\mathbf{b}_{m'}^{\mathsf{OPT}}(\mathcal{H}_-)$ is a $k$-uniform bidding strategy where $k \leq m \leq m'$, by definition, $\mathbf{b}_{m'}^{\mathsf{OPT}}(\mathcal{H}_-) \in \mathcal{F}_m^{\mathcal{H}_-}$ implying $\Lambda_{\mathcal{F}_{m'}^{\mathcal{H}_-}, \mathcal{F}_m^{\mathcal{H}_-}}(\mathcal{H}_-) = 1 \leq \frac{m'}{m}$.

Without loss of generality, let the optimal bidding strategy in $\mathcal{F}_{m'}^{\mathcal{H}_-}$ be $\mathbf{b}_{m'}^{\mathsf{OPT}}(\mathcal{H}_-) = \langle (b_1^*, q_1^*), \dots, (b_{m'}^*, q_{m'}^*) \rangle$ and $Q_\ell^* = \sum_{j \leq \ell} q_j^*$ for all $\ell \in [m']$. [11] Let $r_t^*$ be the number of units allocated to $\mathbf{b}_{m'}^{\mathsf{OPT}}(\mathcal{H}_-)$ in round $t$. Recall that $T_j$ is the set of rounds when $b_j^*$ is the least winning bid when $\mathbf{b}_{m'}^{\mathsf{OPT}}(\mathcal{H}_-)$ is submitted (see Eq. (15)). So,

$$\sum_{t=1}^T V(\mathbf{b}_{m'}^{\mathsf{OPT}}(\mathcal{H}_-); \boldsymbol{\beta}_-^t) = \sum_{j=1}^{m'} \sum_{t \in T_j} V(\mathbf{b}_{m'}^{\mathsf{OPT}}(\mathcal{H}_-); \boldsymbol{\beta}_-^t) \,.$$

Observe that, for any $t \in T_j$,

$$V(\mathbf{b}_{m'}^{\mathsf{OPT}}(\mathcal{H}_-); \boldsymbol{\beta}_-^t) = V(\mathbf{b}_{m'}^{\mathsf{OPT}}(\mathcal{H}_-)[1:j]; \boldsymbol{\beta}_-^t) = \max_{\ell \in [j]} V((b_\ell^*, Q_\ell^*); \boldsymbol{\beta}_-^t) \,,$$

---

[11] A similar analysis also follows if $\mathbf{b}_{m'}^{\mathsf{OPT}}(\mathcal{H}_-)$ is a $k$-uniform strategy where $m \leq k \leq m'$.

where recall that $\mathbf{b}[1:j]$ represents the first $j$ bid-quantity pairs and the last equality holds due to Lemma C.3. For any $t \in T_j$, $r_t^* > Q_{j-1}^*$. So, $\forall \ell < j$, $V(\mathbf{b}_{m'}^{\mathsf{OPT}}(\mathcal{H}_-); \boldsymbol{\beta}_-^t) > V((b_\ell^*, Q_\ell^*); \boldsymbol{\beta}_-^t)$ as the demand of the strategy $(b_\ell^*, Q_\ell^*)$ is strictly less than the number of units obtained by $\mathbf{b}_{m'}^{\mathsf{OPT}}(\mathcal{H}_-)$ in that round. Hence, for any $t \in T_j$,

$$V(\mathbf{b}_{m'}^{\mathsf{OPT}}(\mathcal{H}_-); \boldsymbol{\beta}_-^t) = V((b_j^*, Q_j^*); \boldsymbol{\beta}_-^t) \implies \sum_{t=1}^{T} V(\mathbf{b}_{m'}^{\mathsf{OPT}}(\mathcal{H}_-); \boldsymbol{\beta}_-^t) = \sum_{j=1}^{m'} \sum_{t \in T_j} V((b_j^*, Q_j^*); \boldsymbol{\beta}_-^t). \tag{37}$$

Let $\mathcal{G}_m^{\mathcal{H}_-} \subseteq \mathcal{F}_m^{\mathcal{H}_-}$ be the class of feasible strategies for $\mathcal{H}_-$ that contain *exactly* $m$ bid-quantity pairs. Then,

$$\Lambda_{\mathcal{F}_{m'}^{\mathcal{H}_-}, \mathcal{F}_m^{\mathcal{H}_-}}(\mathcal{H}_-) \stackrel{(37)}{=} \frac{\sum_{j=1}^{m'} \sum_{t \in T_j} V((b_j^*, Q_j^*); \boldsymbol{\beta}_-^t)}{\max_{\mathbf{b} \in \mathcal{F}_m^{\mathcal{H}_-}} \sum_{t=1}^{T} V(\mathbf{b}; \boldsymbol{\beta}_-^t)} \leq \frac{\sum_{j=1}^{m'} \sum_{t \in T_j} V((b_j^*, Q_j^*); \boldsymbol{\beta}_-^t)}{\max_{\mathbf{b} \in \mathcal{G}_m^{\mathcal{H}_-}} \sum_{t=1}^{T} V(\mathbf{b}; \boldsymbol{\beta}_-^t)}. \tag{38}$$

**Constructing the class of bidding strategies, $\widetilde{\mathcal{G}}_m^{\mathcal{H}_-}$.** We now construct a class of $m$-uniform bidding strategies, $\widetilde{\mathcal{G}}_m^{\mathcal{H}_-}$ using $\mathbf{b}_{m'}^{\mathsf{OPT}}(\mathcal{H}_-)$ as a 'parent' bidding strategy, in Algorithm 3.

---

**Algorithm 3** Constructing $\widetilde{\mathcal{G}}_m^{\mathcal{H}_-}$

---

**Require:** $\mathbf{b}_{m'}^{\mathsf{OPT}}(\mathcal{H}_-) = \langle (b_1^*, q_1^*), \ldots, (b_{m'}^*, q_{m'}^*) \rangle$. The collection, $\mathcal{C}$, of all subsets $\mathcal{S} \subseteq [m']$ such that $|\mathcal{S}| = m$. Initialize $\widetilde{\mathcal{G}}_m^{\mathcal{H}_-} = \emptyset$.
1: **for** $\mathcal{S} \in \mathcal{C}$ **do**
2:   Suppose $\mathcal{S} = \{k_1, \ldots, k_m\}$ such that $k_1 < k_2 < \cdots < k_m$.
3:   Construct a $m$-uniform bidding strategy $\mathbf{b} = \langle (b_1, q_1), \ldots, (b_m, q_m) \rangle$ where

$$b_j = b_{k_j}^* \quad \text{and} \quad q_j = Q_{k_j}^* - Q_{k_{j-1}}^*, \forall j \in [m]. \tag{39}$$

4:   $\widetilde{\mathcal{G}}_m^{\mathcal{H}_-} \leftarrow \widetilde{\mathcal{G}}_m^{\mathcal{H}_-} \cup \{\mathbf{b}\}$
5: **end for**
   **Return:** The class of bidding strategies, $\widetilde{\mathcal{G}}_m^{\mathcal{H}_-}$.

---

For example: suppose $m' = 4$ and $\mathbf{b}_{m'}^{\mathsf{OPT}}(\mathcal{H}_-) = \langle (w_3, 3), (w_7, 4), (w_8, 1), (w_{10}, 2) \rangle$. Let $m = 2$ and $\mathcal{S} = \{2, 4\}$. Then, $\mathbf{b} = \langle (w_7, 7), (w_{10}, 3) \rangle$.

**Lemma C.7.** *Let $b_{(j)}^*$ be the $j^{th}$ entry when $\mathbf{b}_{m'}^{\mathsf{OPT}}(\mathcal{H}_-)$ is expressed as a bid vector. Similarly, for any $\mathbf{b} \in \widetilde{\mathcal{G}}_m^{\mathcal{H}_-}$, let $b_{(j)}$ be the $j^{th}$ entry when $\mathbf{b}$ is expressed as a bid vector. Then, for any $j \in [Q_{k_m}]$, $b_{(j)} \leq b_{(j)}^*$. Thus, by Lemma A.2, $\mathbf{b}$ is also feasible for $\mathcal{H}_-$ and as $\mathbf{b}$ was chosen arbitrarily, we conclude that $\widetilde{\mathcal{G}}_m^{\mathcal{H}_-} \subseteq \mathcal{G}_m^{\mathcal{H}_-}$.*

*Proof.* Let $\mathcal{S}$ contain $k_1 < \cdots < k_m$ and $\mathbf{b}$ be constructed per Algorithm 3. Then, for any $j \in [m]$ and any entry $\ell \in (Q_{k_{j-1}}^*, Q_{k_j}^*]$,

$$b_{(\ell)}^* \geq b_{(Q_{k_j}^*)}^* = b_{k_j}^* = b_{(\ell)},$$

where the first inequality follows because entries in bid vector are in non increasing order, the first equality follows as $b_{(Q_r)}^* = b_r^*$ for all $r \in [m']$ and second equality follows from Eq. (39). As $j$ and $\ell$ were picked arbitrarily, we get that for all $j \in [Q_{k_m}]$, $b_{(j)} \leq b_{(j)}^*$. $\qquad\square$

By Lemma C.7,

$$\Lambda_{\mathcal{F}_{m'}^{\mathcal{H}_-}, \mathcal{F}_m^{\mathcal{H}_-}}(\mathcal{H}_-) \stackrel{(38)}{\leq} \frac{\sum_{j=1}^{m'} \sum_{t \in T_j} V((b_j^*, Q_j^*); \boldsymbol{\beta}_-^t)}{\max_{\mathbf{b} \in \mathcal{G}_m^{\mathcal{H}_-}} \sum_{t=1}^{T} V(\mathbf{b}; \boldsymbol{\beta}_-^t)} \leq \frac{\sum_{j=1}^{m'} \sum_{t \in T_j} V((b_j^*, Q_j^*); \boldsymbol{\beta}_-^t)}{\max_{\mathbf{b} \in \widetilde{\mathcal{G}}_m^{\mathcal{H}_-}} \sum_{t=1}^{T} V(\mathbf{b}; \boldsymbol{\beta}_-^t)}. \tag{40}$$

Consider any $\mathbf{b} \in \widetilde{\mathcal{G}}_m^{\mathcal{H}_-}$ obtained from the set $\mathcal{S} = \{k_1, \ldots, k_m\} \subseteq [m']$. As $\mathbf{b}$ is feasible for $\mathcal{H}_-$, invoking Lemma C.3 for any round $t \in [T]$, we get:

$$V(\mathbf{b}; \boldsymbol{\beta}_-^t) = \max_{\ell \in [m]} V((b_{k_\ell}^*, Q_{k_\ell}^*); \boldsymbol{\beta}_-^t) = \max_{\ell \in \mathcal{S}} V((b_\ell^*, Q_\ell^*); \boldsymbol{\beta}_-^t), \tag{41}$$

because for any $\ell \in [m]$, $b_\ell = b^*_{k_\ell}$ and $Q_\ell = \sum_{j=1}^{j} q_j = \sum_{j=1}^{j} Q^*_{k_j} - Q^*_{k_{j-1}} = Q^*_{k_\ell}$.

Recall that $T_j$s, the set of rounds in which $b^*_j$ is the least winning bid, form a partition of $[T]$. So,

$$\sum_{t=1}^{T} V(\mathbf{b}; \boldsymbol{\beta}^t_-) = \sum_{t=1}^{m'} \sum_{t \in T_j} V(\mathbf{b}; \boldsymbol{\beta}^t_-) = \sum_{j \in \mathcal{S}} \sum_{t \in T_j} V(\mathbf{b}; \boldsymbol{\beta}^t_-) + \sum_{j \in [m'] \setminus \mathcal{S}} \sum_{t \in T_j} V(\mathbf{b}; \boldsymbol{\beta}^t_-) \geq \sum_{j \in \mathcal{S}} \sum_{t \in T_j} V(\mathbf{b}; \boldsymbol{\beta}^t_-) \tag{42}$$

For any $j \in \mathcal{S}$, consider any round $t \in T_j$:

$$V(\mathbf{b}; \boldsymbol{\beta}^t_-) \overset{(41)}{=} \max_{\ell \in \mathcal{S}} V((b^*_\ell, Q^*_\ell); \boldsymbol{\beta}^t_-) \geq V((b^*_j, Q^*_j); \boldsymbol{\beta}^t_-) \implies \sum_{t \in T_j} V(\mathbf{b}; \boldsymbol{\beta}^t_-) \geq \sum_{t \in T_j} V((b^*_j, Q^*_j); \boldsymbol{\beta}^t_-) \tag{43}$$

which implies,

$$\sum_{t=1}^{T} V(\mathbf{b}; \boldsymbol{\beta}^t_-) \overset{(42)}{\geq} \sum_{j \in \mathcal{S}} \sum_{t \in T_j} V(\mathbf{b}; \boldsymbol{\beta}^t_-) \overset{(43)}{\geq} \sum_{j \in \mathcal{S}} \sum_{t \in T_j} V((b^*_j, Q^*_j); \boldsymbol{\beta}^t_-) . \tag{44}$$

Note that a bidding strategy $\mathbf{b} \in \widetilde{\mathcal{G}}^{\mathcal{H}_-}_m$ is uniquely determined by the set $\mathcal{S} \subseteq [m']$ that generates $\mathbf{b}$. Hence,

$$\max_{\mathbf{b} \in \widetilde{\mathcal{G}}^{\mathcal{H}_-}_m} \sum_{t=1}^{T} V(\mathbf{b}; \boldsymbol{\beta}^t_-) = \max_{\mathcal{S} \subseteq [m']: |\mathcal{S}| = m} \sum_{t=1}^{T} V(\mathbf{b}; \boldsymbol{\beta}^t_-)$$

$$\overset{(44)}{\geq} \max_{\mathcal{S} \subseteq [m']: |\mathcal{S}| = m} \sum_{j \in \mathcal{S}} \sum_{t \in T_j} V((b^*_j, Q^*_j); \boldsymbol{\beta}^t_-)$$

$$\geq \frac{m}{m'} \sum_{j=1}^{m'} \sum_{t \in T_j} V((b^*_j, Q^*_j); \boldsymbol{\beta}^t_-) . \tag{45}$$

Hence,

$$\Lambda_{\mathcal{G}^{\mathcal{H}_-}_{m'}, \mathcal{G}^{\mathcal{H}_-}_m}(\mathcal{H}_-) \leq \frac{\sum_{j=1}^{m'} \sum_{t \in T_j} V((b^*_j, Q^*_j); \boldsymbol{\beta}^t_-)}{\max_{\mathbf{b} \in \widetilde{\mathcal{G}}^{\mathcal{H}_-}_m} \sum_{t=1}^{T} V(\mathbf{b}; \boldsymbol{\beta}^t_-)} \overset{(45)}{\leq} \frac{m'}{m} .$$

Maximizing over all bid histories, we get that $\Lambda_{\mathcal{G}^{\mathcal{H}_-}_{m'}, \mathcal{G}^{\mathcal{H}_-}_m} \leq \frac{m'}{m}$.

$\square$

Now, we prove that the result also holds for safe bidding strategies. Formally, we show that for $m, m' \in \mathbb{N}$ and $m' \geq m$, $\Lambda_{\mathcal{U}^\star_{m'}, \mathcal{U}^\star_m} \leq \frac{m'}{m}$. To this end, we consider the quantity $\Lambda_{\mathcal{U}^\star_{m'}, \mathcal{U}^\star_m}(\mathcal{H}_-)$ analogous to $\Lambda_{\mathcal{G}^{\mathcal{H}_-}_{m'}, \mathcal{G}^{\mathcal{H}_-}_m}(\mathcal{H}_-)$, upper bound it and maximize over all bid histories to get the desired result.

For any bid history, $\mathcal{H}_- = [\boldsymbol{\beta}^t_-]_{t \in [T]}$, let $\mathbf{b}^{\mathsf{SAFE}}_{m'}(\mathcal{H}_-) := \mathrm{argmax}_{\mathbf{b}' \in \mathcal{U}^\star_{m'}} \sum_{t=1}^{T} V(\mathbf{b}'; \boldsymbol{\beta}^t_-)$. Without loss of generality, let $\mathbf{b}^{\mathsf{SAFE}}_{m'}(\mathcal{H}_-) = \langle (w_{Q^*_1}, q^*_1), \ldots, (w_{Q^*_{m'}}, q^*_{m'}) \rangle$. The proof to bound $\Lambda_{\mathcal{U}^\star_{m'}, \mathcal{U}^\star_m}(\mathcal{H}_-)$ is similar to that of Theorem C.6 till Eq. (38), where we have

$$\Lambda_{\mathcal{U}^\star_{m'}, \mathcal{U}^\star_m}(\mathcal{H}_-) \leq \frac{\sum_{j=1}^{m'} \sum_{t \in T_j} V((w^*_{Q_j}, Q^*_j); \boldsymbol{\beta}^t_-)}{\max_{\mathbf{b} \in \mathcal{S}_m} \sum_{t=1}^{T} V(\mathbf{b}; \boldsymbol{\beta}^t_-)} .$$

Recall that $\mathcal{S}_m$ is the class of $m$-uniform safe bidding strategies. Suppose $\mathcal{S}_m(\mathcal{H}_-)$ be the class that is constructed analogous to the class $\widetilde{\mathcal{G}}^{\mathcal{H}_-}_m$ obtained in Algorithm 3. We now show that $\mathcal{S}_m(\mathcal{H}_-) \subseteq \mathcal{S}_m$.

**Lemma C.8.** *For any subset $\mathcal{S} \subseteq [m']$ such that $|\mathcal{S}| = m$, suppose $\mathbf{b}$ is the bidding strategy obtained from $\mathcal{S}$. Then, $\mathbf{b}$ is a safe bidding strategy. As $\mathcal{S}$ was chosen arbitrarily, $\mathcal{S}_m(\mathcal{H}_-) \subseteq \mathcal{S}_m$.*

*Proof.* Let $\mathcal{S}$ contain $k_1 < \cdots < k_m$ and $\mathbf{b}$ be constructed per Algorithm 3. Then, for any $j \in [m]$,

$$b_j = b^*_{k_j} = w_{Q^*_{k_j}}, \quad \text{and} \quad Q_j = Q^*_{k_j}.$$

Thus, $\mathbf{b}$ is a safe bidding strategy. □

The rest of the proof follows similar to that of Theorem C.6 which gives the desired result.

### C.5. Proof of Theorem 5.5

We need to establish upper bounds on $\Lambda_{\mathcal{F}^{\mathcal{H}_-}_{m'}, \mathcal{U}^\star_m}$. From Theorem C.6 and Theorem 5.3, we have:

$$
\begin{aligned}
\Lambda_{\mathcal{F}^{\mathcal{H}_-}_{m'}, \mathcal{U}^\star_m} &= \frac{\max_{\mathbf{b}' \in \mathcal{F}^{\mathcal{H}_-}_{m'}} \sum_{t=1}^T V(\mathbf{b}'; \boldsymbol{\beta}^t_-)}{\max_{\mathbf{b} \in \mathcal{U}^\star_m} \sum_{t=1}^T V(\mathbf{b}; \boldsymbol{\beta}^t_-)} \\
&= \frac{\max_{\mathbf{b}' \in \mathcal{F}^{\mathcal{H}_-}_{m'}} \sum_{t=1}^T V(\mathbf{b}'; \boldsymbol{\beta}^t_-)}{\max_{\mathbf{b}'' \in \mathcal{F}^{\mathcal{H}_-}_{m}} \sum_{t=1}^T V(\mathbf{b}''; \boldsymbol{\beta}^t_-)} \cdot \frac{\max_{\mathbf{b}'' \in \mathcal{F}^{\mathcal{H}_-}_{m}} \sum_{t=1}^T V(\mathbf{b}''; \boldsymbol{\beta}^t_-)}{\max_{\mathbf{b} \in \mathcal{U}^\star_m} \sum_{t=1}^T V(\mathbf{b}; \boldsymbol{\beta}^t_-)} \\
&= \Lambda_{\mathcal{F}^{\mathcal{H}_-}_{m'}, \mathcal{F}^{\mathcal{H}_-}_{m}} \cdot \Lambda_{\mathcal{F}^{\mathcal{H}_-}_{m}, \mathcal{U}^\star_m} \leq \frac{2m'}{m}.
\end{aligned}
$$

## D. Tight Lower Bounds for Results in Section 5

### D.1. Tight Lower Bound for Theorem 5.3 (For $m = 1$)

In this section, we construct a bid history, $\mathcal{H}_-$ and valuation curve $\mathbf{v}$ (equivalently the $\mathcal{U}^\star_m$) for which the upper bound on $\Lambda_{\mathcal{F}^{\mathcal{H}_-}_m, \mathcal{U}^\star_m}$, presented in Theorem 5.3, is tight. Recall that for any choice of $\mathcal{H}_-$, $\Lambda_{\mathcal{F}^{\mathcal{H}_-}_m, \mathcal{U}^\star_m}(\mathcal{H}_-) \leq 2 - \theta_{\mathcal{H}_-}$, where $\theta_{\mathcal{H}_-} \leq \max_{j \in [m]}(W_{\widehat{Q}_j}/W_{Q^*_j})$. To minimize $\theta_{\mathcal{H}_-}$, we choose a valuation curve that is very weakly decreasing. We then set the competing bids such that $\widehat{Q}_j \ll Q^*_j$ for all $j \in [m]$. Finally, we determine the values of $|T_{j,0}|$ and $|T_{j,1}|$ for which the upper bound is tight. See the definition of these quantities in Appendix C.1, where the first part of the theorem is proven.

We present the case when $m = 1$ below. The case for $m \geq 2$ is deferred to Appendix D.2. Although the main idea for both the cases are the same, for $m \geq 2$, the construction is more involved, and hence presented separately. Formally, for any $\delta \in (0, 1/2]$, we design a bid history, $\mathcal{H}_-$, and valuation vector, $\mathbf{v}$ for which $\Lambda_{\mathcal{F}^{\mathcal{H}_-}_1, \mathcal{U}^\star_1}(\mathcal{H}_-, \mathbf{v}) \geq 2 - \delta$.

Let $M = 2\lceil \frac{1}{\delta} \rceil$. Suppose $\mathbf{v} = [1, v, \cdots, v] \in \mathbb{R}^M$, target RoI $\gamma = 0$ and $v = 1 - 4\epsilon$ where $\epsilon = \frac{\delta - 1/M}{4(1 - 1/M)} < \frac{\delta}{4}$. Observe that $\epsilon \in (0, 1/8)$ as $\delta \leq 1/2$. Let $T = M$ and $K = M + 1$. The bid history is defined as:

$$
\boldsymbol{\beta}^{-(j)}_{-,t} = \begin{cases} 1 - \epsilon, & \text{if } t \leq M - 1 \text{ and } j = 1 \\ C, & \text{if } t \leq M - 1 \text{ and } 2 \leq j \leq K \\ \epsilon, & \text{if } t = M \text{ and } 1 \leq j \leq K \end{cases}
$$

where $C \gg w_1$. The bid history is presented in Table 1.

Table 1. Bid history for tight lower bound for $m = 1$. Here, $C \gg 1$ and $\epsilon > 0$ is a small real number.

| $t = 1$ | $t = 2$ | $\cdots$ | $t = M - 1$ | $t = M$ |
|---|---|---|---|---|
| $C$ | $C$ | $\cdots$ | $C$ | $\epsilon$ |
| $\vdots$ | $\vdots$ | $\cdots$ | $\vdots$ | $\vdots$ |
| $C$ | $C$ | $\cdots$ | $C$ | $\vdots$ |
| $1 - \epsilon$ | $1 - \epsilon$ | $\cdots$ | $1 - \epsilon$ | $\epsilon$ |

Notice that the constructed $\mathcal{H}_-$ contains $M - 1$ rounds with high competition (where the bidder can acquire at most 1 unit), while there is one round with minimal competition, allowing the bidder to obtain any desired number of units. The

1-uniform safe bidding strategies of the form $(w_q, q)$ do not perform well universally on all rounds because with increase in $q$, despite the increasing demand (i.e., $q$), the bid value $w_q$ decreases, thereby reducing the likelihood of acquiring any units.

**Notations.** For a given bid history $\mathcal{H}_-$, recall that for any $m \in \mathbb{N}$, $\mathbf{b}_m^{\mathsf{OPT}}(\mathcal{H}_-) = \operatorname{argmax}_{\mathbf{b} \in \mathcal{F}_m^{\mathcal{H}_-}} \sum_{t=1}^{T} V(\mathbf{b}; \boldsymbol{\beta}_-^t)$. We define $V_m^{\mathsf{OPT}}(\mathcal{H}_-) = \sum_{t=1}^{T} V(\mathbf{b}_m^{\mathsf{OPT}}(\mathcal{H}_-); \boldsymbol{\beta}_-^t)$, the value obtained by $\mathbf{b}_m^{\mathsf{OPT}}(\mathcal{H}_-)$ over the bid history $\mathcal{H}_-$. Similarly, $\mathbf{b}_m^{\mathsf{SAFE}}(\mathcal{H}_-) := \operatorname{argmax}_{\mathbf{b} \in \mathcal{U}_m^\star} \sum_{t=1}^{T} V(\mathbf{b}; \boldsymbol{\beta}_-^t)$ and $V_m^{\mathsf{SAFE}}(\mathcal{H}_-) = \sum_{t=1}^{T} V(\mathbf{b}_m^{\mathsf{SAFE}}(\mathcal{H}_-); \boldsymbol{\beta}_-^t)$.

**Computing the optimal strategy in $\mathcal{F}_1^{\mathcal{H}_-}$.** We observe that no bidding strategy in $\mathcal{F}_1^{\mathcal{H}_-}$ can obtain more than 1 unit in the first $M - 1$ rounds and $M$ units in the final round.

We claim that $\mathbf{b}_1^{\mathsf{OPT}}(\mathcal{H}_-) = (1, M)$. Observe that $\mathbf{b}_1^{\mathsf{OPT}}(\mathcal{H}_-)$ is allocated 1 unit in each of the first $M - 1$ rounds and $M$ units in the final round. Hence, it is allocated the maximum number of units possible. To verify that $\mathbf{b}_1^{\mathsf{OPT}}(\mathcal{H}_-)$ is feasible for the bid history, notice that for $\forall t \in [M - 1], p(\boldsymbol{\beta}^t) = 1 \leq w_1 = 1$. For round $t = M$, as $K > M$,

$$p(\boldsymbol{\beta}^t) = \epsilon \leq 1 - 4\epsilon < w_M = \mu \cdot 1 + (1 - \mu) \cdot (1 - 4\epsilon), \quad \text{where} \quad \mu = \frac{1}{M},$$

implying that RoI constraint is satisfied by $\mathbf{b}_1^{\mathsf{OPT}}(\mathcal{H}_-)$ in round $t = T$. So, $V_1^{\mathsf{OPT}}(\mathcal{H}_-) = M + v(M - 1)$.

**Computing the optimal strategy in $\mathcal{U}_1^\star$.** Here, we argue that the optimal strategy in $\mathcal{U}_1^\star$ is $(w_1, 1) = (1, 1)$. Observe that the strategy $(1, 1)$ is allocated 1 unit in each round. Hence, it obtains a total value of $M$.

As $\boldsymbol{\beta}_{-,t}^{-(1)} = 1 - \epsilon > 1 - 2\epsilon = w_2$, for $t = 1, 2, \ldots, M - 1$, bidding $(w_q, q)$ for $q \geq 2$ does not get any value in the first $M - 1$ rounds. Bidding $(w_q, q)$ gets exactly $q$ units in round $M$ as $w_q > \epsilon$ for any $q \geq 2$. So, for $2 \leq q \leq M$, the total value obtained by bidding $(w_q, q)$ is $1 + (q - 1)v < q$. Hence, $V_1^{\mathsf{SAFE}}(\mathcal{H}_-) = M$, which is the value obtained by the safe strategy $(w_1, 1) = (1, 1)$. So,

$$\Lambda_{\mathcal{F}_1^{\mathcal{H}_-}; \mathcal{U}_1^\star}(\mathcal{H}_-, \mathbf{v}) = \frac{V_1^{\mathsf{OPT}}(\mathcal{H}_-)}{V_1^{\mathsf{SAFE}}(\mathcal{H}_-)} = \frac{M + v(M - 1)}{M} = \frac{2M - 1 - 4\epsilon(M - 1)}{M} = 2 - \delta.$$

## D.2. Tight Lower Bound for Theorem 5.3 (For $m \geq 2$)

In this section, we design a bid history $\mathcal{H}_-$ and valuation vector, $\mathbf{v}$ such that for any $\delta \in (0, 1/2]$, $\Lambda_{\mathcal{F}_m^{\mathcal{H}_-}, \mathcal{U}_m^\star}(\mathcal{H}_-, \mathbf{v}) \geq 2 - \delta$. By definition, for any $\mathcal{H}_-$ and $m \geq 2$,

$$\Lambda_{\mathcal{F}_m^{\mathcal{H}_-}, \mathcal{U}_m^\star}(\mathcal{H}_-, \mathbf{v}) = \frac{V_m^{\mathsf{OPT}}(\mathcal{H}_-)}{V_m^{\mathsf{SAFE}}(\mathcal{H}_-)} \geq \frac{V_m^{\mathsf{OPT}}(\mathcal{H}_-)}{m V_1^{\mathsf{SAFE}}(\mathcal{H}_-)}, \tag{46}$$

where the second inequality follows because $\Lambda_{\mathcal{U}_m^\star, \mathcal{U}_1^\star} \leq m$ (by Theorem 5.4). So, instead of computing $V_m^{\mathsf{SAFE}}(\mathcal{H}_-)$ directly which requires solving a DP, we obtain $V_1^{\mathsf{SAFE}}(\mathcal{H}_-)$ and show that the bound is tight.

### D.2.1. CONSTRUCTION OF $\mathcal{H}_-$.

We first decide all the parameters.

- Fix $m \geq 2$ and any integer $N \geq 2\left\lceil \frac{1}{\delta} \right\rceil$.

- Let $M = N^{2m-1}$. Consider $T = N^{2m-1}$ rounds and $K = N^{2m-1} + 1$ units in each auction.

- Let $\epsilon' = \frac{m\delta/(2m-1) - 1/N}{2(1 - 1/N)} < \frac{\delta}{2} \leq \frac{1}{4}$. Set $\epsilon$ such that $\epsilon' = \epsilon N^{2m-1}(N^{2m-1} + 1)$.

Consider a valuation vector $\mathbf{v} = [1, v, \cdots, v]$ such that $v = 1 - 2\epsilon'$, and target RoI $\gamma = 0$. Partition the $N^{2m-1}$ rounds into $2m$ partitions such that the first partition has 1 round and the $j^{th}$ partition has $N^{j-1} - N^{j-2}$ rounds for $2 \leq j \leq 2m$. Each partition has identical competing bid profile submitted by other bidders. In particular,

1. The first partition (containing one round) has all the bids submitted by others as $w_{N^{2m-1}+1} + \epsilon$.

2. If $j > 1$ and $j$ is odd, for the $j^{th}$ partition (of size $N^{j-1} - N^{j-2}$), the smallest $N^{2m-j} + 1$ winning competing bids are $w_{N^{2m-j+1}} + \epsilon$ and the remaining bids are $C \gg w_1$.

3. If $j > 1$ and $j$ is even, for the $j^{th}$ partition (of size $N^{j-1} - N^{j-2}$), the smallest $N^{2m-j}$ winning competing bids are $w_{N^{2m-j+1}} + \epsilon$ and the remaining bids are $C \gg w_1$.

We present an example for such a bid history in Table 2.

*Table 2.* Bid history achieving tight lower bound for $m = 2$. Each round in the same partition has identical competing bid profile. Total number of units in each auction is $K = N^3 + 1$.

| Partition 1 $t = 1$ | Partition 2 $t \in [2, N]$ | Partition 3 $t \in [N+1, N^2]$ | Partition 4 $t \in [N^2 + 1, N^3]$ |
|---|---|---|---|
| 0 bids are $C$ | $N^3 - N^2 + 1$ bids are $C$ | $N^3 - N$ bids are $C$ | $N^3$ bids are $C$ |
| $N^3 + 1$ bids are $w_{N^3+1} + \epsilon$ | $N^2$ bids are $w_{N^2+1} + \epsilon$ | $N + 1$ bids are $w_{N+1} + \epsilon$ | 1 bid is $w_2 + \epsilon$ |

### D.2.2. COMPUTING $\mathbf{b}_m^{\mathsf{OPT}}(\mathcal{H}_-)$.

We make the following claim about the optimal $m$-uniform bidding strategy for the constructed $\mathcal{H}_-$.

**Lemma D.1.** *For the aforementioned $\mathcal{H}_-$, $\mathbf{b}_m^{\mathsf{OPT}}(\mathcal{H}_-) = \langle (b_1, q_1), \ldots, (b_m, q_m) \rangle$ where*

$$(b_j, q_j) = \begin{cases} (1, N), & \text{if } j = 1 \\ \left( w_{N^{2j-2}}, N^{2j-1} - N^{2j-3} \right), & \text{if } 2 \leq j \leq m. \end{cases} \tag{47}$$

*Furthermore,*

$$V_m^{\mathsf{OPT}}(\mathcal{H}_-) = N^{2m-1} + (2m-1)(N^{2m-1} - N^{2m-2})v.$$

*Proof.* We begin by a crucial observation that the bid history does not allow obtaining more than $N^{2m-j}$ units in the $j^{th}$ partition while satisfying the RoI constraint, *irrespective of the number of bids* submitted by the bidder. To verify this, note that, the maximum number of units that can be allocated to any bidding strategy in the $j^{th}$ partition is either $N^{2m-j}$ or $N^{2m-j} + 1$ (depending on if $j$ is even or odd). Suppose contrary to our claim, the bidder is allocated $N^{2m-j} + 1$ units in the some round $t$ in the $j^{th}$ partition by bidding some $\mathbf{b}$. Let $\boldsymbol{\beta}^t = (\mathbf{b}, \boldsymbol{\beta}_-^t)$ be the complete bid profile. So, $p(\boldsymbol{\beta}^t) \geq w_{N^{2m-j+1}} + \epsilon$ but $x(\boldsymbol{\beta}^t) = N^{2m-j} + 1$ indicating that the RoI constraint is violated, which verifies our claim.

So, the total number of units, $N_{\text{total}}$, that can be obtained by the bidder over all the rounds is:

$$N_{\text{total}} \leq N^{2m-1} + \sum_{j=2}^{2m} N^{2m-j}(N^{j-1} - N^{j-2}) = 2mN^{2m-1} - (2m-1)N^{2m-2} =: N_{\max}.$$

Now, we compute the the number of units obtained by bidding $\mathbf{b}_m^{\mathsf{OPT}}(\mathcal{H}_-)$ and show that it is allocated $N_{\max}$ units for the constructed bid history, demonstrating that it is the optimal bidding strategy.

Consider any auction in the $j^{th}$ partition. The lowest winning bid in the bid profile is $w_{N^{2m-j+1}} + \epsilon$. Note that the unique bid values (ignoring the quantity for the sake of brevity) in $\mathbf{b}_m^{\mathsf{OPT}}(\mathcal{H}_-)$ are $\mathbf{b} = \{1, w_{N^2}, \ldots, w_{N^{2m-2}}\}$. We claim that the winning bid values of $\mathbf{b}_m^{\mathsf{OPT}}(\mathcal{H}_-)$ in the $j^{th}$ partition are $\widehat{\mathbf{b}} = \{1, w_{N^2}, \ldots, w_{N^{2m+2\lfloor -j/2 \rfloor}}\}$. This is true because the least bid value in $\widehat{\mathbf{b}}$ is greater than $w_{N^{m-j+1}} + \epsilon$, i.e.,

$$w_{N^{2m+2\lfloor -j/2 \rfloor}} - (w_{N^{2m-j+1}} + \epsilon) \geq w_{N^{2m-j}} - (w_{N^{2m-j+1}} + \epsilon)$$

$$= \frac{1-v}{N^{2m-j}(N^{2m-j}+1)} - \epsilon = \frac{2\epsilon N^{2m-1}(N^{2m-1}+1)}{N^{2m-j}(N^{2m-j}+1)} - \epsilon \geq \epsilon > 0.$$

Let $N_j$ denote the number of units allocated to $\mathbf{b}_m^{\mathsf{OPT}}(\mathcal{H}_-)$ in each auction in the $j^{th}$ partition. There are two cases:

(a) for $j$ odd, recall that for the $j^{th}$ partition (of size $N^{j-1} - N^{j-2}$), the smallest $N^{2m-j} + 1$ winning competing bids are $w_{N^{2m-j+1}} + \epsilon$ and the remaining bids are $C \gg w_1$. Then,

$$N_j = N + \sum_{\ell=2}^{m-\frac{j-1}{2}} (N^{2\ell-1} - N^{2\ell-3}) = N^{2m-j}.$$

(b) For $j$ even, recall that for the $j^{th}$ partition (of size $N^{j-1} - N^{j-2}$), the smallest $N^{2m-j}$ winning competing bids are $w_{N^{2m-j+1}} + \epsilon$ and the remaining bids are $C \gg w_1$.

$$N_j = \min\left\{ N^{2m-j}, N + \sum_{\ell=2}^{m+1-\frac{j}{2}} (N^{2\ell-1} - N^{2\ell-3}) \right\} = \min\{N^{2m-j}, N^{2m-j+1}\} = N^{2m-j}.$$

Here, the minimum is taken over two quantities where the first quantity is the number of competing bids less than $C$ in any round $t$ in the $j^{th}$ partition and the second quantity represents the total demand of the winning bids in $\mathbf{b}_m^{\mathsf{OPT}}(\mathcal{H}_-)$ for that round. So, the total number of units obtained across all rounds is

$$N^{2m-1} + \sum_{j=2}^{2m} N^{2m-j}(N^{j-1} - N^{j-2}) = 2mN^{2m-1} - (2m-1)N^{2m-2}.$$

As this is the maximum number of units that can be obtained by the bidder, $\mathbf{b}_m^{\mathsf{OPT}}(\mathcal{H}_-)$ is optimal. The total value obtained by bidding $\mathbf{b}_m^{\mathsf{OPT}}(\mathcal{H}_-)$ is

$$V_m^{\mathsf{OPT}}(\mathcal{H}_-) = 1 + (N^{2m-1} - 1)v + \sum_{j=2}^{2m} (N^{j-1} - N^{j-2})(1 + (N^{2m-j} - 1)v)$$

$$= N^{2m-1} + (2m-1)(N^{2m-1} - N^{2m-2})v.$$

$\square$

D.2.3. COMPUTING $V_1^{\mathsf{SAFE}}(\mathcal{H}_-)$

Recall that we compute $V_1^{\mathsf{SAFE}}(\mathcal{H}_-)$ and invoke the bounds on $\Lambda_{u_m^\star, u_1^\star}$, instead of directly evaluating $V_m^{\mathsf{SAFE}}(\mathcal{H}_-)$.

**Lemma D.2.** *For the aforementioned $\mathcal{H}_-$, $\mathbf{b}_1^{\mathsf{SAFE}}(\mathcal{H}_-) = (1,1)$ and $V_1^{\mathsf{SAFE}}(\mathcal{H}_-) = N^{2m-1}$.*

*Proof.* The basic idea is to enumerate the total units (value) that can be obtained by bidding $(w_q, q)$ for $q \in [N^{2m-1}]$ and then finding the maximum of those values. As $q$ can be exponential in $m$, we exploit the structure of the bid history to compute the objective in an efficient manner.

Suppose $q = 1$. The maximum number of units (value) that can be obtained by bidding $(1,1)$ is trivially $N^{2m-1}$, So, $(1,1)$ obtains a total value $N^{2m-1}$.

Suppose $q \geq 2$. Furthermore, assume $N^{2m-j} < q \leq N^{2m-j+1}$, for some $2 \leq j \leq 2m$. Consider any bid of the form $(w_q, q)$. Bidding $(w_q, q)$ does not obtain any units in the partitions indexed by $j, j+1, \ldots, 2m$ as $w_q$ is strictly less than the least winning competing bids in those partitions, i.e., $w_q \leq w_{N^{2m-j+1}} < w_{N^{2m-\ell+1}} + \epsilon$, for any $\ell \in \{j, j+1, \ldots, 2m\}$. So, if $N^{2m-j} < q \leq N^{2m-j+1}$, $(w_q, q)$ gets no units in

$$\sum_{\ell=j}^{2m} N^{\ell-1} - N^{\ell-2} = N^{2m-1} - N^{j-2} \text{ auctions.}$$

In the remaining $N^{j-2}$ auctions it can win at most $q$ units. So, the maximum value obtained by $(w_q, q)$ for any $q \geq 2$ is

$$N^{j-2}(1 + (q-1)v) \leq N^{j-2}(1 + (N^{2m-j+1} - 1)v) = N^{j-2} + (N^{2m-1} - N^{j-2})v < N^{2m-1},$$

where the last inequality holds as $v < 1$. Hence, $\mathbf{b}_1^{\mathsf{SAFE}}(\mathcal{H}_-) = (1,1)$ and $V_1^{\mathsf{SAFE}}(\mathcal{H}_-) = N^{2m-1}$. $\square$

Hence, from Lemma D.1, Lemma D.2 and Eq. (46)

$$
\begin{aligned}
\Lambda_{\mathcal{F}_m^{\mathcal{H}_-}, \mathcal{U}_m^\star}(\mathcal{H}_-, \mathbf{v}) &\geq \frac{V_m^{\mathsf{OPT}}(\mathcal{H}_-)}{m V_1^{\mathsf{SAFE}}(\mathcal{H}_-)} = \frac{N^{2m-1} + (2m-1)(N^{2m-1} - N^{2m-2})v}{m N^{2m-1}} \\
&= \frac{2m N^{2m-1} - (2m-1)N^{2m-2} - 2\epsilon'(2m-1)(N^{2m-1} - N^{2m-2})}{m N^{2m-1}} \\
&= 2 - \frac{2m-1}{m}\left(\frac{1}{N} + 2\epsilon'\left(1 - \frac{1}{N}\right)\right) \\
&= 2 - \delta\,,
\end{aligned}
$$

where the last step follows by substituting the value of $\epsilon'$ and thus concludes the proof.

### D.3. Tight Lower Bound for Theorem 5.4

In this section, we construct a bid history $\mathcal{H}_-$ and valuation vector, $\mathbf{v}$ such that for any $\delta \in (0, \frac{1}{2}]$, $\Lambda_{\mathcal{U}_{m'}^\star, \mathcal{U}_m^\star}(\mathcal{H}_-, \mathbf{v}) \geq \frac{m'}{m} - \delta$, i.e., the upper bound is tight for $m' \geq m + 1$ (If $m' = m$, $\Lambda_{\mathcal{U}_{m'}^\star, \mathcal{U}_m^\star} = 1$ implying the bound is tight).

Recall that Theorem 5.4 states that $\Lambda_{\mathcal{U}_m^\star, \mathcal{U}_1^\star} \leq m$ for any $m \geq 1$. So,

$$
\Lambda_{\mathcal{U}_{m'}^\star, \mathcal{U}_m^\star}(\mathcal{H}_-, \mathbf{v}) = \frac{V_{m'}^{\mathsf{SAFE}}(\mathcal{H}_-)}{V_m^{\mathsf{SAFE}}(\mathcal{H}_-)} \geq \frac{V_{m'}^{\mathsf{SAFE}}(\mathcal{H}_-)}{m V_1^{\mathsf{SAFE}}(\mathcal{H}_-)} = \frac{1}{m} \cdot \Lambda_{\mathcal{U}_{m'}^\star, \mathcal{U}_1^\star}(\mathcal{H}_-, \mathbf{v})\,. \tag{48}
$$

We now establish a lower bound on $\Lambda_{\mathcal{U}_{m'}^\star, \mathcal{U}_1^\star}(\mathcal{H}_-, \mathbf{v})$.

D.3.1. CONSTRUCTION OF $\mathcal{H}_-$.

We first decide all the parameters.

- Fix $m' \geq 2$ and any integer $N \geq \left\lceil \frac{m'}{\delta} \right\rceil$.

- Let $M = N^{m'-1}$. Consider $T = N^{m'-1}$ rounds and $K = N^{m'-1}$ units in each auction.

- Let $\epsilon' = \frac{m\delta/(m'-1) - 1/N}{2(1-1/N)} < \frac{m\delta}{2(m'-1)} \leq \frac{1}{4}$. Set $\epsilon$ such that $\epsilon' = \epsilon N^{m'-1}(N^{m'-1} + 1)$.

Let the valuation vector be $\mathbf{v} = [1, v, \cdots, v]$ such that $v = 1 - 2\epsilon'$, and target RoI, $\gamma = 0$. Partition the $N^{m'-1}$ rounds into $m'$ partitions such that the first partition has 1 round and the $j^{th}$ partition has $N^{j-1} - N^{j-2}$ rounds for $2 \leq j \leq m'$. Each partition has identical competing bid profile submitted by the other bidders. In particular,

1. The first partition (containing one round) has all the competing bids to be $w_{N^{m'-1}+1} + \epsilon$.

2. For $2 \leq j \leq m'$, the $j^{th}$ partition (of size $N^{j-1} - N^{j-2}$); the smallest $N^{m'-j} + 1$ competing winning bids are $w_{N^{m'-j}+1} + \epsilon$ and the remaining bids are $C \gg w_1$.

We present an example for such a bid history in Table 3.

Table 3. Bid history attaining tight lower bound for $m' = 4$ (and $m = 1$). Each round in the same partition has identical competing bid profile. Total number of units in each auction is $K = N^3$.

| Partition 1 | Partition 2 | Partition 3 | Partition 4 |
|---|---|---|---|
| $t = 1$ | $t \in [2, N]$ | $t \in [N+1, N^2]$ | $t \in [N^2+1, N^3]$ |
| 0 bids are $C$ | $N^3 - N^2 - 1$ bids are $C$ | $N^3 - N - 1$ bids are $C$ | $N^3 - 2$ bids are $C$ |
| $N^3$ bids are $w_{N^3+1} + \epsilon$ | $N^2 + 1$ bids are $w_{N^2+1} + \epsilon$ | $N + 1$ bids are $w_{N+1} + \epsilon$ | $2$ bids are $w_2 + \epsilon$ |

D.3.2. COMPUTING $\mathbf{b}_{m'}^{\mathsf{SAFE}}(\mathcal{H}_-)$

We make the following claim about the optimal $m'$-uniform safe bidding strategy for $\mathcal{H}_-$.

**Lemma D.3.** *For the aforementioned $\mathcal{H}_-$, $\mathbf{b}_{m'}^{\mathsf{SAFE}}(\mathcal{H}_-) = \langle (b_1, q_1), \dots, (b'_m, q'_m) \rangle$ where*

$$(b_j, q_j) = \begin{cases} (1, 1), & \text{if } j = 1 \\ \left( w_{N^{j-1}}, N^{j-1} - N^{j-2} \right), & \text{if } 2 \le j \le m' \end{cases} . \tag{49}$$

*Furthermore,*

$$V_{m'}^{\mathsf{SAFE}}(\mathcal{H}_-) = N^{m'-1} + (m' - 1)(N^{m'-1} - N^{m'-2})v .$$

*Proof.* We begin by a crucial observation that the bid history does not allow obtaining more than $N^{m'-j}$ units in the $j^{th}$ partition, while satisfying the RoI constraint, *irrespective of the number of bids* submitted by the bidder. To verify this, suppose contrary to our claim, that the bidder is allocated $N^{m'-j} + 1$ units in the some round $t$ in the $j^{th}$ partition by bidding some $\mathbf{b}$. Let $\boldsymbol{\beta}^t = (\mathbf{b}, \boldsymbol{\beta}_-^t)$ be the complete bid profile. Hence, $p(\boldsymbol{\beta}^t) \ge w_{N^{m'-j}+1} + \epsilon$ but $x(\boldsymbol{\beta}^t) = N^{m'-j} + 1$ indicating that the RoI constraint is violated which verifies our claim.

So, the total number of units, $N_{\text{total}}$, that can be obtained by the bidder across all the rounds is:

$$N_{\text{total}} \le N^{m'-1} + \sum_{j=2}^{m'} N^{m'-j}(N^{j-1} - N^{j-2}) = m'N^{m'-1} - (m'-1)N^{m'-2} := N_{\max}.$$

Now, we compute the number of units obtained by bidding $\mathbf{b}_{m'}^{\mathsf{SAFE}}(\mathcal{H}_-)$ and show that it is allocated $N_{\max}$ units across all the auctions, implying that it is the optimal bidding strategy.

To show $\mathbf{b}_{m'}^{\mathsf{SAFE}}(\mathcal{H}_-)$ is optimal, consider any auction in the $j^{th}$ partition. The lowest winning competing bid is $w_{N^{m'-j}+1} + \epsilon$. Note that the unique bid values (ignoring the quantity for the sake of brevity) in $\mathbf{b}_{m'}^{\mathsf{SAFE}}(\mathcal{H}_-)$, provided in Eq. (49), are $\mathbf{b} = \{1, w_N, \dots, w_{N^{m'-1}}\}$. We claim that the winning bid values of $\mathbf{b}_{m'}^{\mathsf{SAFE}}(\mathcal{H}_-)$ in the $j^{th}$ partition are $\widetilde{\mathbf{b}} = \{1, w_N, \dots, w_{N^{m'-j}}\}$. Again recall that for $2 \le j \le m'$, for the $j^{th}$ partition (of size $N^{j-1} - N^{j-2}$), the smallest $N^{m'-j} + 1$ competing winning bids are $w_{N^{m'-j}+1} + \epsilon$ and the remaining bids are $C \gg w_1$. And here, the least bid value in $\widetilde{\mathbf{b}}$ is greater than $w_{N^{m'-j}+1} + \epsilon$, i.e.,

$$w_{N^{m'-j}} - (w_{N^{m'-j}+1} + \epsilon) = \frac{1 - v}{N^{m'-j}(N^{m'-j} + 1)} - \epsilon = \frac{2\epsilon N^{m'-1}(N^{m'-1} + 1)}{N^{m'-j}(N^{m'-j} + 1)} - \epsilon \ge \epsilon > 0 .$$

Moreover, observe that the bidder is allocated the maximum number of units demanded for each of the bid value in $\widetilde{\mathbf{b}}$. So, the number of units in each auction in the $j^{th}$ partition by bidding $\mathbf{b}_{m'}^{\mathsf{SAFE}}(\mathcal{H}_-)$ is

$$N_j = 1 + \sum_{\ell=2}^{m'-j+1} (N^{\ell-1} - N^{\ell-2}) = N^{m'-j} .$$

Hence, the total number of units obtained across all rounds is

$$N^{m'-1} + \sum_{j=2}^{m'} N^{m'-j}(N^{j-1} - N^{j-2}) = m'N^{m'-1} - (m'-1)N^{m'-2} = N_{\max}.$$

As this is the maximum number of units that can be obtained by the bidder, $\mathbf{b}_{m'}^{\mathsf{SAFE}}(\mathcal{H}_-)$ is optimal. The total value obtained by bidding $\mathbf{b}_{m'}^{\mathsf{SAFE}}(\mathcal{H}_-)$ is given by

$$V_{m'}^{\mathsf{SAFE}}(\mathcal{H}_-) = 1 + (N^{m'-1} - 1)v + \sum_{j=2}^{m'} (N^{j-1} - N^{j-2})(1 + (N^{m'-j} - 1)v)$$

$$= N^{m'-1} + (m' - 1)(N^{m'-1} - N^{m'-2})v .$$

$\square$

D.3.3. COMPUTING $\mathbf{b}_1^{\mathsf{SAFE}}(\mathcal{H}_-)$

In this section, we compute the optimal 1-uniform safe bidding strategy for the bid history, $\mathcal{H}_-$.

**Lemma D.4.** *For the constructed $\mathcal{H}_-$, $\mathbf{b}_1^{\mathsf{SAFE}}(\mathcal{H}_-) = (1, 1)$ and $V_1^{\mathsf{SAFE}}(\mathcal{H}_-) = N^{m-1}$.*

*Proof.* The proof is similar to that of Lemma D.2 in the sense that it involves enumerating the total units obtaining by submitting $(w_q, q)$ for all $q \in [N^{m-1}]$ in an efficient manner by leveraging the structure in the bid history. $\square$

Substituting values from Lemma D.3, Lemma D.4 and Eq. (48)

$$
\begin{aligned}
\Lambda_{\mathcal{U}_{m'}^\star, \mathcal{U}_m^\star}(\mathcal{H}_-, \mathbf{v}) \geq \frac{1}{m} \cdot \Lambda_{\mathcal{U}_{m'}^\star, \mathcal{U}_1^\star}(\mathcal{H}_-, \mathbf{v}) &= \frac{V_{m'}^{\mathsf{SAFE}}(\mathcal{H}_-)}{m V_1^{\mathsf{SAFE}}(\mathcal{H}_-)} \\
&= \frac{N^{m'-1} + (m'-1)(N^{m'-1} - N^{m'-2})v}{m N^{m'-1}} \\
&= \frac{m' N^{m'-1} - (m'-1)N^{m'-2} - 2\epsilon'(m'-1)(N^{m'-1} - N^{m'-2})}{m N^{m'-1}} \\
&= \frac{m'}{m} - \frac{(m'-1)}{m}\left(\frac{1}{N} + 2\epsilon'\left(1 - \frac{1}{N}\right)\right) \\
&= \frac{m'}{m} - \delta,
\end{aligned}
$$

where the last step follows by substituting the value of $\epsilon'$ and thus concludes the proof.

**Corollary D.5.** *As $\mathcal{U}_m^\star \subseteq \mathcal{F}_m^{\mathcal{H}_-}$ for all $m \in \mathbb{N}$, the upper bound in Theorem C.6 is also tight, i.e., for any $m' \geq m$ and $\delta \in (0, \frac{1}{2}]$, there exists a bid history $\mathcal{H}_-$ and valuation curve $\mathbf{v}$ such that $\Lambda_{\mathcal{F}_{m'}^{\mathcal{H}_-}, \mathcal{F}_m^{\mathcal{H}_-}}(\mathcal{H}_-, \mathbf{v}) \geq \frac{m'}{m} - \delta$.*

### D.4. Tight Lower Bound for Theorem 5.5

In this section, we present a bid history $\mathcal{H}_-$ and valuation vector, $\mathbf{v}$ such that for any $\delta \in (0, \frac{1}{2}]$, $\Lambda_{\mathcal{F}_{m'}^{\mathcal{H}_-}, \mathcal{U}_m^\star}(\mathcal{H}_-, \mathbf{v}) \geq \frac{2m'}{m} - \delta$, i.e., the upper bound is tight for $m' \geq m$.

Recall that Theorem 5.4 states that $\Lambda_{\mathcal{U}_m^\star, \mathcal{U}_1^\star} \leq m$ for any $m \geq 1$. So,

$$
\Lambda_{\mathcal{F}_{m'}^{\mathcal{H}_-}, \mathcal{U}_m^\star}(\mathcal{H}_-, \mathbf{v}) = \frac{V_{m'}^{\mathsf{OPT}}(\mathcal{H}_-)}{V_m^{\mathsf{SAFE}}(\mathcal{H}_-)} \geq \frac{V_{m'}^{\mathsf{OPT}}(\mathcal{H}_-)}{m V_1^{\mathsf{SAFE}}(\mathcal{H}_-)} = \frac{1}{m} \cdot \Lambda_{\mathcal{F}_{m'}^{\mathcal{H}_-}, \mathcal{U}_1^\star}(\mathcal{H}_-, \mathbf{v}). \tag{50}
$$

Recall that for showing tightness of the bound for Theorem 5.3 for any $m \geq 2$, we computed a lower bound on $\Lambda_{\mathcal{F}_m^{\mathcal{H}_-}, \mathcal{U}_1^\star}(\mathcal{H}_-, \mathbf{v})$ (cf. Eq. (46)). Thus, we consider a valuation curve and a bid history that has a structure identical to the one presented in Appendix D.2 but with the following modified parameters,

- Fix $m' \geq 1$ and any integer $N \geq 2\left\lceil \frac{2m'}{\delta} \right\rceil$.

- Let $M = N^{2m'-1}$. Consider $T = N^{2m'-1}$ rounds and $K = N^{2m'-1} + 1$ units in each auction.

- Let $\epsilon' = \frac{m\delta/(2m'-1) - 1/N}{2(1-1/N)} < \frac{m\delta}{2(2m'-1)} \leq \frac{1}{4}$. Set $\epsilon$ such that $\epsilon' = \epsilon N^{2m'-1}(N^{2m'-1} + 1)$.

Consider a valuation vector $\mathbf{v} = [1, v, \cdots, v]$ such that $v = 1 - 2\epsilon'$, and target RoI $\gamma = 0$. Substituting the values from Lemma D.1 and Lemma D.2,

$$
\begin{aligned}
\Lambda_{\mathcal{F}_{m'}^{\mathcal{H}_-}, \mathcal{U}_m^\star}(\mathcal{H}_-, \mathbf{v}) \geq \frac{1}{m} \cdot \Lambda_{\mathcal{F}_{m'}^{\mathcal{H}_-}, \mathcal{U}_1^\star}(\mathcal{H}_-, \mathbf{v}) &= \frac{N^{2m'-1} + (2m'-1)(N^{2m'-1} - N^{2m'-2})v}{m N^{2m'-1}} \\
&= \frac{2m' N^{2m'-1} - (2m'-1)N^{2m'-2} - 2\epsilon'(2m'-1)(N^{2m'-1} - N^{2m'-2})}{m N^{2m'-1}} \\
&= \frac{2m'}{m} - \frac{(2m'-1)}{m}\left(\frac{1}{N} + 2\epsilon'\left(1 - \frac{1}{N}\right)\right)
\end{aligned}
$$

$$= \frac{2m'}{m} - \delta \,,$$

where the last step follows by substituting the value of $\epsilon'$ and thus concludes the proof.

## E. Simulation Details

In Section 5.3, we conducted numerical experiments to estimate the richness ratio (or equivalently $\Lambda_{\mathcal{B}_c, \mathcal{U}_m^\star}$) in practice for the classes of bidding strategies discussed earlier. We now discuss a few details regarding the same.

**Dataset.** We estimate the richness ratio for the EU ETS emission permit auction data for 2022 and 2023. However, only aggregate statistics of the submitted bids is publicly available for privacy reasons. Hence, we synthesize individual level bid data from these available statistics. The exact procedure to reconstruct the bid data is presented in Appendix E.1. We sample the values from the Unif$[0, 1]$ distribution. In each simulation, we sample $T \sim$ Unif$[100, 300]$ auctions and let $M \sim$ Unif$[10, 80]$. We vary $m = 1$ to $m = 10$ and average over 100 simulations to obtain plots in Fig. 4. As computing $V_m^{\mathsf{OPT}}(\mathcal{H}_-)$ (the value obtained by the optimal bidding strategy in $\mathcal{F}_m^{\mathcal{H}_-}$) can be non-trivial, we obtain a uniform upper bound for $V_m^{\mathsf{OPT}}(\mathcal{H}_-)$ that is *independent of* $m$ (see details in Appendix E.2).

### E.1. Reconstructing Individual Bid Data

We obtained the publicly available auction data for $T_{\max} = 443$ EU ETS emission permit auctions held in 2022 and 2023 (EEX, 2023). For each auction indexed by $t \in [T_{\max}]$, we have the following relevant information: the minimum bid ($b_{\min}^t$), the maximum bid ($b_{\max}^t$), the average of the bids ($b_{\mathrm{avg}}^t$), the median of the bids ($b_{\mathrm{med}}^t$), and the number of bid-quantity pairs submitted ($n_{\mathrm{sub}}^t$). We normalized the bids to be in $[0, 1]$. For all rounds $t$, $b_{\mathrm{avg}}^t \approx b_{\mathrm{med}}^t$ (linear regression yields coefficient 1.01 and intercept $-0.008$).

Upon further investigation, we observed that, except a few, a significant number of auctions had either $b_{\min}^t \approx b_{\mathrm{avg}}^t \ll b_{\max}^t$ (Type I) or $b_{\min}^t \ll b_{\mathrm{avg}}^t \approx b_{\max}^t$ (Type II). As $b_{\mathrm{avg}}^t \approx b_{\mathrm{med}}^t, \forall t$, we deduce that for Type I, most of the bids are concentrated in the interval $[b_{\min}^t, 2b_{\mathrm{avg}}^t - b_{\min}^t]$ whereas for Type II, most of the bids are in the interval $[2b_{\mathrm{avg}}^t - b_{\max}^t, b_{\max}^t]$. We posit that for Type I (resp. Type II) auctions, $f \in (0, 1)$ fraction of the bids ($n_{\mathrm{sub}}^t$) are in $[b_{\min}^t, 2b_{\mathrm{avg}}^t - b_{\min}^t]$ (resp. $[2b_{\mathrm{avg}}^t - b_{\max}^t, b_{\max}^t]$) and the $1 - f$ fraction of bids are in $[2b_{\mathrm{avg}}^t - b_{\max}^t, b_{\max}^t]$ (resp. $[b_{\min}^t, 2b_{\mathrm{avg}}^t - b_{\max}^t]$). If for Type I (resp. Type II) auctions, $2b_{\mathrm{avg}}^t > b_{\min}^t + b_{\max}^t$ (resp. $2b_{\mathrm{avg}}^t < b_{\min}^t + b_{\max}^t$), we assume that all the bids are uniformly present in the interval $[b_{\min}^t, b_{\max}^t]$. With these assumptions, we generate individual bid data for each auction by sampling uniformly from these intervals.

After generating the individual bid data, we compute the metrics for the reconstructed bids (say $\widehat{b}_{\mathrm{avg}}^t$) for each auction and reject those with a relative error of at least $\delta$ (tolerance). For our simulations, we set $\delta = 0.05$. We perform a grid search for $f$ to maximize the number of auctions where the metrics of the reconstructed data are within $\delta$ relative error of the actual metrics, and obtain that $f = 0.97$. Following this pre-processing, we have reconstructed individual bid data for $T = 341$ auctions. The bids are normalized to be in $[0, 1]$.

### E.2. An Uniform Upper Bound for $V_m^{\mathsf{OPT}}(\mathcal{H}_-)$.

For any bid history, $\mathcal{H}_- = [\boldsymbol{\beta}_-^t]_{t \in [T]}$, suppose $\mathbf{b}_m^{\mathsf{OPT}}(\mathcal{H}_-)$ is allocated $r_t$ units in any round $t$. Then, by Lemma C.5 (1), we know that $(w_{r_t}, r_t)$ also obtains $r_t$ units in round $t$. So,

$$V(\mathbf{b}_m^{\mathsf{OPT}}(\mathcal{H}_-); \boldsymbol{\beta}_-^t) = V((w_{r_t}, r_t); \boldsymbol{\beta}_-^t) \leq \max_{\mathbf{b} \in \mathcal{U}_1^\star} V(\mathbf{b}; \boldsymbol{\beta}_-^t)$$

$$\implies V_m^{\mathsf{OPT}}(\mathcal{H}_-) = \sum_{t=1}^T V(\mathbf{b}_m^{\mathsf{OPT}}(\mathcal{H}_-); \boldsymbol{\beta}_-^t) \leq \sum_{t=1}^T \max_{\mathbf{b} \in \mathcal{U}_1^\star} V(\mathbf{b}; \boldsymbol{\beta}_-^t) \,.$$

## F. Resolving Ties

We first emphasize that the effect of resolution of ties on the objective function under the value maximization behavioral model is fundamentally different from the quasilinear utility maximization model. To illustrate this, consider a single-item second price auction with two bidders each valuing the item equally at $v$. Assume that both the bidders bid $v$ and there exists a definitive tie breaking rule (either randomized or deterministic). If the bidders are considered quasilinear utility

maximizers, the objective function value (value obtained minus the payments) for both the bidders is 0. However, under the value maximization model (assume $\gamma = 0$ for both the bidders), the objective function value for the winning bidder is $v$ and 0 for the losing bidder.[12] Hence, we need to carefully analyze the key results of our work in the event of ties.

In the context of multi-unit auctions, we can classify ties into two informal types: (a) 'good ties'—ties occurring at any bid other than the last accepted bid (LAB) and (b) 'bad ties', which are the ties occurring at the LAB. All the results in this work are unaffected in case only 'good ties' occur. Thus, we focus on 'bad ties' for the rest of this section. For this discussion, we consider a public, deterministic tie breaking rule under which ties are always broken in favor of lower indexed bidder. In the presence of ties, the undominated class of safe bidding strategies is still the class of nested strategies (Theorem 3.2). The tie breaking rule is incorporated into the decomposition in Lemma 4.1 and computing the edge weights in the DAG in the offline and online settings in Section 4, ensuring that the maximum weight path in the DAG gives the optimal offline solution and Algorithm 1 achieves sublinear regret in the online setting.

---

[12]Although, the total (liquid) welfare is $v$ in both the cases.

