# OpenReview forum: "Learning Safe Strategies for Value Maximizing Buyers in Uniform Price Auctions"
_ICML.cc/2025/Conference — ICML 2025 poster_

### Official Review · Reviewer_gJHK · 2025-03-11

**Overall Recommendation:** 4

**Summary:**

The authors study repeated uniform price auctions with respect to bidder behavior. They consider value maximizing buyer that has return of investment (RoI) constraint. The paper proposes safe bidding strategies that allow bidder to be sure that RoI will not be violated in the future rounds. The main contribution is learning algorithms that are efficient (they have sublinear regret in number of rounds T in both full-information and bandit settings). The work is enriched by considering setups with richer class of competing bidding strategies (non-safe) and relaxation of RoI constraints on each round. Finally, some claims are supported by synthetic experiments.

**Claims And Evidence:**

The submission is strongly supported by claims in the form of Theorems and Lemmas supplied with proofs. The main contribution of the work – theoretical results. So, the evidence is presented correctly. The work is supported by a large appendix, where all the statements are carefully proven.

**Essential References Not Discussed:**

I’ve not found such

**Experimental Designs Or Analyses:**

I’ve read experimental design and analyses (both in the main part – Section 5.3 and Appendix G).


Line  2024 “We sample the values from the Unif[0,1] distribution”. This assumption sounds very theoretical. Is it true that in real life (common case) the values are distributed uniformly?

**Methods And Evaluation Criteria:**

The main contribution of the work – theoretical results. The main methods – proofs are made correctly and are presented in clear way.

There are some experimental synthetic results to support the claims. They are adequate (I have a minor question – see Q section)

**Other Comments Or Suggestions:**

-	I would suggest to reduce Abstract (~ 3 times) saving space (most content presents in Intro) and use this space to add more proof sketches

**Other Strengths And Weaknesses:**

Strengths
-	Solid paper with clear theoretical contributions

-	Very nice storyline and claim support (decomposition in lemmas/theorems, proofs, structure overall)


Weaknesses
-	presentation:

o	a) the story (and contributions) of the paper repeats 3 times with different level details: Abstract, Into, Main part (Sections 2-6). A few numbers of proof sketches

o	b) several missed introduction of terms / variables in code

**Questions For Authors:**

-	Lines 124-125, right part: What is “bid”? In this place of the paper this term is introduced, while not being defined. Is it 1 number, M numbers (per each good), or K numbers? Far-far from this place (page 4) there is the discussion about bidding language, however it does not help define the initial intention in Sec.2.1..

-	Lines 141-142, right part: What is “the j-th smallest winning bid” formally? The questions refers to the previous one as the term bid is not defined earlier (is it 1 number per bidder, or M numbers.. etc). E.g., if a bid is not a single number, then how a comparison between them is organized.

-	Lines 144-145, right part: “the clearing price” is not defined. What is it formally?

-	Line 255, left part: “removing weakly dominated strategies” I have not found discussion on what is happening with weakly dominant strategies, if keep them. Can it be clarified? I see an example in Lines 236-242 about infeasible strategies that forces to introduce “safe” strategies. Is it true that “unsafe strategy” = “weakly dominant”? If not what is the difference?

-	Algorithm 2: Please, specify its inputs and outputs

-	Definition of w: In Theorem 4.2 and Algorithm 1 there is usage of w (b_l = w_{z_l}) while it is not defined earlier (within the theorem and the algorithm). In Line 118 right side, I see some definition of w as a function of v (so, in fact, there is w = w(v)), but it is very far from the theorem statement and the pseudocode. I highly recommend to state how w is obtained in Theorem 4.2 and Algorithm 1

**Relation To Broader Scientific Literature:**

The key contributions are related to works:

-	Brˆanzei, S., Derakhshan, M., Golrezaei, N., and Han, Y. Learning and collusion in multi-unit auctions. In Thirtyseventh Conference on Neural Information Processing Systems, 2023.

-	Galgana, R. and Golrezaei, N. Learning in repeated multiunit pay-as-bid auctions. Manufacturing & Service Operations Management, 2024.

-	Potfer, M., Baudry, D., Richard, H., Perchet, V., and Wan, C. Improved learning rates in multi-unit uniform price auctions. In The


Thirty-eighth Annual Conference on Neural Information Processing Systems, 2024.
The main difference: That works assume that the bidders are quasilinear utility maximizers, while this paper study value-maximizing buyers with RoI constraints (which is drastically different setup).

**Theoretical Claims:**

I’ve not checked all the proofs in detailed way (checking each implication). However, checked all main blocks – they are OK (do not seem contradictory) + Checked carefully several proofs: for Lemma 4.1 and for Theorem 4.2. No issues found.

---

> ### Author Rebuttal · Authors · 2025-03-31
>
> We thank the reviewer for the encouraging feedback and are glad the theory and structure were clear and well-received. We believe the changes suggested in response to your thoughtful comments can be incorporated into the camera-ready version using the extra page.
>
> **Re the use of Unif[0,1]:** We tested other distributions (e.g., Laplace, truncated half-normal) and found similar trends with minor differences. We chose Unif[0,1] for consistency with prior theoretical [1] and experimental work (Galgana & Golrezaei, 2024).
>
> [1] Ausubel, LM., et al. Demand reduction and inefficiency in multi-unit auctions. The Rev. of Econ. Stud., 2014.
>
> **Re the presentation:** Given the length and technical depth of the manuscript, we chose to present the main contributions comprehensively in the introduction and provide a clear roadmap. As suggested, we will shorten the abstract and use the space to add more proof sketches—especially for Theorem 4.4 (regret lower bound) and the tight instances in Theorems 5.3–5.5, which we believe are novel and of independent interest.
>
> **Re the comment on missing terms/variables:** Could the reviewer clarify which terms or variables are unclear? As noted in our response to Reviewer 9dTj, we plan to add a notation subsection at the start of Section 2 to define all terms more clearly.
>
> **Re the term “bid”:** The term “bid” refers to a single number here. We imply this in the next line (Line 126):  *“… if bidder $i$ has $j$ bids in the top $K$ positions ...”*
>
> The bidder submits a vector of bids $\mathbf{b}\in\mathbb{R}^M$ using the $m$-uniform bidding format (the bidding language): $\mathbf{b}=\langle (b_1, q_1), \dots, (b_m, q_m)\rangle,$ where the bidder bids $b_1$ for the first $q_1$ units (i.e., the first $q_1$ entries of the vector are $b_1$), $b_2$ for the next $q_2$ units, and so on.
>
> **Re the  j-th smallest winning bid:** As mentioned earlier, the word “bid” refers to a single number. In a multi-unit auction with $K$ units, each bidder submits multiple such bids and the $K$ highest bids across all bidders win. Thus, the $j$-th smallest winning bid refers to the $(K - j + 1)$-th highest bid overall.
>
> **Re the notion of Clearing Price:** The clearing price (or per-unit price) refers to the $K$-th highest bid, which each bidder pays per unit they receive (see Line 133, right).
>
> We use Example 1 (Lines 176–186) in the paper to illustrate each term:
> Consider an auction with $n=2$ bidders and $K=5$ identical units. There, $m=2$, and the submitted bids are: $\mathbf{b}_1 = \langle (5, 2), (3, 3) \rangle = [5, 5, 3, 3, 3]$, $\mathbf{b}_2 = \langle (4, 2), (2, 2) \rangle = [4, 4, 2, 2]$. Each entry in $\mathbf{b}_1$ and $\mathbf{b}_2$ corresponds to a single bid.
>
> The sorted list of all bids is: $[5, 5, 4, 4, 3, 3, 3, 2, 2]$. The top $K = 5$ bids (i.e., the winning bids) are $[5, 5, 4, 4, 3]$. The clearing price (i.e., the $5$th highest bid) is $3$. Here, the 2nd smallest winning bid is $4$.
>
> **Re “removing weakly dominated strategies”:** By the definition in Line 256-257, If a safe strategy $\mathbf{b}$ is weakly dominated by another safe strategy $\mathbf{b}'$—that is, $\mathbf{b}'$ yields at least as much value as $\mathbf{b}$ for **every** competing bid profile, —then a RoI-constrained value-maximizing bidder can safely ignore $\mathbf{b}$. Therefore, we remove such strategies from the class of safe bidding strategies.
> Keeping them does not improve the bidder’s performance, but removing them yields a **finite** strategy class, which significantly simplifies the design of online learning algorithms.
>
> **Re the unsafe and weakly dominant strategies:** To clarify: **unsafe strategies and weakly dominant strategies are conceptually unrelated** as stated below.
>
> Per Def 2 (Lines 243-251), a safe strategy satisfies the RoI constraint for **every** possible competing bid profile; otherwise, it is not safe (or as the reviewer described,  “unsafe”). Thus, safe and unsafe strategy classes are **disjoint**.
>
> Within the **safe class**, we further refine the strategy space by removing weakly dominated strategies—those that are never better and sometimes worse than another safe strategy. The notion of weak dominance is therefore defined **only within the safe class**. Considering this, **an unsafe strategy cannot be weakly dominated or weakly dominant** based on our definition in Lines 253–255.
>
> **Re inputs and Outputs of Alg. 2:** The output of Alg. 2 is the updated value $\varphi^t(e)$ for all edges $e$.
> The inputs are: the structure of the DAG, for $t \geq 2$, the values $\Gamma^{t-1}(v)$, $\varphi^{t-1}(e)$, and $\mathsf{w}^{t-1}(e)$ for all nodes $v$ and edges $e$. For $t = 1$ (initialization), only the DAG structure is required.
>
> **Re $\mathbf{w}$ in Thm 4.2 and Alg. 1:** $\mathbf{w}\in\mathbb{R}^M$ is the average cumulative valuation vector where $w_j=\frac{1}{j}\sum_{k=1}^j v_k$, i.e., the $j$-th entry is average of the first $j$ entries of the valuation vector as defined earlier in Line 118.

---

> > ### Comment · Reviewer_gJHK · 2025-04-02
> >
> > Thanks for the answers
> >
> > Considering
> > > Re the comment on missing terms/variables: Could the reviewer clarify which terms or variables are unclear?
> >
> > In fact, I've listed the ones in the section titled "Questions For Authors". So, you have clarified all of them. However, my main point w.r.t. it is to highlight that text in the lines can be improved. Hope for seeing in a reviewed version of the manuscript.
> >
> > All questions have been anwered clearly for me.

---

### Official Review · Reviewer_9dTJ · 2025-03-14

**Overall Recommendation:** 3

**Summary:**

This paper introduces the notion of safe bidding strategies for value-maximizing buyers in uniform price multi-unit auctions, ensuring return-on-investment (RoI) constraints are met. A value-maximizing buyer aims to maximize the received value, while only factors in the payment in the RoI constraint. In a uniform price multi-unit auction, the auctioneer sells $K$ identical units of a single good to buyers who may demand multiple units at decreasing marginal values, with the per-unit price set at the $K$-th highest bid. The private type for each buyer is then a vector of values describing the decreasing marginal value for the $k$-unit. To reduce the exponentially large bidding space, in practice, buyers adopt an $m$-uniform bidding format, where they submit $m$ bid-quantity pairs $(b_i, q_i)$ to demand $q_i$ units at bid $b_i$ instead of the vector of bids for each additional unit.

This work characterizes a finite class of safe strategies and develops a polynomial-time algorithm to learn the optimal safe strategy with sublinear regret. The paper also evaluates the performance of safe strategies against a clairvoyant with a richer class of strategies, computing the optimal richness ratio $\alpha$.  Notably, when the clairvoyant selects the optimal bidding from the class of strategies that are RoI-feasible (not necessarily safe) and have at most $m$ bid-quantity pairs, the richness ratio $\alpha$ is $1/2$, independent of $m$.

**Claims And Evidence:**

The analysis and proof in this work is comprehensive and non-trivial.

**Essential References Not Discussed:**

I don’t know any.

**Experimental Designs Or Analyses:**

I didn’t check carefully as this is mainly a theory paper.

**Methods And Evaluation Criteria:**

The main results from this work are i) characterization of the safe bidding strategies, ii) poly-time learning to bid algorithms, and iii) various bounds proved on regret and richness ratios. Although there are some complementary empirical sections, the most technical part seems to be the theoretical ones.

**Other Comments Or Suggestions:**

See above.

**Other Strengths And Weaknesses:**

The theoretical contribution of this work is non-trivial and the mount is pretty impressive. I see one main shortcoming of this work is that the paper seems to be too dense, and the presentation could be largely improved. For example, in the “Learning to Bid in Repeated Settings” paragraph in Section 2.1, the space of the bid is undefined, and only gets further discussed in “Bidding Language” on the next page. For the repeated setting, it remains unclear to me whether those values are identical across rounds or re-drawn at the beginning of each round. It is also confusing whether the $K$ unit of goods are sold within one round or spread across multiple rounds. It is also confusing whether the RoI constraint is imposed per round or across multiple rounds. If I understand correctly, both the $K$ unit of goods and the RoI constraints are well-defined within each round, and only the learning to bid algorithm needs to span over multiple rounds. If that is the case, it might be better to first describe the complete auction, including RoI constraints, etc. Only after the auction is completely defined, state the learning to bid problem on top of the underlying repeated uniform price auctions.

I would have given a higher score if this work is better organized.

**Questions For Authors:**

No further questions.

**Relation To Broader Scientific Literature:**

The proof of bounds might be of broader interests.

**Theoretical Claims:**

I didn’t verify all the details, but the theoretical results look sound to me. The amount of work is really impressive (~25 pages of different proofs).

---

> ### Author Rebuttal · Authors · 2025-03-31
>
> We thank the reviewer for their positive feedback. We're glad the theoretical depth and effort came through. The changes suggested by the reviewer are easily manageable with the additional page allowed for the camera-ready version, and we believe they will improve the organization of the manuscript.
>
> **Regarding paper being dense:** The density of the presentation is partly due to the space constraints given the breadth of our results. We clarify the questions the reviewer has raised and present a plan to update the model section for better readability.
>
> **Regarding the notion of bids:** Each bidder submits a vector of bids $\mathbf{b} \in \mathbb{R}^M$ using the $m$-uniform bidding format:  $$\mathbf{b} = \langle (b_1, q_1), \dots, (b_m, q_m) \rangle.$$
> That is, the bidder bids $b_1$ for the first $q_1$ units (i.e., the first $q_1$ entries of the vector are $b_1$), $b_2$ for the next $q_2$ units (i.e., the next $q_2$ entries are $b_2$), and so on. This description is included in the introduction (Line 50-52, right side), and as the reviewer suggested, we will add further clarification in Section 2.1.
>
> **Regarding valuations:** In the repeated setting, the bidder's valuation curve remains fixed across rounds, which is standard in the literature on learning in multi-unit auctions (e.g., [1, 2, 3]). The bidder is allowed to submit different bidding strategies in each round to minimize (approximate) regret over time, which is ensured to be sublinear. We will make this assumption explicit in the camera-ready version.
>
> [1] Brânzei, S., et al. Learning and Collusion in Multi-Unit Auctions. NeurIPS, 2023.
>
> [2] Galgana, R. and Golrezaei, N. Learning in Repeated Multiunit Pay-as-Bid Auctions. MSOM, 2024.
>
> [3] Potfer, M., et al. Improved Learning Rates in Multi-Unit Uniform Price Auctions. NeurIPS, 2024.
>
> **Regarding the number of sold units and RoI constraints:**  The reviewer’s understanding is correct. In each round, $K$ units of the goods are sold independently (see Lines 143–147, right column), and the RoI constraint is imposed on a per-round basis (see Remark 2.1, Lines 165–187, left column, for a detailed discussion).
>
> **Regarding Section 2.1:** We thank the reviewer for the helpful suggestion. In the camera-ready version, we will restructure Section 2.1 into three subsections: (1) preliminaries, including notations and terminology; (2) the standalone multi-unit auction format, including allocation and payment rules, the $m$-uniform bidding language, and the per-round RoI constraints; and (3) the repeated setting, introducing the new notations and formally stating the learning-to-bid problem and performance metrics.

---

### Official Review · Reviewer_zN7V · 2025-03-20

**Overall Recommendation:** 3

**Summary:**

This paper focus on one buyer’s bidding strategy in repeated uniform price multi-unit auctions. The buyer aims to maximize value under RoI constraints in each round. The authors restrict the buyers to adopt an $m$-uniform bidding format and introduce the notion of safe bidding strategies, which ensure that RoI constraints satisfied regardless of other buyers’ bids. They characterize a class of safe strategies and show that computing the optimal m-uniform safe bidding strategy is equivalent to finding the maximum weight path in the directed acyclic graph, simplifying the computational problem significantly. Building on this, the authors propose two online algorithm, the one under full-information feedback and the other one under bandit feedback.

**Claims And Evidence:**

I find all the claims clear and convincing.

**Essential References Not Discussed:**

All the necessary references are properly discussed.

**Experimental Designs Or Analyses:**

There is only one experiment, which looks convincing to me.

**Methods And Evaluation Criteria:**

The proposed methods and evaluation criteria look reasonable to me.

**Other Comments Or Suggestions:**

I do not have other comments.

**Other Strengths And Weaknesses:**

Strengths:
1. The paper characterized the minimal class of safe bidding strategies and show that the solution of the optimal safe bidding strategies is equivalent to finding the maximum weight path in the directed acyclic graph. This reduction simplifies the problem and provides a clear computational pathway for finding optimal strategies within this class.
2. The paper proposed two online algorithms achieving sub-linear regret under full-information feedback and bandit feedback respectively.

Weaknesses
While the focus on safe bidding strategies provides valuable insights, the performance of the proposed strategies diminishes when considering a larger class of bidding strategies. In particular, Thm 5.4 and Thm 5.5 indicate that when $m’$ is significantly larger than $m$, the term $m’/m-\sigma$ becomes very large. Thus, the performance of safe bidding strategy is worse comparing to a larger class of bidding strategies. Consequently, the effectiveness of the proposed algorithms may be limited when extended beyond the specific class of safe bidding strategies.

**Questions For Authors:**

Is it possible to solve the issue mentioned in the weaknesses above?

**Relation To Broader Scientific Literature:**

The paper may only interest a small group of researchers (the mechanism design community and the bandit algorihtms community).

**Theoretical Claims:**

I did not check the proofs. But the theoretic results align with my understanding of the techniques used and many similar results appear in the literature.

---

> ### Author Rebuttal · Authors · 2025-03-31
>
> We thank the reviewer for their encouraging feedback. We're glad the reduction to the maximum weight path in the directed acyclic graph and regret guarantees were clear and appreciated.
>
> **Regarding the comment that the paper might only interest a small group of researchers:** We believe that the paper appeals to both practitioners and theorists. The key motivation of this work is emission permit auctions which are becoming popular to curb industry emissions. We provide a simple class of strategies and a learning algorithm that is robust to much stronger benchmarks (Section 5) and pliable to more nuanced bidder behaviors (Section 6).  Moreover, several proof techniques, specifically in Section 5, are novel and might be of broader interest, as pointed out by Reviewer 9dTj.
>
> **Regarding the comment on the performance drop of safe strategies compared to a larger class of bidding strategies**, we would appreciate clarification on what is meant by the “larger class.”
>
> **Comparison to $m$-Uniform RoI-Feasible Non-Safe Strategies.** If the reviewer is referring to the class of $m$-uniform RoI-feasible bidding strategies that are not safe, the performance drop of safe strategies compared to this larger class of strategies (chosen by the clairvoyant) is natural and unavoidable: the bidder (learner) must ensure that RoI constraints are satisfied **without knowing the competing bids** and therefore resorts to safe strategies, while the clairvoyant can choose an optimal strategy with full knowledge of the competing bids.
>
>  - **Performance Gap is Bounded.**  We would like to highlight that the performance gap, as stated in Theorem 5.3, is **bounded and independent of m**. Moreover, this bound is tight (please see Appendix F.1 and F.2) implying that it can not be theoretically improved.
>
>  - **Empirical Performance Gap is Even Smaller.** Empirically, as illustrated in Figure 3 (left side), in practice, the optimal RoI feasible strategy that is not safe achieves at most 1.25x of the value obtained by the optimal safe strategy; significantly better than the theoretical bound of 2.
>
> **On the Gap Between $m$- and $m'$-Uniform Strategies.** We thank the reviewer for highlighting the potential performance gap when the bidder is restricted to $m$-uniform strategies, while the clairvoyant benchmark selects from a richer set of $m'$-uniform strategies with $m' > m$. **Below, we explain why this gap is expected, how our theoretical bounds characterize it tightly, and why it is unlikely to be a concern in practice.**
>
>   - **The Choice of $m$ is Flexible.** We would like to clarify that the choice of $m$ (i.e., the bid granularity) is made by the bidder and is not fixed. In other words, the bidder is not constrained to use a small value of $m$. While a larger $m$ offers the potential to improve performance, it also increases the size of the action space and the time it takes to learn the optimal $m$-uniform bidding strategy (the space/time complexity of the learning algorithm is $O(mM^2)$ as stated in Appendix C.3.3).
>
>  - **Our Theoretical Bounds are Tight.** The upper bounds in Theorems 5.4 and 5.5 are tight, e.g., for Theorem 5.4, for any $\delta \in (0, 1]$, we construct a problem instance for which the ratio of the value obtained by the optimal safe bidding strategy with at most $m’$ bid-quantity pairs to the value obtained by the optimal safe strategy chosen with at most $m$ bid-quantity pairs is at least $\frac{m'}{m} - \delta.$
>
> Please refer to Appendix D.5 and D.6 for the formal proofs and Appendix F.3 and F.4 for the construction of these instances. Thus, theoretically, it is not possible to improve this bound. In practice, bidders typically use small values of $m$, such as 4 or 5 (Line 173-175, right column), which we believe are reasonable trade-offs.
>
>   - **Theoretical Worst-Case is Rare in Practice.** As shown in Appendix F.3 and F.4, the problem instances that attain the worst-case upper bounds are highly nontrivial, and their constructions are quite intricate. For instance, the valuation vector must take the form $ \mathbf{v} = [1, 1 - \epsilon, \ldots, 1 - \epsilon]$ with $ \epsilon = O(\frac{m \delta}{m'}),$ and parameters such as  $M = K = T = O(N^{m'})$ where $N = O(\frac{m'}{\delta}).$ The competing bids also need to be carefully designed across rounds. **This level of intricacy strongly suggests that such worst-case gaps are unlikely to arise in realistic settings.**
>
>   - **Empirical Gaps are Much Smaller.** Our experimental results corroborate this insight. In particular, Figure 3 (right) illustrates the empirical gap when fixing $m = 1$ and varying $m'$. While Theorem 5.4 provides an upper bound that grows linearly with $m'$, in practice, the observed gains plateau quickly. For instance, increasing $m'$ to 10 results in only a $\sim$1.25x improvement—far below the 10x theoretical bound. This demonstrates that while the theoretical worst-case exists, it is rarely encountered in practical scenarios.

---

### Decision · Program_Chairs · 2025-05-01

**Decision:**

Accept (poster)

**Comment:**

The paper studies a value-maximizing, ROI-constrained buyer's bidding problem in repeated uniform price multi-unit auctions.  The reviewers all agreed that the paper is a valuable contribution, and the complaints were mostly about the presentation.  We encourage the authors to improve the paper accordingly based on the detailed suggestions and comments in the reviews.